# Safe Exploration Incurs Nearly No Additional Sample Complexity for Reward-free RL

**Ruiquan Huang**
The Pennsylvania State University
rzh5514@psu.edu

**Jing Yang**
The Pennsylvania State University
yangjing@psu.edu

**Yingbin Liang**
The Ohio State University
liang.899@osu.edu

## Abstract

Reward-free reinforcement learning (RF-RL), a recently introduced RL paradigm, relies on random action-taking to explore the unknown environment without any reward feedback information. While the primary goal of the exploration phase in RF-RL is to reduce the uncertainty in the estimated model with minimum number of trajectories, in practice, the agent often needs to abide by certain safety constraint at the same time. It remains unclear how such safe exploration requirement would affect the corresponding sample complexity in order to achieve the desired optimality of the obtained policy in planning. In this work, we make a first attempt to answer this question. In particular, we consider the scenario where a safe baseline policy is known beforehand, and propose a unified Safe reWard-frEe ExploraTion (SWEET) framework. We then particularize the SWEET framework to the tabular and the low-rank MDP settings, and develop algorithms coined Tabular-SWEET and Low-rank-SWEET, respectively. Both algorithms leverage the concavity and continuity of the newly introduced truncated value functions, and are guaranteed to achieve zero constraint violation during exploration with high probability. Furthermore, both algorithms can provably find a near-optimal policy subject to any constraint in the planning phase. Remarkably, the sample complexities under both algorithms match or even outperform the state of the art in their constraint-free counterparts up to some constant factors, proving that safety constraint hardly increases the sample complexity for RF-RL.

## 1 Introduction

Reward-free reinforcement learning (RF-RL) is an RL paradigm under which a learning agent first explores an unknown environment without any reward signal in the exploration phase, and then utilizes the gathered information to obtain a near-optimal policy for *any* reward function during the planning phase. Since formally introduced in Jin et al. (2020b), RF-RL has attracted increased attention in the research community (Kaufmann et al., 2021; Zhang et al., 2020; 2021; Wang et al., 2020; Modi et al., 2021). It is particularly attractive for applications where many reward functions may be of interest, such as multi-objective RL (Miryoosefi & Jin, 2021), or the reward function is not specified by the environment but handcrafted in order to incentivize some desired behavior of the RL agent (Jin et al., 2020b).

The ability of RF-RL to identify a near-optimal policy in response to an arbitrary reward function relies on the fact that the agent is allowed to explore any action during exploration. However, in practice, unrestricted exploration is often unrealistic or even harmful. In order to build safe, responsible and reliable artificial intelligence (AI), the RL agent often has to abide by certain application-dependent constraints, even during the exploration phase. Two motivating applications are provided as follows.

- **Autonomous driving.** In order to learn a near-optimal driving strategy, an RL agent needs to try various actions at different states through exploration. While RF-RL is an appealing approach as the reward function is difficult to specify, it is of critical importance for the RL agent to take safe actions (even during exploration) in order to avoid catastrophic consequences.

- **Cellular network optimization.** The operation of cellular network needs to take a diverse corpus of key performance indicators into consideration, which makes RF-RL a plausible solution. Meanwhile, the exploration also needs to meet certain system requirements, such as power consumption.

While meeting these constraints throughout the learning process is a pressing need for the broad adoption of RL in real-world applications, it is a mission impossible to accomplish if no other information is provided, as the learner has little knowledge of the underlying MDP at the beginning of the learning process and will inevitably take undesirable actions (in hindsight) and violate the constraints. On the other hand, in various engineering applications, there often exist either rule-based (e.g., autonomous driving) or human expert-guided (e.g., cellular network optimization) solutions to ensure safe operation of the system. One natural question is, *is it possible to leverage such existing safe solutions to ensure safety throughout the learning process? If so, how would the safe exploration requirement affect the corresponding RF-RL performances in terms of the sample complexity of exploration and the optimality and safety guarantees of the obtained policy in planning?*

To answer these questions, in this work, we introduce a new **safe RF-RL** framework. In the proposed safe RF-RL framework, the agent does not receive any reward information in the exploration phase, but is aware of a cost function associated with actions at a given state. We require that the cumulative cost in *each episode* is below a given threshold during exploration, with the aid of a pre-existing safe baseline policy $\pi^0$. The ultimate learning goal of safe RF-RL is to find a safe and near-optimal policy for any given reward and cost functions after exploration.

**Main contributions.** We summarize our main contributions as follows.

- First, we introduce a novel safe RF-RL framework that imposes safety constraints during both exploration and planning of RF-RL, which may have implications in various applications.
- Second, we propose a unified safe exploration strategy coined SWEET to leverage the prior knowledge of a safe baseline policy $\pi^0$. SWEET admits general model estimation and safe exploration policy construction modules, thus can accommodate various MDP structures and different algorithmic designs. Under the assumption that the approximation error function is concave and continuous in the policy space, SWEET is guaranteed to achieve zero constraint violation during exploration, and output a near-optimal safe policy for *any* given reward function and safety constraint under some assumptions in planning, both with high probability.
- Third, in order to facilitate the specific design of the approximation error function to ensure its concavity, we introduce a novel definition of truncated value functions. It relies on a new clipping method to avoid underestimation of the approximation error captured by the corresponding value function, and ensures the concavity of the resulted value function.
- Finally, we particularize the SWEET algorithm for both tabular and low-rank MDPs, and propose Tabular-SWEET and Low-rank-SWEET, respectively. Both algorithms inherit the optimality guarantee during planning, and the safety guarantees in both exploration and planning. Remarkably, the sample complexities under both algorithms match or even outperform the state of the art of their constraint-free counterparts up to some constant factors, proving that safety constraint incurs nearly no additional sample complexity for RF-RL.

## 2 PRELIMINARIES AND PROBLEM FORMULATION

### 2.1 EPISODIC MARKOV DECISION PROCESSES

We consider episodic Markov decision processes (MDPs) in the form of $\mathcal{M} = (\mathcal{S}, \mathcal{A}, P, H, s_1)$, where $\mathcal{S}$ is the state space and $\mathcal{A}$ is the finite action space, $H$ is the number of time steps in each episode, $P = \{P_h\}_{h=1}^H$ is a collection of transition kernels, and $P_h(s_{h+1}|s_h, a_h)$ denotes the transition probability from the state-action pair $(s_h, a_h)$ at step $h$ to state $s_{h+1}$ in the next step. Without loss of generality, we assume that in each episode of the MDP, the initial state is fixed at $s_1$. In addition, an MDP may be equipped with certain specified utility functions $u = \{u_h\}_{h=1}^H$, where we assume $u_h : \mathcal{S} \times \mathcal{A} \to [0, 1]$ is a deterministic function for ease of exposition.

A Markov policy $\pi$ is a set of mappings $\{\pi_h : \mathcal{S} \to \Delta(\mathcal{A})\}_{h=1}^H$, where $\Delta(\mathcal{A})$ is the set of all possible distributions over the action space $\mathcal{A}$. In particular, $\pi_h(a|s)$ denotes the probability of selecting action $a$ in state $s$ at time step $h$. We denote the set of all Markov policies by $\mathcal{X}$. For an agent adopting policy $\pi$ in an MDP $\mathcal{M}$, at each step $h \in [H]$ where $[H] := \{1, \dots, H\}$, she observes state $s_h \in \mathcal{S}$, and takes an action $a_h \in \mathcal{A}$ according to $\pi$, after which the environment transits to the next state $s_{h+1}$

with probability $P_h(s_{h+1}|s_h, a_h)$. The episode ends after $H$ steps, and we use a virtual state $s_{H+1}$ to denote the terminal state at step $H + 1$. We use $\mathbb{E}_{P,\pi}$ to denote the expectation of the distribution induced by the transition kernel $P$ and policy $\pi$.

Let $Q_{h,P,u}^\pi(s_h, a_h)$ and $V_{h,P,u}^\pi(s_h)$ be the corresponding action-value function and value function at step $h$, respectively, for a given collection of utility functions $u$. Then, $V_{h,P,u}^\pi(s_h) :=$ $\mathbb{E}_{P,\pi}\left[\sum_{h'=h}^H u_{h'}(s_{h'}, a_{h'})\big|s_h\right]$, and $Q_{h,P,u}^\pi(s_h, a_h) := \mathbb{E}_{P,\pi}\left[\sum_{h'=h}^H u_{h'}(s_{h'}, a_{h'})\big|s_h, a_h\right]$. We also use the shorthand $V_{P,u}^\pi$ to denote $V_{1,P,u}^\pi(s_1)$ due to the fixed initial state, and $P_h f(s_h, a_h) = \mathbb{E}_{s_{h+1}\sim P_h(\cdot|s_h, a_h)}\left[f(s_{h+1})\right]$ for any function $f: \mathcal{S} \to \mathbb{R}$. We further assume that the utility functions are normalized such that for any trajectory generated under a policy, the cumulative value over one episode is bounded by 1, i.e., $\sum_{h=1}^H u_h(s_h, a_h) \leq 1$.

## 2.2 SAFE REWARD-FREE REINFORCEMENT LEARNING

The safe policy considered in this work is formally defined as follows.

**Definition 1.** Given an MDP $\mathcal{M}^* = (\mathcal{S}, \mathcal{A}, P^*, H, s_1)$, a set of cost functions $c = \{c_h\}_{h=1}^H$ and $\tau \in (0, 1]$, a policy $\pi$ is $(c, \tau)$-safe if $V_{P^*,c}^\pi \leq \tau$.

Based on this definition of $(c, \tau)$-safe policies, we now elaborate the proposed safe RF-RL framework, which contains two phases. In the first phase of "**exploration**", the agent is required to efficiently explore the unknown environment without reward signals, and simultaneously not to violate a predefined safety constraint $(c, \tau)$ *in each episode* during this exploration phase. Let $\pi^{(n)}$ be the policy implemented in the $n$-th episode of the exploration. Then the agent's exploration should satisfy the safety constraint in every episode with high probability, namely,

$$\mathbb{P}\left[V_{P^*,c}^{\pi^{(n)}} \leq \tau, \forall n \in [N]\right] \geq 1 - \delta, \tag{1}$$

where $\delta \in (0, 1)$ and $N$ is the total number of episodes in the exploration phase. Note that the agent is only given a set of cost functions $c$ but not the reward $r$ in this phase. This is reasonable for many RL applications, where the purpose of exploration is not to maximize certain reward but to learn the environment, while the safety constraint need to be satisfied throughout the learning process.

In the second phase of "**planning**", the agent is given an *arbitrary* set of reward functions $r^*$ and a *new* set of safety constraint $(c^*, \tau^*)$. Without further exploration, she is required to learn an $\epsilon$-optimal policy $\bar{\pi}$ with respect to the given reward $r^*$, and subject to the safety constraint $(c^*, \tau^*)$.

**Definition 2.** Given an MDP $\mathcal{M}^* = (\mathcal{S}, \mathcal{A}, P^*, H, s_1)$, reward functions $r^*$, cost functions $c^*$ and $\tau^* \in (0, 1]$, $\bar{\pi}$ is an $\epsilon$-optimal $(c^*, \tau^*)$-safe policy if

$$V_{P^*,r^*}^{\pi^*} - V_{P^*,r^*}^{\bar{\pi}} \leq \epsilon, \text{ and } V_{P^*,c^*}^{\bar{\pi}} \leq \tau^*, \tag{2}$$

where $\epsilon \in (0, 1)$, and $\pi^*$ is the policy satisfying $\pi^* = \arg\max_\pi V_{P^*,r^*}^\pi$ s.t. $V_{P^*,c^*}^\pi \leq \tau^*$.

The design goal of safe RF-RL algorithms is *three-fold*: 1) to collect as few sample trajectories as possible, 2) to satisfy the safety constraint $(c, \tau)$ in the exploration phase, and 3) to obtain an $\epsilon$-optimal $(c^*, \tau^*)$-safe policy for any given reward $r^*$ and constraint $(c^*, \tau^*)$ in the planning phase.

We note that it is impossible to ensure zero constraint violation with high probability during exploration if an agent starts with no information about the system. Therefore, we make the assumption that a safe baseline policy is available to the learning agent during exploration. Besides, we also assume the constrained MDP always has enough feasible solutions, either during exploration or planning.

**Assumption 1** (Feasibility). The agent has knowledge of a baseline policy $\pi^0$ and $\kappa \in (0, \tau)$ such that $V_{P^*,c}^{\pi^0} \leq \tau - \kappa$. Besides, for any given constraint $(c, \tau)$ in exploration or planning phases, the safety margin, defined as $\Delta(c, \tau) := \tau - \min_\pi V_{P^*,c}^\pi$, is bounded away from zero, i.e. $\Delta(c, \tau) \geq \Delta_{\min} > 0$.

We remark that assuming the existence of a safe baseline policy is reasonable in practice. Many engineering applications already have existing solutions deployed and verified to be safe, although their reward performances may not necessarily be near-optimal. Such solutions can naturally serve as the baseline for safe RF-RL. Additionally, there are practical ways to construct safe baseline policies, e.g., via imitation learning using expert demonstrations, or via policy gradient algorithms to reduce the cost value function to be below the required safety threshold. This assumption is also widely adopted in the safe RL literature (see Section 6 for more discussions).

## 3 The SWEET framework

Compared with constraint-free RF-RL, the additional safety requirements during both exploration and planning bring two main **challenges** in the design of safe RF-RL algorithms. First, in order to obtain an $\epsilon$-optimal policy for *any* given reward during planning, it requires all actions to be sufficiently covered in the exploration phase. In particular, uniform action selection is one of the enablers for reward-free exploration when the state space is undesirably large (Agarwal et al., 2020; Uehara et al., 2021; Modi et al., 2021). On the other hand, the predefined safety constraint $(c, \tau)$ may preclude the agent from taking certain actions in exploration, which may affect the estimation accuracy of the environment and degrade the optimality of the output policy in planning. This dilemma requires a novel design to balance safety and state-action space coverage during exploration. Second, there may exist *safety constraint mismatch* between exploration and planning. Intuitively, the information obtained under a given set of constraint $(c, \tau)$ during exploration may not provide enough coverage for the optimal policy under another set of constraint $(c^*, \tau^*)$ during planning. How to design the safe exploration algorithm to handle such constraint mismatch is non-trivial.

In this section, we introduce a unified framework for safe reward-free exploration, termed as SWEET. We will show that the general framework achieves the second and third design objectives, i.e., safe exploration, and $\epsilon$-optimality and $(c^*, \tau^*)$-safety for the output policy in planning. The first design objective, i.e., low sample complexity for exploration, is dependent on the underlying MDP structure and will be investigated in Section 4 and Section 5 for tabular MDPs and low-rank MDPs, respectively.

### 3.1 Algorithm design

The SWEET framework relies on several key design components, namely, the $(\epsilon_0, t)$-*greedy policy*, the *approximation error function*, and the *empirical safe policy set*, as elaborated below.

**Definition 3** $((\epsilon_0, t)$-greedy policy). Given $\epsilon_0 \in (0, 1)$ and $t \in \{0, 1, \cdots, H\}$, $\pi'$ is an $(\epsilon_0, t)$-greedy version of $\pi$ if there exists $\mathcal{H} \subset [H]$ with $|\mathcal{H}| = t$ such that $\pi'_h = \pi_h$ for all $h \notin \mathcal{H}$, and

$$\pi'_h(a|s) = (1 - \epsilon_0)\pi_h(a|s) + \epsilon_0/|\mathcal{A}|, \forall h \in \mathcal{H}, s \in \mathcal{S}, a \in \mathcal{A}.$$

Essentially, under an $(\epsilon_0, t)$-greedy version of a given policy $\pi$, the agent follows policy $\pi$ except for $t$ out of $H$ steps, at which with probability $\epsilon_0$, she takes actions uniformly at random from the state space $\mathcal{A}$. One critical property of the $(\epsilon_0, t)$-greedy policy is that, the difference between the value functions under the $(\epsilon_0, t)$-greedy policy and its original policy is bounded by $\epsilon_0 t$ for any normalized utility function (See Lemma 1 in Appendix A).

The *approximation error function* $\mathtt{U}(\hat{P}, \pi)$ measures the uncertainty in the model estimate $\hat{P}$ under a policy $\pi$. Specifically, for a given MDP $\mathcal{M}^*$, $\mathtt{U}(\hat{P}, \pi)$ upper bounds the value function difference under $\hat{P}$ and $P^*$, i.e. $\mathtt{U}(\hat{P}, \pi) \geq \max_u |V^\pi_{\hat{P}, u} - V^\pi_{P^*, u}|$, where $u$ is a *normalized* utility function.

The *empirical safe policy set*, which is critical for constructing safe exploration policies, is defined as

$$\mathcal{C}_{\hat{P}, \mathtt{U}}(\tilde{\kappa}, \epsilon_0, t) = \begin{cases} \{\pi^0\}, & \text{if } V^{\pi^0}_{\hat{P}, c} + \mathtt{U}(\hat{P}, \pi^0) \geq \tau - \epsilon_0 t - \tilde{\kappa}, \\ \left\{\pi : V^\pi_{\hat{P}, c} + \mathtt{U}(\hat{P}, \pi) \leq \tau - \epsilon_0 t\right\}, & \text{otherwise}, \end{cases} \tag{3}$$

where $\tilde{\kappa}, \epsilon_0$ and $t$ are constants satisfying the condition that $\tau - \epsilon_0 t - \tilde{\kappa} > \tau - \kappa$.

The intuition behind the construction of the empirical safe policy can be explained as follows (Liu et al., 2021): if $V^{\pi^0}_{\hat{P}, c} + \mathtt{U}(\hat{P}, \pi^0) \geq \tau - \epsilon_0 t - \tilde{\kappa}$, it indicates that $\hat{P}$ is not sufficiently accurate. Thus, the empirical safe policy set only contains the safe baseline policy $\pi^0$. On the other hand, if $V^{\pi^0}_{\hat{P}, c} + \mathtt{U}(\hat{P}, \pi^0) < \tau - \epsilon_0 t - \tilde{\kappa}$, which happens when $\mathtt{U}(\hat{P}, \pi^0)$ is sufficiently small, it indicates that $\hat{P}$ is sufficiently accurate on $\pi^0$. Then, we relax the constraint on $V^\pi_{\hat{P}, c} + \mathtt{U}(\hat{P}, \pi)$ from $\tau - \epsilon_0 t - \tilde{\kappa}$ to $\tau - \epsilon_0 t$ to include $\pi^0$ and other policies in the empirical safe policy set. Since $V^\pi_{\hat{P}, c} + \mathtt{U}(\hat{P}, \pi)$ is an upper bound of the true value $V^\pi_{P^*, c}$ for any $\pi$, it ensures that $V^\pi_{P^*, c} < \tau - \epsilon_0 t$ for all $\pi$ included in $\mathcal{C}_{\hat{P}, \mathtt{U}}$. Moreover, all $(\epsilon_0, t)$-greedy versions of such policies satisfy the safety constraint $(c, \tau)$.

With those salient components, SWEET proceeds as follows. At the beginning of each episode, the agent executes a set of *behavior policies*, which are $(\epsilon_0, t)$-greedy versions of a *reference policy*

$\pi_r$ obtained in the previous episode. For the first episode, the reference policy would be $\pi^0$. The general construction of $\pi_r$ will be elaborated below. By collecting trajectories generated under the behavior policies, the agent updates the estimated model $\hat{P}$ and the corresponding approximation error function $\mathtt{U}(\hat{P}, \cdot)$.

The agent then seeks a reference policy $\pi_r$ that maximizes the approximation error $\mathtt{U}(\hat{P}, \pi)$ within the constructed empirical safe policy set. Intuitively, $\mathtt{U}(\hat{P}, \pi)$ is an upper bound of certain distance of distributions over trajectories induced by $\pi$ under $\hat{P}$ and $P^*$. Therefore, $\pi_r$ induces a distribution that captures the most uncertainty in $\hat{P}$. Choosing $\pi_r$ thus reduces the uncertainty in $\hat{P}$ in a greedy fashion. If $\mathtt{U}(\hat{P}, \pi_r)$ is less than a termination threshold $\mathtt{T}$ defined in SWEET, it indicates that the estimated model $\hat{P}$ is sufficiently accurate for the planning task. The exploration phase then terminates. Otherwise, the agent continues to the next episode with the new $\pi_r$.

After termination, SWEET enters the planning phase and receives arbitrary reward functions $r^*$ and a safety constraint $(c^*, \tau^*)$. The agent utilizes $\hat{P}$ to compute a policy $\bar{\pi}$, which maximizes $V^\pi_{\hat{P}, r^*}$ subject to an empirical safety constraint $V^\pi_{\hat{P}, c} + \mathtt{U}(\hat{P}, \pi) \leq \tau^*$. Algorithm 1 has the detail of SWEET.

---

**Algorithm 1** SWEET (**S**afe Re**W**ard Fr**E**e **E**xplora**T**ion)

---

1: **Input:** Reference policy $\pi_r = \pi^0$, uncertainty function $\mathtt{U}$, $\epsilon_0, t, \tilde{\kappa}$ and $\mathtt{T}$.
2: // Exploration:
3: **while** TRUE **do**
4:     Construct a set of $(\epsilon_0, t)$-greedy policies of $\pi_r$ (see Definition 3) and use them to collect data;
5:     Model estimation: Update $\hat{P}$ using collected data;
6:     Obtain $\pi_r = \arg\max_{\pi \in \mathcal{C}_{\hat{P}, \mathtt{U}}(\tilde{\kappa}, \epsilon_0, t)} \mathtt{U}(\hat{P}, \pi)$ where $\mathcal{C}_{\hat{P}, \mathtt{U}}(\tilde{\kappa}, \epsilon_0, t)$ is defined in Equation (3);
7:     **if** $V^{\pi^0}_{\hat{P}, c} + \mathtt{U}(\hat{P}, \pi^0) \leq \tau - \epsilon_0 t - \tilde{\kappa}$ and $\mathtt{U}(\hat{P}, \pi_r) \leq \mathtt{T}$ **then**
8:         Output $\hat{P}$; **break;**
9:     **end if**
10: **end while**
11: // Planning:
12: Receive reward function $r^*$ and a safety constraint $(c^*, \tau^*)$.
13: **Output:** $\bar{\pi} = \arg\max_\pi V^\pi_{\hat{P}, r^*}$    s.t.    $V^\pi_{\hat{P}, c^*} + \mathtt{U}(\hat{P}, \pi) \leq \tau^*$.

---

### 3.2 THEORETICAL ANALYSIS

Before we present the theoretical guarantee for SWEET, we first introduce the notion of mixture policies and equivalent policies and characterize the concavity over Markov policy space.

**Definition 4.** Given two Markov policies $\pi, \pi' \in \mathcal{X}$, we use $\gamma\pi \oplus (1 - \gamma)\pi'$ to denote the mixture policy that uses $\pi$ with probability $\gamma$ and uses $\pi'$ with probability $1 - \gamma$ during an episode.

**Definition 5.** Given an MDP $\mathcal{M}$, two policies, including mixture policies, are equivalent if they induce the same marginal distribution over any state-action pair $(a, s)$ in any step $h \in [H]$.

By Theorem 6.1 in Altman (1999), for any mixture policy $\gamma\pi \oplus (1 - \gamma)\pi'$, there exists an equivalent Markov policy $\pi^\gamma(\pi, \pi') \in \mathcal{X}$. For ease of presentation, in the following, we simply use $\pi^\gamma$ to denote it when the definition is clear from the context. Therefore, the Markov policy space $\mathcal{X}$ is equipped with an abstract convexity by mapping all mixture policies to their equivalent Markov policies in $\mathcal{X}$. With this convexity, we can define concave functions on $\mathcal{X}$ as follows.

**Definition 6.** A function $\mathtt{f} : \mathcal{X} \to [0, 1]$ is concave and continuous on the Markov policy space $\mathcal{X}$ if for any $\pi, \pi' \in \mathcal{X}$ and $\gamma \in [0, 1]$, $\mathtt{f}(\pi^\gamma) \geq \gamma\mathtt{f}(\pi) + (1 - \gamma)\mathtt{f}(\pi')$, and is continuous in $\gamma \in [0, 1]$.

With Definition 6, we have the following result of SWEET.

**Theorem 1** ($\epsilon$-optimality and safety guarantee of SWEET). *Given an MDP $\mathcal{M}^*$ and model estimate $\hat{P}$, assume $\mathtt{U}(\hat{P}, \pi)$ is concave and continuous over the Markov policy space $\mathcal{X}$ and $\left|V^\pi_{P^*, u} - V^\pi_{\hat{P}, u}\right| \leq \mathtt{U}(\hat{P}, \pi)$ for any normalized utility $u$ and policy $\pi$, and Assumption 1 holds. Let $\epsilon_0, t$ and $\tilde{\kappa}$ be constants that satisfy $\epsilon_0 t + \tilde{\kappa} < \kappa$. Let $\mathfrak{U} = \min\left\{\frac{\epsilon}{2}, \frac{\Delta_{\min}}{2}, \frac{\epsilon\Delta_{\min}}{5}, \frac{\tau - \epsilon_0 t}{4}, \frac{\tilde{\kappa}(\Delta(c, \tau) - \epsilon_0 t - \tilde{\kappa})}{4(\Delta(c, \tau) - \epsilon_0 t)}\right\}$, and*

$\mathtt{T} \leq (\Delta(c, \tau) - \epsilon_0 t)\mathfrak{U}/2$ *be the termination condition of SWEET. If SWEET terminates in finite episodes, then, the following statements hold:*

*(i) The exploration phase is safe.*

*(ii) The output $\bar{\pi}$ of SWEET in the planning phase is an $\epsilon$-optimal $(c^*, \tau^*)$-safe policy.*

The detailed proof of Theorem 1 is deferred to Appendix A. We highlight the main idea behind the analysis as follows. While the construction of $\mathcal{C}_{\hat{P},\mathtt{U}}(\tilde{\kappa}, \epsilon_0, t)$ ensures safe exploration, the ability for SWEET to find an $\epsilon$-optimal $(c^*, \tau^*)$-safe policy in planning relies on the concavity and continuity of $\mathtt{U}(\hat{P}, \cdot)$. Note that when SWEET terminates, it is only guaranteed that the approximation error $\mathtt{U}(\hat{P}, \pi)$ is upper bounded by $\mathtt{T}$ for the policies within $\mathcal{C}_{\hat{P},\mathtt{U}}(\tilde{\kappa}, \epsilon_0, t)$. Due to a possibly different constraint in planning, it is desirable to have $\mathtt{U}(\hat{P}, \pi)$ sufficiently small under any $\pi$, so that the agent is able to achieve the learning goal in planning with the estimated model $\hat{P}$.

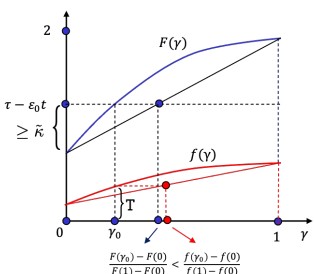

Figure 1: Illustration of the proof.

Let $\tilde{\pi} = \arg\max_\pi \mathtt{U}(\hat{P}, \pi)$, $\pi^\gamma$ be the equivalent Markov policy of $\gamma\tilde{\pi} \oplus (1 - \gamma)\pi^0$, $f(\gamma) = \mathtt{U}(\hat{P}, \pi^\gamma)$ and $g(\gamma) = V_{\hat{P},c}^{\pi^\gamma}$. Then, $f$ is concave and $g$ is linear in $\gamma$ by Theorem 6.1 in Altman (1999). Let $F(\gamma) = f(\gamma) + g(\gamma)$. The definition of $\mathcal{C}_{\hat{P},\mathtt{U}}(\tilde{\kappa}, \epsilon_0, t)$ ensures that $F(0) \leq \tau - \epsilon_0 t - \tilde{\kappa}$, and $F(\gamma) \leq \tau - \epsilon_0 t$ if $\pi^\gamma$ lies in $\mathcal{C}_{\hat{P},\mathtt{U}}$. It suffices to consider the case when $F(1) > \tau - \epsilon_0 t$, under which we can show that both $g$ and $f$ increase with $\gamma$. Then, the concavity of $f$ and linearity of $g$ ensure that $\frac{f(\gamma)-f(0)}{f(1)-f(0)} \geq \frac{F(\gamma)-F(0)}{F(1)-F(0)}$, as illustrated in Figure 1. Let $\pi^{\gamma_0}$ be the policy under which $F(\gamma_0) = \tau - \epsilon_0 t$. Then, $F(\gamma_0) - F(0) \geq \tilde{\kappa}$. Combining with the fact that $F(1) - F(0) \leq 2$, we have $f(1) \leq f(0) + (f(\gamma_0) - f(0))\frac{2}{\tilde{\kappa}} \leq 2\mathtt{T}/\tilde{\kappa}$, which provides an upper bound on $\mathtt{U}(\hat{P}, \pi)$ for any $\pi$.

### 3.3 TRUNCATED VALUE FUNCTION

Theorem 1 highlights the importance of the concave and continuous approximation error function on the Markov policy space. In the following, we introduce a prototype function, coined *truncated value function*, which is concave and continuous on the Markov policy space and can be used for the construction of the approximation error functions for both tabular and low-rank MDPs.

**Definition 7** (Truncated value function). Given an MDP $\mathcal{M}$, $\alpha > 0$, and a set of (un-normalized) utility functions $u$, the truncated value function $\bar{V}_{P,u}^{\alpha,\cdot} = \bar{V}_{1,P,u}^{\alpha,\cdot}(s_1) : \mathcal{X} \to \mathbb{R}$ is defined as follows:

$$\begin{cases} \bar{Q}_{h,P,u}^{\alpha,\pi}(s_h, a_h) = u(s_h, a_h) + \alpha P_h \bar{V}_{h+1,P,u}^{\alpha,\pi}(s_h, a_h), \\ \bar{V}_{h,P,u}^{\alpha,\pi}(s_h) = \min\left\{1, \mathbb{E}_\pi\left[\bar{Q}_{h,P,u}^{\alpha,\pi}(s_h, a_h)\right]\right\}, \end{cases} \tag{4}$$

where $V_{H+1,P,u}^{\alpha,\pi}(s_{H+1}) = 0$ and we omit the upper index $\alpha$ for simplicity when $\alpha = 1$.

It is worth noting that the clipping technique is applied to the value function as opposed to the action-value function, where the latter is more conventional in the existing literature. This new method is critical for achieving the superior sample complexity in safe RF-RL, as will be elaborated later. Meanwhile, it preserves the desired concavity and continuity (see Lemma 3 in Appendix A), which ensures the safety guarantee for both exploration and planning.

## 4 THE TABULAR-SWEET ALGORITHM

### 4.1 ALGORITHM DESIGN

Under tabular MDPs, state space $\mathcal{S}$ and action space $\mathcal{A}$ are both finite (with sizes $S$ and $A$, respectively). We instantiate the modules of model estimation and exploration policy construction of the SWEET framework, and specify the approximation error function and parameter selection as follows. The details of the Tabular-SWEET algorithm is shown in Algorithm 2 in Appendix B.

**Model estimation.** At each episode $n$, the agent uses $\pi^{(n-1)}$, which is the reference policy derived from the last episode $n-1$, to collect a trajectory $\{s_1^{(n)}, a_1^{(n)}, \dots, s_H^{(n)}, a_H^{(n)}\}$. For that, we set $\epsilon_0$ and

$t$ as 0, i.e., it is essentially a $(0, 0)$-greedy version of policy $\pi^{(n-1)}$. We note that although $\pi^{(n-1)}$ is a greedy policy, the uncertainty captured by the approximation error function $\mathtt{U}(\hat{P}, \pi)$ will guide the agent to explore the uncertain state-action pairs and obtain sufficient coverage for the entire space.

The agent then adds new data triples $\{s_h^{(n)}, a_h^{(n)}, s_{h+1}^{(n)}\}_{h=1}^H$ to a maintained dataset $\mathcal{D}$. Let $N_h^{(n)}(s_h, a_h) = \sum_{m=1}^n \mathbb{1}\{s_h^{(n)} = s_h, a_h^{(n)} = a_h\}$ and $N_h^{(n)}(s_h, a_h, s_{h+1}) = \sum_{m=1}^n \mathbb{1}\{s_h^{(n)} = s_h, a_h^{(n)} = a_h, s_{h+1}^{(n)} = s_{h+1}\}$ be the visitation counters. The agent estimates $\hat{P}_h^{(n)}(s_{h+1}|s_h, a_h)$ as $\frac{N_h^{(n)}(s_h, a_h, s_{h+1})}{N_h^{(n)}(s_h, a_h)}$ if $N_h^{(n)}(s_h, a_h) > 1$ and as $\frac{1}{S}$ otherwise.

**Approximation error function.** Inspired by Ménard et al. (2021), we adopt an uncertainty-driven virtual reward function $\hat{b}_h^{(n)}(s_h, a_h) = \frac{\beta_0 H}{N_h^{(n)}(s_h, a_h)}$ to guide the exploration, where $\beta_0$ is a fixed parameter. Let $\alpha_H = 1 + 1/H$. Then, the approximation error function is specified as $\mathtt{U}^{(n)}(\pi) := 4\sqrt{\bar{V}_{\hat{P}^{(n)}, \hat{b}^{(n)}}^{\alpha_H, \pi}}$. According to Lemma 3, $\mathtt{U}^{(n)}(\pi)$ is concave and continuous in $\pi$. Besides, as shown in Lemma 8 in Appendix B, we have $|V_{P^*, u}^\pi - V_{\hat{P}, u}^\pi| \le \mathtt{U}^{(n)}(\pi)$ for any normalized utility $u$, i.e., $\mathtt{U}^{(n)}(\pi)$ is a valid upper bound on the estimate error for the corresponding value function. The required properties of $\mathtt{U}$ in Theorem 1 are thus satisfied.

**Exploration policy.** To guarantee that the exploration is safe, we set $\tilde{\kappa} = \kappa/2$, and construct an empirical safety set $\mathcal{C}^{(n)} := \mathcal{C}_{\hat{P}^{(n)}, \mathtt{U}^{(n)}}(\kappa/2, 0, 0)$ (Equation (3)). Hence, the algorithm finds a policy $\pi^{(n)}$ used for the next episode, which is in the safe set $\mathcal{C}^{(n)}$ and maximizes the truncated value function $\bar{V}_{\hat{P}^{(n)}, \hat{b}^{(n)}}^{\alpha_H, \pi}$. The exploration phase stops at episode $n_\epsilon$ when $\mathtt{U}^{(n_\epsilon)}(\pi^{(n_\epsilon)}) \le \mathtt{T}$. The algorithm will utilize the model learned in episode $n_\epsilon$ to design an $\epsilon$-optimal policy with respect to arbitrary given reward $r^*$ and safety constraint $(c^*, \tau^*)$.

## 4.2 Theoretical Analysis

The theoretical guarantee of Tabular-SWEET is characterized in the theorem below, whose proof can be found in Appendix B.

**Theorem 2** (Sample complexity of Tabular-SWEET). *Given $\epsilon, \delta \in (0, 1)$, and safety constraint $(c, \tau)$, under Assumption 1, let $\mathfrak{U} = \min\left\{\frac{\epsilon}{2}, \frac{\Delta_{\min}}{2}, \frac{\epsilon\Delta_{\min}}{5}, \frac{\tau}{4}, \frac{\kappa}{16}\right\}$, and $\mathtt{T} = \Delta(c, \tau)\mathfrak{U}/2$ be the termination condition of Tabular-SWEET. Then, with probability at least $1 - \delta$, Tabular-SWEET achieves the learning objective of safe reward-free exploration (Equations (1) and (2)), and the number of trajectories collected in the exploration phase is at most $\tilde{O}\left(\frac{HSA(S+\log(1/\delta))}{\Delta(c, \tau)^2\mathfrak{U}^2} + \frac{HSA(S+\log(1/\delta))}{\kappa^2}\right)$.*

We discuss several possible scenarios and the corresponding selections of $\mathfrak{U}$ as follows.
- *Constraint-free RF-RL.* For this case $\Delta(c, \tau) = \Delta_{\min} = \kappa = 1$, and $c = 0$. Thus, $\mathfrak{U} = \Theta(\epsilon)$ and the sample complexity is $\tilde{O}\left(HS^2A/\epsilon^2\right)$, which matches the state of the art (Ménard et al., 2021).
- *Constraint-free planning.* If only safe exploration is required, we set $\mathfrak{U} = \Theta(\min\{\epsilon, \kappa\})$, and the sample complexity scales in $\tilde{O}\left(\frac{HS^2A}{\Delta(c, \tau)^2}\left(\frac{1}{\epsilon^2} + \frac{1}{\kappa^2}\right)\right)$. The blow-up factor $\frac{1}{\Delta(c, \tau)^2}$ depends on the safety margin, and the impact of baseline policy only appears in the $\epsilon$-independent term.
- *Constraint mismatch between exploration and planning.* For this case, we set $\mathfrak{U} = \Theta(\epsilon\Delta_{\min})$, and the sample complexity is at most $\tilde{O}\left(\frac{HS^2A}{\Delta(c, \tau)^2}\left(\frac{1}{\epsilon^2\Delta_{\min}^2} + \frac{1}{\kappa^2}\right)\right)$.

## 5 The Low-rank-SWEET Algorithm

### 5.1 Low-rank MDP

In this section, we present another SWEET variant for low-rank MDPs.

**Definition 8** (Low-rank MDP (Jiang et al., 2017; Agarwal et al., 2020; Uehara et al., 2021)). An MDP $\mathcal{M}$ is a low-rank MDP with dimension $d \in \mathbb{N}$ if for each $h \in [H]$, the transition kernel $P_h$ admits a $d$-dimensional decomposition, i.e., there exist two features $\phi_h : \mathcal{S} \times \mathcal{A} \to \mathbb{R}^d$ and $\mu_h : \mathcal{S} \to \mathbb{R}^d$ such that $P_h(s_{h+1}|s_h, a_h) = \langle\phi_h(s_h, a_h), \mu_h(s_{h+1})\rangle, \forall s_h, s_{h+1} \in \mathcal{S}, a_h \in \mathcal{A}$. Let $\phi = \{\phi_h\}_{h\in[H]}$ and $\mu = \{\mu_h^*\}_{h\in[H]}$ be the features for $P$. Then, $\|\phi_h^*(s, a)\|_2 \le 1, \|\int \mu_h^*(s)g(s)ds\|_2 \le \sqrt{d}, \forall(s, a) \in \mathcal{S} \times \mathcal{A}, \forall g : \mathcal{S} \to [0, 1]$.

Differently from linear MDPs (Wang et al., 2020; Jin et al., 2020b), low-rank MDP does not assume that the features $\phi$ are known a priori. The lack of knowledge on features in fact invokes a nonlinear structure, which makes it impossible to learn a model in polynomial time if there is no assumption on features $\phi$ and $\mu$. We hence adopt the following conventional assumption (Jiang et al., 2017; Agarwal et al., 2020; Uehara et al., 2021) from the recent studies on low-rank MDPs.

**Assumption 2** (Realizability). A learning agent can access a finite model class $\{(\Phi, \Psi)\}$ that contains the true model, i.e., $(\phi^*, \mu^*) \in \Phi \times \Psi$, where $\langle \phi_h^*(s_h, a_h), \mu^*(s_{h+1}) \rangle = P_h^*(s_{h+1}|s_h, a_h)$.

We note that finite model class assumption can be relaxed to the infinite case with bounded statistical complexity (Sun et al., 2019; Agarwal et al., 2020). Then, we present the following standard oracle as a computational abstraction, which is commonly adopted in the literature (Agarwal et al., 2020; Uehara et al., 2021).

**Definition 9** (MLE oracle). Given the model class $(\Phi, \Psi)$ and a dataset $\mathcal{D}$ of $(s_h, a_h, s_{h+1})$, the MLE oracle $\text{MLE}(\mathcal{D})$ takes $\mathcal{D}$ as the input and returns the following estimators as the output:

$$(\hat{\phi}_h, \hat{\mu}_h) = \text{MLE}(\mathcal{D}) = \arg \max_{\phi_h \in \Phi, \mu_h \in \Psi} \sum_{(s_h, a_h, s_{h+1}) \in \mathcal{D}} \log \langle \phi_h(s_h, a_h), \mu_h(s_{h+1}) \rangle.$$

## 5.2 Algorithm design

The instantiated SWEET algorithm, termed as Low-rank-SWEET, can be found in Algorithm 3 in Appendix C. It proceeds as follows. In each iteration $n$ during the exploration phase, the agent samples $H$ trajectories, indexed by $\{(n, h)\}_{h=1}^H$. During the $(n, h)$-th episode, the agent executes an $(\epsilon_0, 2)$-greedy version of the reference policy $\pi^{(n-1)}$, where $\epsilon_0 = \kappa/6$ and the $\epsilon_0$-greedy action selection only takes place at time steps $h$ and $h - 1$. Denote the trajectory collected in episode $(n, h)$ as $\{s_1^{(n,h)}, a_1^{(n,h)}, \dots, s_H^{(n,h)}, a_H^{(n,h)}\}$. The agent maintains a dataset $\mathcal{D}_h$ for each time step $h$, which is updated through $\mathcal{D}_h^{(n)} \leftarrow \mathcal{D}_h^{(n-1)} \cup \{s_h^{(n,h)}, a_h^{(n,h)}, s_{h+1}^{(n,h)}\}$. Note that both $s_h^{(n,h)}$ and $a_h^{(n,h)}$ are affected by the $\epsilon_0$-greedy action selection.

**Model estimation.** Then, the agent obtains the model estimate $\hat{P}^{(n)}$ through the MLE oracle:

$$(\hat{\phi}_h^{(n)}, \hat{\mu}_h^{(n)}) = \text{MLE}(\mathcal{D}_h), \text{ and } \hat{P}_h^{(n)}(s_{h+1}|s_h, a_h) = \langle \hat{\phi}_h^{(n)}(s_h, a_h), \hat{\mu}_h^{(n)}(s_{h+1}) \rangle. \tag{5}$$

**Approximation error function.** The algorithm will also use the estimated representation $\hat{\phi}_h^{(n)}$ to update the empirical covariance matrix $\hat{U}^{(n)}$ as

$$\hat{U}_h^{(n)} = \sum_{m=1}^n \hat{\phi}_h^{(n)}(s_h^{(m,h+1)}, a_h^{(m,h+1)})(\hat{\phi}_h^{(n)}(s_h^{(m,h+1)}, a_h^{(m,h+1)}))^\top + \lambda_n I. \tag{6}$$

It is worth noting that only $a_h^{(m,h+1)}$ is affected by the $\epsilon_0$-greedy action selection, which is different from the dataset augmentation step. Next, the agent uses both $\hat{\phi}_h^{(n)}$ and $\hat{U}^{(n)}$ to derive an exploration-driven virtual reward function as $\hat{b}_h^{(n)}(s, a) = \hat{\alpha} \|\hat{\phi}_h^{(n)}(s, a)\|_{(\hat{U}_h^{(n)})^{-1}}$ where $\|x\|_A := \sqrt{x^\top A x}$ and $\hat{\alpha}$ is a pre-determined parameter. As shown in Lemma 14 in Appendix C, the approximation error can be bounded by the truncated value function with factor $\alpha = 1$ up to a constant additive term, i.e., $|V_{P^*,u}^\pi - V_{\hat{P},u}^\pi| \leq \bar{V}_{\hat{P}^{(n)}, \hat{b}^{(n)}}^\pi + \sqrt{\tilde{A}\zeta/n} := \mathtt{U}_L^{(n)}(\pi)$, where "$L$" stands for "Low-rank".

**Exploration policy.** Based on SWEET, we choose $\tilde{\kappa} = \kappa/3$ such that $\tilde{\kappa} + \epsilon_0 t < \kappa$. Then, the algorithm defines the empirical safe policy set as $\mathcal{C}_L^{(n)} := \mathcal{C}_{\hat{P}^{(n)}, \mathtt{U}_L^{(n)}}(\kappa/3, \kappa/6, 2)$. It then finds a reference policy $\pi^{(n)}$ in $\mathcal{C}_L^{(n)}$ that maximizes $\mathtt{U}_L^\pi(\pi)$, which is used for exploration at the next iteration.

## 5.3 Theoretical analysis

We summarize the results of Low-rank-SWEET in Theorem 3, and defer the proof to Appendix C.

**Theorem 3** (Sample complexity of Low-rank-SWEET). *Given $\epsilon, \delta \in (0, 1)$, and safety constraint $(c, \tau)$, let $\mathfrak{U} = \min\left\{\frac{\epsilon}{2}, \frac{\Delta_{\min}}{2}, \frac{\epsilon\Delta_{\min}}{5}, \frac{\tau}{6}, \frac{\kappa}{24}\right\}$, and $\mathtt{T} = \Delta(c, \tau)\mathfrak{U}/3$ be the termination condition of Low-rank-SWEET. Then, under Assumption 1,2, with probability at least $1 - \delta$, Low-rank-SWEET achieves the learning objective of safe reward-free exploration (Equations (1) and (2)) and the number of trajectories collected in the exploration phase is at most $\tilde{O}\left(\frac{H^3 d^4 A^2 \log(1/\delta)}{\kappa^2 \Delta(c, \tau)^2 \mathfrak{U}^2} + \frac{H^3 d^4 A^2 \log(1/\delta)}{\kappa^4}\right)$.*

*Remark* 1. For the constraint-free scenario, we set $\Delta(c, \tau) = \Delta_{\min} = \kappa = 1$, $\mathfrak{U} = \Theta(\epsilon)$, and $c$ to be zero. Then, the sample complexity scales as $\tilde{O}\left(H^3 d^4 A^2/\epsilon^2\right)$, which outperforms the best known sample complexity of RF-RL (Agarwal et al., 2020; Modi et al., 2021) and even reward-known RL with computational feasibility (Uehara et al., 2021), all for low-rank MDPs.

## 6 RELATED WORKS

**Reward-free reinforcement learning.** Reward-free exploration is formally introduced by Jin et al. (2020a) for tabular MDP, where an algorithm called RF-RL-Explore is proposed, which achieves $\tilde{O}\left(H^3 S^2 A/\epsilon^2\right)$ sample complexity[1]. The result is then improved to $\tilde{O}\left(H^2 S^2 A/\epsilon^2\right)$ by Kaufmann et al. (2021). By leveraging an empirical Bernstein inequality, RF-Express (Ménard et al., 2021) achieves $\tilde{O}\left(H S^2 A/\epsilon^2\right)$ sample complexity, which matches the minimax lower bound in $H$ (Domingues et al., 2020). Zhang et al. (2020) considers the stationary case, and achieves $\tilde{O}\left(S^2 A/\epsilon^2\right)$ sample complexity, which is nearly minimax optimal. When structured MDPs are considered, Wang et al. (2020) studies linear MDPs and obtains $\tilde{O}\left(d^3 H^4/\epsilon^2\right)$ sample complexity, where $d$ is the dimension of feature vectors. Zhang et al. (2021) investigates linear mixture MDPs and achieves $\tilde{O}\left(H^3 d^2/\epsilon^2\right)$ sample complexity. Zanette et al. (2020b) considers a class of MDPs with low inherent Bellman error introduced by Zanette et al. (2020a). Agarwal et al. (2020) studies low-rank MDPs and proposes FLAMBE, whose learning objective can be translated to a reward-free learning goal with sample complexity $\tilde{O}\left(H^{22} d^7 A^9/\epsilon^{10}\right)$. Subsequently, Modi et al. (2021) proposes a model-free algorithm MOFFLE for low-nonnegative-rank MDPs, for which the sample complexity scales as $\tilde{O}(\frac{H^5 A^5 d_{LV}^3}{\epsilon^2 \eta})$, where $d_{LV}$ denotes the non-negative rank of the transition kernel. Recently, Chen et al. (2022) studies RF-RL with more general function approximation, but their result scales in $\tilde{O}(H^6 d^3 A/\epsilon^2)$ when specializes to low-rank MDPs, and cannot recover our upper bound.

**Safe reinforcement learning.** Safe RL is often cast in the Constrained MDP (CMDP) framework (Altman, 1999) under which the learning agent must satisfy a set of constraints (Efroni et al., 2020; Turchetta et al., 2020; Zheng & Ratliff, 2020; Qiu et al., 2020; Ding et al., 2020; Kalagarla et al., 2020; Liu et al., 2021; Wei et al., 2022; Ghosh et al., 2022). However, most of the constraints considered in the existing works require the cumulative expected cost over a horizon falling below a certain threshold, which is less stringent than the episodic-wise constraint imposed in this work. Other forms of constraints such as minimizing the variance (Tamar et al., 2012) or more generally maximizing some utility function (Ding et al., 2021), have also been investigated. Amani et al. (2021) studies safe RL with linear function approximation, where the constraint is defined using an (unknown) linear cost function of each state and action pair. In particular, Miryoosefi & Jin (2021) utilizes a reward-free oracle to solve CMDP which, however, does not have any safety guarantee for the exploration phase.

By assuming availability of a safe baseline policy, Zheng & Ratliff (2020) considers a known MDP with unknown rewards and cost functions and presents C-UCRL that achieves regret $\tilde{O}(N^{3/4})$ and zero constraint violation, where $N$ is the number of episodes. Liu et al. (2021) improves the result by proposing OptPess-LP, and achieves a regret of $\tilde{O}\left(H^2\sqrt{S^3 AN}/\kappa\right)$, where $\kappa$ is the cost value gap between the baseline policy and the constraint boundary. Safe baseline policy has been widely utilized in conservative RL as well, which is a special case of safe RL, as the constraint is defined in terms of the total expected reward being above a threshold (Garcelon et al., 2020; Yang et al., 2021).

## 7 CONCLUSION

We proposed a novel safe RF-RL framework where safety constraints are imposed during the exploration and planning phases of RF-RL. A unified algorithmic framework called SWEET was developed, which leverages an existing baseline policy to guide safe exploration. Leveraging a concave approximation error function, SWEET can achieve zero constraint violation in exploration and provably produce a near-optimal safe policy for any given reward function and safety constraint he feasible assumption in planning. We also instantiated SWEET to both tabular and low-rank MDPs, resulting in Tabular-SWEET and Low-rank-SWEET. The sample complexities of both algorithms match or outperform the state of the art in their constraint-free counterparts, proving that the safety constraint does not fundamentally impact the sample complexity of RF-RL.

---

[1]The bound is adapted from the original result by normalizing the reward function.

ACKNOWLEDGMENTS

The work of R.Huang and J. Yang was supported by the U.S. National Science Foundation under the grant CNS-2003131. The work of Y. Liang was supported in part by the U.S. National Science Foundation under the grant RINGS-2148253.

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

# Supplementary Material

## A   ANALYSIS OF THE SWEET FRAMEWORK AND THE TRUNCATED VALUE FUNCTION

In this section, we first prove two supporting lemmas in Appendix A.1, which are useful for the proof of Theorem 1. Then, we provide the proof for Theorem 1 in Appendix A.2 and the proof for an important concavity property of the truncated value function in Appendix A.3, which are essential for instantiating Theorem 1 for Tabular-SWEET and Low-rank-SWEET.

### A.1   SUPPORTING LEMMAS

We first show that the value function under an $(\epsilon_0, t)$-greedy policy deviates from that under the original policy by at most $\epsilon_0 t$.

**Lemma 1.** *Let $\pi'$ be an $(\epsilon_0, t)$-greedy version of policy $\pi$. Then, for an MDP with transition kernel $P$ and normalized utility function $u$, we must have*

$$\left| V_{P,u}^{\pi'} - V_{P,u}^{\pi} \right| \leq \epsilon_0 t.$$

*Proof.* First, we prove the statement for the case when $t = 1$.

Assume policy $\pi'$ deviates from policy $\pi$ at step $h$, and denote $\rho_h^{\pi}(s_h)$ as the marginal distribution induced by $\pi$ under the transition kernel $P$. Let $\pi^{\mathcal{U}} = (\pi_1, \cdots, \pi_{h-1}, \mathcal{U}, \pi_{h+1}, \cdots, \pi_H)$, where $\mathcal{U}$ is the uniform policy over the action space, i.e., $\mathcal{U}(a_h|s_h) = 1/|\mathcal{A}|$.

Consider the equivalent Markov policy of $\epsilon_0 \pi^{\mathcal{U}} \oplus (1 - \epsilon_0)\pi$, denoted by $\pi^{\epsilon_0}$. By Lemma 16, we have

$$\pi_h'(a_h|s_h) = \frac{\epsilon_0}{|\mathcal{A}|} + (1 - \epsilon_0)\pi(a_h|s_h)$$

$$= \frac{\epsilon_0 \rho_h^{\pi^{\mathcal{U}}}(s_h)\pi_h^{\mathcal{U}}(a_h|s_h) + (1 - \epsilon_0)\rho_h^{\pi}(s_h)\pi(a_h|s_h)}{\epsilon_0 \rho_h^{\pi^{\mathcal{U}}}(s_h) + (1 - \epsilon_0)\rho_h^{\pi}(s_h)}$$

$$= \pi_h^{\epsilon_0}(a_h|s_h),$$

where the second equality follows from the fact that $\rho_h^{\pi^{\mathcal{U}}}(s_h) = \rho_h^{\pi}(s_h)$, since the first $h - 1$ policies of $\pi$ and $\pi^{\mathcal{U}}$ are the same.

For any $h' \geq h + 1$, we have

$$\pi_{h'}^{\epsilon_0}(a_{h'}|s_{h'}) = \frac{\epsilon_0 \rho_{h'}^{\pi^{\mathcal{U}}}(s_{h'})\pi_{h'}(a_{h'}|s_{h'}) + (1 - \epsilon_0)\rho_{h'}^{\pi}(s_{h'})\pi_{h'}(a_{h'}|s_{h'})}{\epsilon_0 \rho_{h'}^{\pi^{\mathcal{U}}}(s_{h'}) + (1 - \epsilon_0)\rho_{h'}^{\pi}(s_{h'})}$$

$$= \pi_{h'}(a_{h'}|s_{h'})$$

$$= \pi_{h'}'(a_{h'}|s_{h'}),$$

where the last equality is due to the definition of $\pi'$. Therefore, $\pi' = \pi^{\epsilon_0}$, which further yields that

$$V_{P,u}^{\pi'} = V_{P,u}^{\epsilon_0 \pi^{\mathcal{U}} \oplus (1-\epsilon_0)\pi} = \epsilon_0 V_{P,u}^{\pi^{\mathcal{U}}} + (1 - \epsilon_0)V_{P,u}^{\pi}.$$

Since the value function is upper bounded by 1 with normalized utility function $u$, we immediately obtain that

$$\left| V_{P,u}^{\pi'} - V_{P,u}^{\pi} \right| \leq \epsilon_0.$$

For the general case where $\pi'$ differs from $\pi$ at steps in $\mathcal{H} \subset [H]$ with $|\mathcal{H}| = t \leq H$, consider a sequence of subsets $\{\mathcal{H}_i\}_{i=1}^t$ such that $\mathcal{H}_i \subset \mathcal{H}_{i+1}$, $|\mathcal{H}_{i+1}| - |\mathcal{H}_i| = 1$, and $\mathcal{H}_t = \mathcal{H}$. Then, we can define a sequence of policies $\{\pi^i\}_{i=1}^t$, such that $\pi^t$ is the $(\epsilon_0, i)$-greedy policy of $\pi$ that deviates from $\pi$ at steps in $\mathcal{H}_i$. Then, by the definition of $(\epsilon_0, t)$-greedy policy in Definition 3, $\pi^{i+1}$ is an $(\epsilon_0, 1)$-greedy version of policy $\pi_i$. Thus, by induction, we conclude that

$$\left| V_{P,u}^{\pi'} - V_{P,u}^{\pi} \right| \leq \sum_{i=1}^t \left| V_{P,u}^{\pi^i} - V_{P,u}^{\pi^{i-1}} \right| \leq \epsilon_0 t,$$

where we denote $\pi'$ as $\pi^t$ and $\pi$ as $\pi^0$. $\qquad\square$

The following lemma is critical for handling constraint mismatch not only between the exploration phase and the planning phase, but also between the constraint adopted for the construction of the empirical safe policy set used in exploration and the true constraint $V_{P^*,c}^\pi \le \tau$.

**Lemma 2.** *Consider a set $X$ in which the convex combination is defined through $\gamma x \oplus (1-\gamma)y \in X$, where $x, y \in X$ and $\gamma \in [0,1]$. Let $f, g : X \to [0,1]$ be two functions on $X$ such that $f$ is concave and $g$ is convex, i.e. $f(\gamma x \oplus (1-\gamma)y) \ge \gamma f(x) + (1-\gamma)f(y)$ and $g(\gamma x \oplus (1-\gamma)y) \le \gamma g(x) + (1-\gamma)g(y)$. We further assume that both $f, g$ are continuous w.r.t. $\gamma \in [0,1]$. Define an optimization problem (P) as follows:*

$$(P) \quad \max_x f(x) \ \ s.t. \ \ g(x) + f(x) \le \tilde{\tau}, \ \ \tilde{\tau} \in (0,1].$$

*Assume there exists a strictly feasible solution $x_0 \in X$ such that $g(x_0) + f(x_0) \le \tilde{\tau} - \tilde{\kappa}$ where $\tilde{\kappa} \in (0, \tilde{\tau})$, and denote $x^*$ as an optimal solution to (P).*

*If the optimal value of (P) is strictly less than $\tilde{\kappa}$, i.e. $f(x^*) < \tilde{\kappa}$, then*

$$\max_{x \in X} f(x) \le 2f(x^*)/\tilde{\kappa}.$$

*Proof.* Let $x_1 = \arg\max_{x \in X} f(x)$, and $x_\gamma = \gamma x_1 \oplus (1-\gamma)x_0 \in X$, which is the convex combination of $x_0$ and $x_1$.

If $x_1$ satisfies the constraint in (P), the result is trivial. Therefore, it suffices to consider the case when $g(x_1) + f(x_1) > \tilde{\tau}$.

We first show that $g(x_1) \ge g(x_0)$ through contradiction.

Assume $g(x_1) < g(x_0)$. Then, we have

$$\begin{aligned}
f(x_1) &> \tilde{\tau} - g(x_1) \\
&= \tilde{\tau} - \tilde{\kappa} - g(x_1) + \tilde{\kappa} \\
&\overset{(i)}{\ge} g(x_0) + f(x_0) - g(x_1) + \tilde{\kappa} \\
&\overset{(ii)}{>} \tilde{\kappa},
\end{aligned}$$

where $(i)$ follows because $x_0$ is a strictly feasible solution, and $(ii)$ follows from the assumption that $f(x_0) \in [0,1]$ and $g(x_0) > g(x_1)$. Note that $f(x_0) < \tilde{\kappa}$, and $f(x_\gamma)$ is a continuous function with respect to $\gamma \in [0,1]$. Thus, we can choose $\gamma_1 \in (0,1)$ such that

$$f(x_{\gamma_1}) = \tilde{\kappa} \in [f(x_0), f(x_1)]. \tag{7}$$

In addition, by the convexity of $g$ and the assumption that $g(x_1) < g(x_0)$, we have

$$g(x_{\gamma_1}) \le \gamma_1 g(x_1) + (1-\gamma_1)g(x_0) < g(x_0). \tag{8}$$

Combining Equations (7) and (8), we have

$$\begin{aligned}
f(x_{\gamma_1}) + g(x_{\gamma_1}) &\le g(x_0) + \tilde{\kappa} \\
&\le g(x_0) + f(x_0) + \tilde{\kappa} \\
&\le \tilde{\tau},
\end{aligned}$$

which implies that $x_{\gamma_1}$ is a feasible solution of the optimization problem (P).

Thus, by the optimality of $x^*$, we have $\tilde{\kappa} > f(x^*) \ge f(x_{\gamma_1}) = \tilde{\kappa}$, which is a contradiction. Therefore, $g(x_0) \le g(x_1)$.

Then, let $\gamma_0$ be the solution to the following equation.

$$\gamma_0 \big(g(x_1) + f(x_1)\big) + (1-\gamma_0)\big(g(x_0) + f(x_0)\big) = \tilde{\tau}. \tag{9}$$

Since $g(x_0) + f(x_0) \le \tilde{\tau} - \tilde{\kappa}$ and $f, g \in [0,1]$, we have:

$$\tilde{\tau} \le 2\gamma_0 + \tilde{\tau} - \tilde{\kappa},$$

which implies $\gamma_0 \geq \tilde{\kappa}/2$.

Since $f$ is concave and continuous w.r.t $\gamma$, there exists $\gamma^* \leq \gamma_0$ such that

$$f(x_{\gamma^*}) = \gamma_0 f(x_1) + (1 - \gamma_0)f(x_0). \tag{10}$$

On the other hand, due to the convexity of $g$, we have

$$g(x_{\gamma^*}) \leq \gamma^* g(x_1) + (1 - \gamma^*)g(x_0)$$
$$\overset{(i)}{\leq} \gamma_0 g(x_1) + (1 - \gamma_0)g(x_0), \tag{11}$$

where $(i)$ follows from the fact that $\gamma^* \leq \gamma_0$ and $g(x_1) > g(x_0)$.

Combining Equations (10) and (11), we have

$$g(x_{\gamma^*}) + f(x_{\gamma^*}) \leq \gamma_0 g(x_1) + (1 - \gamma_0)g(x_0) + \gamma_0 f(x_1) + (1 - \gamma_0)f(x_0) = \tilde{\tau},$$

which indicates that $x_{\gamma^*}$ is a feasible solution of the optimization problem (P). Thus, by the optimality of $x^*$, Equation (10), and $\gamma_0 \geq \tilde{\kappa}/2$, we conclude that

$$\max_{x \in X} f(x) = \frac{\gamma_0 f(x_1)}{\gamma_0}$$
$$\leq \frac{\gamma_0 f(x_1) + (1 - \gamma_0)f(x_0)}{\gamma_0}$$
$$= \frac{f(x_{\gamma^*})}{\gamma_0} \leq 2f(x^*)/\tilde{\kappa}.$$

$\square$

### A.2 PROOF OF THEOREM 1

We first formally restate Theorem 1 below and then provide the proof for this theorem.

**Theorem 4** (Restatement of Theorem 1). *Given an MDP $\mathcal{M}^*$ and model estimate $\hat{P}$, assume $\mathtt{U}(\hat{P}, \pi)$ is concave and continuous over the Markov policy space $\mathcal{X}$ and $\left| V_{P^*,u}^\pi - V_{\hat{P},u}^\pi \right| \leq \mathtt{U}(\hat{P}, \pi)$ for any normalized utility $u$ and policy $\pi$. Let $\epsilon_0, t$ and $\tilde{\kappa}$ be constants that satisfy $\epsilon_0 t + \tilde{\kappa} < \kappa$. Let $\mathfrak{U} = \min\left\{ \frac{\epsilon}{2}, \frac{\Delta_{\min}}{2}, \frac{\epsilon \Delta_{\min}}{5}, \frac{\tau - \epsilon_0 t}{4}, \frac{\tilde{\kappa}(\Delta(c,\tau) - \epsilon_0 t - \tilde{\kappa})}{4(\Delta(c,\tau) - \epsilon_0 t)} \right\}$, and $\mathtt{T} \leq (\Delta(c, \tau) - \epsilon_0 t)\mathfrak{U}/2$ be the termination condition of SWEET. If SWEET terminates in finite episodes, then, the following statements hold:*

*(i) The exploration phase is safe (See Equation (1)).*

*(ii) The output $\bar{\pi}$ of SWEET in the planning phase is an $\epsilon$-optimal $(c^*, \tau^*)$-safe policy (See Equation (2)).*

*Proof.* The proof consists of three steps: Step 1 shows that the exploration phase of SWEET is safe; Step 2 shows that SWEET can find an $\epsilon$-optimal policy in the planning phase for any given reward and without the constraint requirement; and Step 3 shows that SWEET can find an $\epsilon$-optimal policy in the planning phase for any given reward and under any constraint $(c^*, \tau^*)$ requirement. We next provide details for each step.

**Step 1.** This step shows that the exploration phase of SWEET is safe.

Note that the exploration policy, denoted by $\pi_b$, is an $(\epsilon_0, t)$-greedy version of the reference policy $\pi_r$, where $\pi_r$ is a solution to the following optimization problem:

$$\max_{\pi \in \mathcal{C}_{\hat{P},\mathtt{U}}(\tilde{\kappa}, \epsilon_0, t)} \mathtt{U}(\hat{P}, \pi),$$

where

$$\mathcal{C}_{\hat{P},\mathtt{U}}(\tilde{\kappa}, \epsilon_0, t) = \begin{cases} \{\pi^0\}, & \text{if } V_{\hat{P},c}^{\pi^0} + \mathtt{U}(\hat{P}, \pi^0) \geq \tau - \epsilon_0 t - \tilde{\kappa}, \\ \left\{\pi : V_{\hat{P},c}^\pi + \mathtt{U}(\hat{P}, \pi) \leq \tau - \epsilon_0 t\right\}, & \text{otherwise.} \end{cases}$$

If $\pi_r = \pi^0$, then by Lemma 1, we have

$$V_{P^*,c}^{\pi_b} \leq V_{P^*,c}^{\pi^0} + \epsilon_0 t \leq \tau - \kappa + \epsilon_0 t < \tau,$$

where the last inequality is due the condition $\epsilon_0 t + \tilde{\kappa} < \kappa$.

If $\pi_r \neq \pi^0$, then $V_{\hat{P},c}^{\pi_r} + \mathtt{U}(\hat{P}, \pi_r) \leq \tau - \epsilon_0 t$. By Lemma 1, we have

$$
\begin{aligned}
V_{P^*,c}^{\pi_b} &\leq V_{P^*,c}^{\pi_r} + \epsilon_0 t \\
&\overset{(i)}{\leq} V_{\hat{P},c}^{\pi_r} + \mathtt{U}(\hat{P}, \pi_r) + \epsilon_0 t \\
&\leq \tau - \epsilon_0 t + \epsilon_0 t = \tau,
\end{aligned}
$$

where $(i)$ follows from the definition of $\mathtt{U}(\hat{P}, \pi)$. Therefore, the exploration phase is safe.

**Step 2:** This step shows that SWEET can find an $\epsilon$-optimal policy in the planning phase for any given reward $r^*$ in the constraint-free setting ($\tau^* = \infty$), i.e., the planning phase does not have a constraint requirement.

Consider the Markov policy space $\mathcal{X}$ with the convex combination defined by the mixture policy $\gamma\pi \oplus (1 - \gamma)\pi'$, and $g(\pi) = V_{\hat{P},c}^{\pi}$.

Let $\pi_r$ be the reference policy when the termination condition is satisfied. Then, by the property of $\mathtt{U}(\hat{P}, \pi)$ and the termination condition in SWEET, the following statements hold:

- $g(\pi)$ is convex (linear) and $\mathtt{U}(\pi)$ is concave on $\mathcal{X}$. Moreover, they are both continuous.

- The baseline policy $\pi^0 \in \mathcal{X}$ and $g(\pi^0) + \mathtt{U}(\hat{P}, \pi^0) \leq \tau - \epsilon_0 t - \tilde{\kappa}$.

- $\pi_r = \arg\max_\pi \mathtt{U}(\hat{P}, \pi)$ s.t. $g(\pi) + \mathtt{U}(\hat{P}, \pi) \leq \tau - \epsilon_0 t$. Moreover, $\mathtt{U}(\hat{P}, \pi_r) \leq \mathtt{T} < \tilde{\kappa}$.

Applying Lemma 2 with the baseline policy $\pi^0$, we have

$$\max_\pi \mathtt{U}(\pi) \leq \frac{2}{\tilde{\kappa}}\mathtt{U}(\pi_r) \leq \frac{2\mathtt{T}}{\tilde{\kappa}} := x_1. \tag{12}$$

Let $\pi^{\min} = \arg\min_\pi V_{P^*,c}^{\pi}$. By the definition of $\mathtt{U}$ and Assumption 1, we have

$$
\begin{aligned}
g(\pi^{\min}) + \mathtt{U}(\pi^{\min}) &\leq V_{P^*,c}^{\pi^{\min}} + 2\mathtt{U}(\pi^{\min}) \\
&\leq V_{P^*,c}^{\pi^{\min}} + \frac{4\mathtt{T}}{\tilde{\kappa}} \\
&= \tau - \epsilon_0 t - \left(\tau - \epsilon_0 t - V_{P^*,c}^{\pi^{\min}} - \frac{4\mathtt{T}}{\tilde{\kappa}}\right). 
\end{aligned} \tag{13}
$$

We again apply Lemma 2 with the feasible solution fixed as policy $\pi^{\min}$ to conclude that

$$\max_\pi \mathtt{U}(\pi) \overset{(i)}{\leq} \frac{2\mathtt{T}}{\tau - \epsilon_0 t - V_{P^*,c}^{\pi^{\min}} - 4\mathtt{T}/\tilde{\kappa}} \overset{(ii)}{=} \frac{2\mathtt{T}}{\Delta(c,\tau) - \epsilon_0 t - 4\mathtt{T}/\tilde{\kappa}} := x_2,$$

where $(i)$ follows from Equation (13), and $(ii)$ follows from the definition of $\Delta(c, \tau)$.

Continuing this process, we get a sequence $\{x_n\}$ with recursive formula

$$x_{n+1} = 2\mathtt{T}/(\Delta(c,\tau) - \epsilon_0 t - 2x_n),$$

and

$$\max_\pi \mathtt{U}(\pi) \leq \inf\{x_n\}_{n=1}^{\infty}.$$

Denote $\Delta(c,\tau) - \epsilon_0 t$ by $\tilde{\Delta}_c$. Then, $\mathtt{T} \leq \tilde{\Delta}_c\mathfrak{U}/2 < \frac{\tilde{\kappa}(\tilde{\Delta}_c - \tilde{\kappa})}{4}$, which implies that

$$\left|\frac{\tilde{\Delta}_c}{2} - x_1\right| \leq \sqrt{\frac{\tilde{\Delta}_c^2}{4} - 4\mathtt{T}^2}.$$

Then, based on Lemma 20, $\{x_n\}$ converges to

$$\frac{\tilde{\Delta}_c - \sqrt{\tilde{\Delta}_c^2 - 16\mathtt{T}}}{4}.$$

Therefore, we conclude that

$$\max_\pi \mathtt{U}(\pi) \leq \frac{\tilde{\Delta}_c - \sqrt{\tilde{\Delta}_c^2 - 16\mathtt{T}}}{4} = \frac{\mathfrak{U}\tilde{\Delta}_c}{\tilde{\Delta}_c + \sqrt{\tilde{\Delta}_c^2 - 16\mathtt{T}}} \leq \mathfrak{U}. \tag{14}$$

Let $\tilde{\pi} = \arg\max_\pi V_{P^*,r^*}^\pi$. By the definition of $\mathtt{U}(\pi)$, we can compute the suboptimality gap of $\bar{\pi}$ as follows.

$$
\begin{aligned}
V_{P^*,r^*}^{\tilde{\pi}} - V_{P^*,r^*}^{\bar{\pi}} &= V_{P^*,r^*}^{\tilde{\pi}} - V_{\hat{P}^\epsilon,r^*}^{\tilde{\pi}} + V_{\hat{P}^\epsilon,r^*}^{\tilde{\pi}} - V_{\hat{P}^\epsilon,r^*}^{\bar{\pi}} + V_{\hat{P}^\epsilon,r^*}^{\bar{\pi}} - V_{P^*,r}^{\bar{\pi}} \\
&\stackrel{(i)}{\leq} \mathtt{U}(\tilde{\pi}) + \mathtt{U}(\bar{\pi}) \\
&\leq 2\max_\pi \mathtt{U}(\pi) \\
&\leq 2\mathfrak{U} \leq \epsilon,
\end{aligned}
$$

where $(i)$ follows from the optimality of $\bar{\pi}$, i.e. $V_{\hat{P}^\epsilon,r}^{\tilde{\pi}} \leq V_{\hat{P}^\epsilon,r}^{\bar{\pi}}$.

**Step 3:** This step shows that SWEET can find an $\epsilon$-optimal policy in the planning phase for any given reward $r^*$ and under any constraint $(c^*, \tau^*)$.

Let $g_0(\pi) = V_{\hat{P},c^*}^\pi$. Recall that

$$
\begin{aligned}
\pi^* &= \arg\max_\pi V_{P^*,r^*}^\pi \quad \text{s.t.} \quad V_{P^*,c^*}^\pi \leq \tau^*, \\
\bar{\pi} &= \arg\max_\pi V_{\hat{P}^\epsilon,r^*}^\pi \quad \text{s.t.} \quad g_0(\pi) + \mathtt{U}(\pi) \leq \tau^*.
\end{aligned}
$$

If $g_0(\pi^*) + \mathtt{U}(\pi^*) \leq \tau^*$, by the optimality of $\bar{\pi}$, we immediately have

$$V_{\hat{P}^\epsilon,r^*}^{\bar{\pi}} \geq V_{\hat{P}^\epsilon,r^*}^{\pi^*}.$$

If $g_0(\pi^*) + \mathtt{U}(\pi^*) > \tau^*$, then by the definition of $\mathtt{U}$ and Equation (14), we have

$$\tau^* < g_0(\pi^*) + \mathtt{U}(\pi^*) \leq V_{P^*,c^*}^\pi + 2\mathtt{U}(\pi^*) \leq \tau^* + 2\mathfrak{U}. \tag{15}$$

Let $\underline{\pi} = \arg\min_\pi V_{P^*,c^*}^\pi$. By Equation (14), we have

$$g_0(\underline{\pi}) + \mathtt{U}(\underline{\pi}) \leq V_{P^*,c^*}^{\underline{\pi}} + 2\mathtt{U}(\underline{\pi}) \leq V_{P^*,c^*}^{\underline{\pi}} + 2\mathfrak{U} < \tau^*. \tag{16}$$

Let $\pi^\gamma$ be the Markov policy equivalent to the mixture policy $\gamma\pi^* \oplus (1-\gamma)\underline{\pi}$ under the estimated model $\hat{P}$. Let $\Delta_{c^*} = \Delta(c^*, \tau^*) = \tau^* - V_{P^*,c^*}^{\underline{\pi}}$ and $\gamma = (\Delta_{c^*} - 3\mathfrak{U})/\Delta_{c^*}$. By the linearity of $g_0$ and Equation (14), we have

$$
\begin{aligned}
g_0(\pi^\gamma) &+ \mathtt{U}(\pi^\gamma) \\
&\leq \gamma g_0(\pi^*) + (1-\gamma)g_0(\underline{\pi}) + \mathfrak{U} \\
&\stackrel{(i)}{\leq} \gamma(\tau^* + 2\mathfrak{U}) + (1-\gamma)(V_{P^*,c^*}^{\underline{\pi}} + 2\mathfrak{U}) + \mathfrak{U} \\
&= \gamma\Delta_{c^*} + 3\mathfrak{U} + V_{P^*,c^*}^{\underline{\pi}} \\
&= \Delta_{c^*} + V_{P^*,c^*}^{\underline{\pi}} = \tau^*,
\end{aligned}
$$

where $(i)$ follows from Equations (15) and (16). This implies that $\pi^\gamma$ is a feasible solution of the optimization problem solved in the planning phase. By the optimality of $\bar{\pi}$ and the linearity of $V_{\hat{P},r^*}^\pi$, we have

$$V_{\hat{P},r^*}^{\bar{\pi}} \geq V_{\hat{P},r^*}^{\pi^\gamma}$$

$$\geq \gamma V^{\pi^*}_{\hat{P},r^*}$$

$$\overset{(i)}{=} V^{\pi^*}_{\hat{P},r^*} - \frac{3\mathfrak{U}}{\Delta_{c^*}} V^{\pi^*}_{\hat{P},r^*}$$

$$\geq V^{\pi^*}_{\hat{P},r^*} - \frac{3\mathfrak{U}}{\Delta_{\min}}, \tag{17}$$

where $(i)$ follows because $\gamma = (\Delta_{c^*} - 3\mathfrak{U})/\Delta_{c^*}$, and the last inequality follows from the normalization condition and Assumption 1.

Recall $\mathfrak{U} \leq \frac{\Delta_{\min}\epsilon}{5}$. Therefore, the suboptimality gap under $\bar{\pi}$ can be computed as follows.

$$V^{\pi^*}_{P^*,r^*} - V^{\bar{\pi}}_{P^*,r^*} \leq V^{\pi^*}_{\hat{P}^\epsilon,r^*} + \mathtt{U}(\pi^*) - V^{\bar{\pi}}_{\hat{P}^\epsilon,r^*} + \mathtt{U}(\bar{\pi})$$

$$\overset{(i)}{\leq} \frac{3\mathfrak{U}}{\Delta_{\min}} + 2\mathfrak{U}$$

$$\leq \epsilon,$$

where $(i)$ follows from Equation (17).

$\square$

## A.3   Proof of Concavity of the Truncated Value Function

In this subsection, we show that the truncated value function defined in Equation (4) is concave and continuous on the Markov policy space $\mathcal{X}$. These are crucial properties to be used for instantiating our theorem to Tabular-SWEET and Low-rank-SWEET.

**Lemma 3** (Concavity of the truncated value function). *Let $\pi^\gamma$ be the equivalent markov policy of $\gamma\pi \oplus (1-\gamma)\pi'$ under a transition model $P$. Then,*

$$\bar{V}^{\pi^\gamma}_{P,u} \geq \gamma \bar{V}^{\pi}_{P,u} + (1-\gamma)\bar{V}^{\pi'}_{P,u}.$$

*In addition, $\bar{V}^{\pi^\gamma}_{P,u}$ is continuous w.r.t. $\gamma \in [0,1]$. Moreover, if the utility function $u$ satisfies the normalization condition, then the equality holds, i.e.,*

$$\bar{V}^{\pi^\gamma}_{P,u} = \gamma \bar{V}^{\pi}_{P,u} + (1-\gamma)\bar{V}^{\pi'}_{P,u}.$$

*Proof.* Recall that the truncated value function is defined recursively as follows:

$$\begin{cases} \bar{Q}^{\pi}_{h,\hat{P}^{(n)},u}(s_h,a_h) = u(s,a) + \alpha \hat{P}^{(n)}_h \bar{V}^{\pi}_{h+1,\hat{P}^{(n)},u}(s_h,a_h) \\ \bar{V}^{\pi}_{h,\hat{P}^{(n)},u} = \min\left\{1, \underset{\pi}{\mathbb{E}}\left[\bar{Q}^{\pi}_{h,\hat{P}^{(n)},u}(s,a)\right]\right\}, \end{cases}$$

with $V^{\alpha,\pi}_{H+1,P,u}(s_{H+1}) = 0$.

The continuity of the truncated value function is straightforward, since it is a composition of $H$ continuous functions. Therefore, we focus on the concavity part in the following analysis.

By Lemma 16, the following equality holds for any utility function $u$ and time step $h$.

$$\mathbb{E}_{P,\pi^\gamma}\left[u_h(s_h,a_h)\right] = \gamma\mathbb{E}_{P,\pi}\left[u_h(s_h,a_h)\right] + (1-\gamma)\mathbb{E}_{P,\pi'}\left[u_h(s_h,a_h)\right]. \tag{18}$$

We then prove the claim by induction.

First, we note that when $h = H+1$,

$$\gamma \underset{P,\pi}{\mathbb{E}}\left[\bar{V}^{\pi}_{H+1,P,u}(s_{H+1})\right] + (1-\gamma) \underset{P,\pi'}{\mathbb{E}}\left[\bar{V}^{\pi'}_{H+1,P,u}(s_{H+1})\right]$$

$$= 0 \leq \min\left\{1, \underset{P,\pi^\gamma}{\mathbb{E}}\left[\bar{Q}^{\pi^\gamma}_{H+1,P,u}(s_{H+1},a_{H+1})\right]\right\}.$$

Assume it holds for step $h + 1$, i.e.,

$$\gamma \mathop{\mathbb{E}}_{P,\pi} \left[ \bar{V}^{\pi}_{h+1,P,u}(s_{h+1}) \right] + (1 - \gamma) \mathop{\mathbb{E}}_{P,\pi'} \left[ \bar{V}^{\pi'}_{h+1,P,u}(s_{h+1}) \right]$$

$$\leq \min \left\{ 1, \mathop{\mathbb{E}}_{P,\pi^{\gamma}} \left[ \bar{Q}^{\pi^{\gamma}}_{h+1,P,u}(s_{h+1}, a_{h+1}) \right] \right\}.$$

Then, for step $h$, by Jensen's inequality, we have

$$\gamma \mathop{\mathbb{E}}_{P,\pi} \left[ \bar{V}^{\pi}_{h,P,u}(s_h) \right] + (1 - \gamma) \mathop{\mathbb{E}}_{P,\pi'} \left[ \bar{V}^{\pi'}_{h,P,u}(s_h) \right]$$

$$\leq \min \left\{ 1, \gamma \mathop{\mathbb{E}}_{P,\pi} \left[ \bar{Q}^{\pi}_{h,P,u}(s_h, a_h) \right] + (1 - \gamma) \mathop{\mathbb{E}}_{P,\pi'} \left[ \bar{Q}^{\pi'}_{h,P,u}(s_h, a_h) \right] \right\}$$

$$= \min \left\{ 1, \gamma \mathop{\mathbb{E}}_{P,\pi} \left[ u(s_h, a_h) \right] + (1 - \gamma) \mathop{\mathbb{E}}_{P,\pi'} \left[ u(s_h, a_h) \right] \right.$$

$$\left. + \alpha\gamma \mathop{\mathbb{E}}_{P,\pi} \left[ \bar{V}^{\pi}_{h+1,P,u}(s_{h+1}) \right] + (1 - \gamma)\alpha \mathop{\mathbb{E}}_{P,\pi'} \left[ \bar{V}^{\pi'}_{h+1,P,u}(s_{h+1}) \right] \right\}$$

$$\overset{(i)}{\leq} \min \left\{ 1, \mathop{\mathbb{E}}_{P,\pi^{\gamma}} \left[ u(s_h, a_h) \right] + \alpha \mathop{\mathbb{E}}_{P,\pi^{\gamma}} \left[ \bar{Q}^{\pi^{\gamma}}_{h+1,P,u}(s_{h+1}, a_{h+1}) \right] \right\}$$

$$= \min \left\{ 1, \mathop{\mathbb{E}}_{P,\pi^{\gamma}} \left[ \bar{Q}^{\pi^{\gamma}}_{h,P,u}(s_h, a_h) \right] \right\},$$

where $(i)$ follows from Equation (18) and the induction hypothesis.

Therefore, at step $h = 1$, we have

$$\gamma \bar{V}^{\pi}_{P,u} + (1 - \gamma) \bar{V}^{\pi'}_{P,u} \leq \min \left\{ 1, \mathop{\mathbb{E}}_{\pi^{\gamma}} \left[ \bar{Q}^{\pi^{\gamma}}_{P,u}(s_1, a_1) \right] \right\} = \bar{V}^{\pi^{\gamma}}_{P,u}.$$

If $u$ satisfies the normalization condition, then $Q^{\pi}_{h,P,u}(s_h, a_h) \leq 1$ holds for any $\pi$ and $h$. By the definition of the truncated value function, we have $\bar{Q}^{\pi}_{h,P,u}(s_h, a_h) = Q^{\pi}_{h,P,u}(s_h, a_h)$ and $\bar{V}^{\pi}_{h,P,u}(s_h) = V^{\pi}_{h,P,u}(s_h)$, which implies that the truncated value function and the true value function are identical.

By Lemma 16, $\pi^{\gamma}$ introduces the same marginal probability over any state-action pair as the mixture policy $\gamma\pi \oplus (1 - \gamma)\pi'$. Therefore, when $u$ is normalized,

$$\gamma \bar{V}^{\pi}_{P,u} + (1 - \gamma) \bar{V}^{\pi'}_{P,u} = \gamma V^{\pi}_{P,u} + (1 - \gamma) V^{\pi'}_{P,u} = V^{\pi^{\gamma}}_{P,u} = \bar{V}^{\pi^{\gamma}}_{P,u},$$

which completes the proof. $\qquad\square$

## B   ANALYSIS OF TABULAR-SWEET

In this section, we first elaborate the Tabular-SWEET algorithm in Appendix B.1. To provide the analysis for this algorithm, we first provide several supporting lemmas in Appendix B.2, and then prove Theorem 2 in Appendix B.3.

### B.1   THE TABULAR-SWEET ALGORITHM

We first specify the parameters in Tabular-SWEET. The detail of Tabular-SWEET is shown in Algorithm 2.

Let $\mathfrak{U} = \min\left\{\frac{\epsilon}{2}, \frac{\Delta_{\min}}{2}, \frac{\epsilon\Delta_{\min}}{5}, \frac{\tau}{4}, \frac{\kappa}{16}\right\}$, and $\mathtt{T} = \Delta(c,\tau)\mathfrak{U}/2$ be the termination condition of Tabular-SWEET. Let the maximum iteration number $N$ be the solution of the following equation:

$$N = \frac{2^{10}e^3 30^2 \beta HSA \log(N+1)}{\Delta(c,\tau)^2 \mathfrak{U}^2} + \frac{2^{15}e^3 \beta HSA \log(N+1)}{\kappa^2}, \tag{19}$$

where $\beta = \log(2SAH/\delta) + S\log(e(1+N))$.

Recall that the estimated model is computed by

$$\hat{P}_h^{(n)}(s_{h+1}|s_h, a_h) = \begin{cases} \frac{N_h^{(n)}(s_h, a_h, s_{h+1})}{N_h^{(n)}(s_h, a_h)}, & \text{if } N_h^{(n)}(s_h, a_h) > 1, \\ \hat{P}_h^{(n)}(s_{h+1}|s_h, a_h) = \frac{1}{S}, & \text{otherwise,} \end{cases} \tag{20}$$

where $N_h^{(n)}(s_h, a_h)$ and $N_h^{(n)}(s_h, a_h, s_{h+1})$ denote the numbers of visits of $(s_h, a_h)$ and $(s_h, a_h, s_{h+1})$ up to $n$-th episode, respectively.

Then, the exploration-driven virtual reward is defined as

$$\hat{b}_h^{(n)}(s_h, a_h) = \frac{\beta_0 H}{N_h^{(n)}(s_h, a_h)}, \tag{21}$$

where $\beta_0 = 8\beta$.

The *approximation error bound* is a concave function of the truncated value function, defined as $\mathtt{U}^{(n)}(\pi) = 4\sqrt{\bar{V}_{\hat{P}^{(n)}, \hat{b}^{(n)}}^{\pi}}$.

Since $\epsilon_0 = t = 0$ and $\tilde{\kappa} = \kappa/2$, the safety set $\mathcal{C}^{(n)}$ is given by

$$\mathcal{C}_{\hat{P}, \mathtt{U}}(\tilde{\kappa}, \epsilon_0, t) = \begin{cases} \{\pi^0\}, & \text{if } V_{\hat{P}, c}^{\pi^0} + \mathtt{U}^{(n)}(\pi^0) \geq \tau - \kappa/2, \\ \left\{\pi : V_{\hat{P}, c}^{\pi} + \mathtt{U}^{(n)}(\pi) \leq \tau\right\}, & \text{otherwise.} \end{cases} \tag{22}$$

---

**Algorithm 2** Tabular-SWEET

---

1: **Input:** Baseline policy $\pi^0$, dataset $\mathcal{D} = \emptyset$, constants $\tau, \kappa, \alpha_H = \frac{H+1}{H}, \mathtt{T} = \Delta(c,\tau)\mathfrak{U}/2$.
2: // Exploration:
3: **for** $n = 1, \ldots, N$ **do**
4:     Use $\pi^{(n-1)}$ to collect $\{s_1^{(n)}, \ldots, a_H^{(n)}\}$; $\mathcal{D} \leftarrow \mathcal{D} \cup \{s_h^{(n)}, a_h^{(n)}, s_{h+1}^{(n)}\}_{h=1}^H$;
5:     Estimate $\hat{P}^{(n)}$ ; update $\hat{b}_h^{(n)}$, $\mathtt{U}^{(n)}(\pi)$ and the empirical safe policy set $\mathcal{C}^{(n)}$ (Equations (20) to (22));
6:     Solve $\pi^{(n)} = \arg\max_{\pi \in \mathcal{C}^{(n)}} \mathtt{U}^{(n)}(\pi)$;
7:     **if** $\left|\mathcal{C}^{(n)}\right| > 1$ and $\mathtt{U}^{(n)}(\pi^{(n)}) \leq \mathtt{T}$ **then**
8:         $\left(n_\epsilon, \hat{P}^\epsilon, \hat{b}^\epsilon\right) \leftarrow \left(n, \hat{P}^{(n)}, \hat{b}^{(n)}\right)$, **break;**
9:     **end if**
10: **end for**
11: // Planning:
12: Receive reward function $r^*$ and safety constraint $(c^*, \tau^*)$;
13: **Output:** $\bar{\pi} = \arg\max_\pi V_{\hat{P}^\epsilon, r^*}^\pi$    s.t.    $V_{\hat{P}^\epsilon, c^*}^\pi + 4\sqrt{\bar{V}_{\hat{P}^\epsilon, \hat{b}^\epsilon}^{\alpha_H, \pi}} \leq \tau^*$.

---

### B.2 SUPPORTING LEMMAS

First, denote $\mathrm{Var}_{P_h} V_{h+1, P', u}^\pi(s_h, a_h)$ as the variance of value function $V_{h+1, P', u}^\pi(s_{h+1})$, where $s_{h+1}$ follows the distribution $P_h(\cdot|s_h, a_h)$, i.e.,

$$\mathrm{Var}_{P_h} V_{h+1, P', u}^\pi(s_h, a_h) = \mathbb{E}_{P_h}\left[\left(V_{h+1, P', u}^\pi(s_{h+1}) - P_h V_{h+1, P', u}^\pi(s_h, a_h)\right)^2 \Big| s_h, a_h\right]. \tag{23}$$

Then, we have the following lemma.

**Lemma 4** (Lemma 3 in Ménard et al. (2021)). *Let $\rho_{*,h}^\pi(s_h, a_h)$ be the marginal probability over state-action pair $(s_h, a_h)$ induced by policy $\pi$ under the true environment $P^*$. Suppose the utility function $u$ satisfies the normalization condition. Denote*

$$\mathcal{E}_0 = \left\{ \forall n, h, s_h, a_h, D_{\mathrm{KL}}(\hat{P}_h^{(n)}(\cdot|s_h, a_h)||P_h^*(\cdot|s_h, a_h)) \leq \frac{\beta}{N_h^{(n)}(s_h, a_h)} \right\},$$

$$\mathcal{E}_1 = \left\{ \forall n, h, s_h, a_h, N_h^{(n)}(s_h, a_h) \geq \frac{1}{2} \sum_{m=0}^{n-1} \rho_{*,h}^{\pi^{(m)}}(s_h, a_h) - \beta_1 \right\},$$

$$\mathcal{E}_2 = \left\{ \forall n, h, s_h, a_h, \left| \left(\hat{P}_h^{(n)} - P_h^*\right) V_{h+1,P^*,u}^\pi(s_h, a_h) \right| \right.$$
$$\left. \leq \sqrt{\frac{2\beta_2 \mathrm{Var}_{P_h^*}\left(V_{h+1,P^*,u}\right)(s_h, a_h)}{N_h^{(n)}(s_h, a_h)}} + \frac{3\beta_2}{N_h^{(n)}(s_h, a_h)} \right\},$$

*where $\beta = \log(3SAH/\delta) + S\log(8e(1 + N))$, $\beta_1 = \log(3SAH/\delta)$, and $\beta_2 = \log(3SAH/\delta) + \log(8e(1 + N))$. Note that $\beta \geq \beta_2 \geq \beta_1$. Let $\mathcal{E} = \mathcal{E}_0 \cap \mathcal{E}_1 \cap \mathcal{E}_2$. Then, we have*

$$\mathbb{P}[\mathcal{E}] \geq 1 - \delta.$$

The following lemma shows the relationship between the visitation counters $N_h^{(n)}(s_h, a_h)$ and the pseudo-counter $\sum_{m=0}^{n-1} \rho_{*,h}^{\pi^{(m)}}(s_h, a_h)$, where $\rho_{*,h}^\pi(s_h, a_h)$ is the marginal distribution on $(s_h, a_h)$ induced by policy $\pi$ under true model $P^*$.

**Lemma 5** (Lemma 7 in Kaufmann et al. (2021), Lemma 8 in Ménard et al. (2021)). *On the event $\mathcal{E}$, we have*

$$\min\left(\frac{\beta}{N_h^{(n)}(s_h, a_h)}, 1\right) \leq \frac{4\beta}{\max\left\{\sum_{m=0}^{n-1} \rho_{*,h}^{\pi^{(m)}}(s_h, a_h), 1\right\}}.$$

In addition, we generalize Lemma 7 in Ménard et al. (2021) from deterministic policies to randomized policies. This is important for safe RL, as the optimal policy in constrained RL is possibly randomized.

**Lemma 6** (Law of total variance with randomized policy). *Given model $P$, policy $\pi$, and normalized utility function $u$, define another utility function $\sigma_h(s_h, a_h)$ as*

$$\sigma_h(s_h, a_h) = \mathrm{Var}_{P_h} V_{h+1,P,u}^\pi(s_h, a_h).$$

*Then, for any Markov policy $\pi$ and $h \in [H]$, the following bound holds:*

$$\mathbb{E}_\pi\left[\left(Q_{h,P,u}^\pi(s_h, a_h) - \sum_{h' \geq h} u(s_{h'}, a_{h'})\right)^2 \middle| s_h, a_h\right] \geq Q_{h,P,\sigma}^\pi(s_h, a_h). \tag{24}$$

*In particular, when $h = 1$, we have*

$$1 \geq \mathbb{E}_\pi\left[\left(Q_{P,u}^\pi(s_1, a_1) - \sum_{h \geq 1} u(s_h, a_h)\right)^2 \middle| s_1\right] \geq \mathbb{E}_\pi\left[Q_{P,\sigma}^\pi(s_1, a_1)|s_1\right]$$
$$= \sum_{h \geq 1} \sum_{s_h, a_h} \rho_h^\pi(s_h, a_h)\sigma_h(s_h, a_h) = \sum_{h \geq 1} \sum_{s_h, a_h} \rho_h^\pi(s_h, a_h)\mathrm{Var}_{P_h} V_{h+1,P,u}^\pi(s_h, a_h),$$

*where $\rho_h^\pi(s_h, a_h)$ is the marginal distribution over state-action pair $(s_h, a_h)$ induced by policy $\pi$ under model $P$.*

*Proof.* First, we note that the statement is trivial for $h = H + 1$ since all Q-value functions are 0.

Then, we prove the result through induction. Assume that at time step $h+1$,

$$\mathbb{E}_\pi\left[\left(Q^\pi_{h+1,P,u}(s_{h+1},a_{h+1}) - \sum_{h'\geq h+1} u(s_{h'},a_{h'})\right)^2 \middle| s_{h+1},a_{h+1}\right] \geq Q^\pi_{h+1,P,\sigma}(s_{h+1},a_{h+1}).$$

Then, at time step $h$, the LHS of Equation (24) can be computed as follows.

$$\mathbb{E}_\pi\left[\left(Q^\pi_{h,P,u}(s_h,a_h) - \sum_{h'\geq h} u(s_{h'},a_{h'})\right)^2 \middle| s_h,a_h\right]$$

$$= \mathbb{E}_\pi\left[\left(P_h V^\pi_{h+1}(s_h,a_h) - \sum_{h'\geq h+1} u(s_{h'},a_{h'}) + Q^\pi_{h+1,P,u}(s_{h+1},a_{h+1}) - Q^\pi_{h+1,P,u}(s_{h+1},a_{h+1})\right)^2 \middle| s_h,a_h\right]$$

$$= \mathbb{E}_\pi\left[\left(Q^\pi_{h+1,P,u}(s_{h+1},a_{h+1}) - \sum_{h'\geq h+1} u(s_{h'},a_{h'})\right)^2 \middle| s_h,a_h\right]$$

$$+ \mathbb{E}_\pi\left[\left(Q^\pi_{h+1,P,u}(s_{h+1},a_{h+1}) - P_h V^\pi_{h+1}(s_h,a_h)\right)^2 \middle| s_h,a_h\right]$$

$$+ 2\mathbb{E}_\pi\left[\left(Q^\pi_{h+1,P,u}(s_{h+1},a_{h+1}) - \sum_{h'\geq h+1} u(s_{h'},a_{h'})\right)\left(Q^\pi_{h+1,P,u}(s_{h+1},a_{h+1}) - P_h V^\pi_{h+1}(s_h,a_h)\right)\middle| s_h,a_h\right].$$

The term within the expectation in the third term equals 0 if we further condition it on $s_{h+1},a_{h+1}$, indicating that the third term is 0. Therefore, from the assumption, we have

$$\mathbb{E}_\pi\left[\left(Q^\pi_{h,P,u}(s_h,a_h) - \sum_{h'\geq h} u(s_{h'},a_{h'})\right)^2 \middle| s_h,a_h\right]$$

$$\geq \mathbb{E}_\pi\left[Q_{h+1,P,\sigma}(s_{h+1},a_{h+1})\middle| s_h,a_h\right]$$

$$+ \mathbb{E}_\pi\left[\mathbb{E}_{a_{h+1}\sim\pi}\left[\left(Q^\pi_{h+1,P,u}(s_{h+1},a_{h+1}) - P_h V^\pi_{h+1}(s_h,a_h)\right)^2 \middle| s_{h+1}\right]\middle| s_h,a_h\right]$$

$$\overset{(i)}{\geq} P_h V_{h+1,P,\sigma}(s_h,a_h) + \mathbb{E}_\pi\left[\left(V^\pi_{h+1,P,u}(s_{h+1}) - P_h V^\pi_{h+1}(s_h,a_h)\right)^2 \middle| s_h,a_h\right]$$

$$= \sigma_h(s_h,a_h) + P_h V_{h+1,P,\sigma}(s_h,a_h)$$

$$= Q^\pi_{h,P,\sigma}(s_h,a_h),$$

where $(i)$ follows from Jensen's inequality.

Thus, Equation (24) holds for all step $h$, and the proof is completed.

$\square$

The following lemma is the key to ensure that Tabular-SWEET satisfies the termination condition.

**Lemma 7.** *On the event $\mathcal{E}$, the summation of $\bar{V}^{\alpha_H,\pi^{(n)}}_{\hat{P}^n,\hat{b}^{(n)}}$ over any subset $\mathcal{N}\subset[N]$ scales in the order of $\log|\mathcal{N}|$, i.e.*

$$\sum_{n\in\mathcal{N}} \bar{V}^{\alpha_H,\pi^{(n)}}_{\hat{P}^{(n)},\hat{b}^{(n)}} \leq 64e^3\beta HSA\log(1+|\mathcal{N}|).$$

*Proof.* First, similar to the truncated value function, we extend the definitions of value function to incorporate the additional factor $\alpha_H$. Specifically, $\forall h \in [H]$,

$$Q_{h,P,u}^{\alpha_H,\pi} = u(s_h, a_h) + \alpha_H P_h V_{h+1,P,u}^{\alpha_H,\pi},$$

$$V_{h,P,u}^{\alpha_H,\pi} = \mathbb{E}_\pi \left[ Q_{h,P,u}^{\alpha_H,\pi} \right],$$

and $V_{H+1,P,u}^{\alpha_H,\pi} = 0$.

We then examine the difference between the truncated Q-value function defined with respect to model $\hat{P}^{(n)}$ and the Q-value function defined with respect to model $P^*$.

$$\bar{Q}_{h,\hat{P}^{(n)},\hat{b}^{(n)}}^{\alpha_H,\pi}(s_h, a_h) - Q_{h,P^*,\hat{b}^{(n)}}^{\alpha_H,\pi}(s_h, a_h)$$

$$= \alpha_H \hat{P}^{(n)} \bar{V}_{h+1,\hat{P}^{(n)},\hat{b}^{(n)}}^{\alpha_H,\pi}(s_h, a_h) - \alpha_H P_h^* V_{h,P^*,\hat{b}^{(n)}}^{\alpha_H,\pi}(s_h, a_h)$$

$$= \alpha_H \left( \hat{P}_h^{(n)} - P_h^* \right) \bar{V}_{h+1,\hat{P}^{(n)},\hat{b}^{(n)}}^{\alpha_H,\pi}(s_h, a_h) + \alpha_H P_h^* \left( \bar{V}_{h+1,\hat{P}^{(n)},\hat{b}^{(n)}}^{\alpha_H,\pi} - V_{h+1,P^*,\hat{b}^{(n)}}^{\alpha_H,\pi} \right)(s_h, a_h).$$

By Lemma 10 in Ménard et al. (2021), we bound the first term as follows.

$$\left( \hat{P}_h^{(n)} - P_h^* \right) \bar{V}_{h,\hat{P}^{(n)},\hat{b}^{(n)}}^{\alpha_H,\pi}(s_h, a_h)$$

$$\leq \min \left\{ 1, \sqrt{2\mathrm{Var}_{P_h^*} \bar{V}_{h+1,\hat{P}^{(n)},\hat{b}^{(n)}}^{\alpha_H,\pi}(s_h, a_h) \frac{\beta}{N_h^{(n)}(s_h, a_h)}} + \frac{2\beta}{3N_h^{(n)}(s_h, a_h)} \right\}$$

$$\overset{(i)}{\leq} \frac{\mathrm{Var}_{P_h^*} \bar{V}_{h+1,\hat{P}^{(n)},\hat{b}^{(n)}}^{\alpha_H,\pi}(s_h, a_h)}{H} + \min \left\{ 1, \frac{(2+H/2)\beta}{3N_h^{(n)}(s_h, a_h)} \right\}$$

$$\overset{(ii)}{\leq} \frac{P_h^* \bar{V}_{h+1,\hat{P}^{(n)},\hat{b}^{(n)}}^{\alpha_H,\pi}(s_h, a_h)}{H} + \min \left\{ \hat{b}_h^{(n)}(s_h, a_h)/8, 1 \right\},$$

where $(i)$ follows from $\sqrt{2AB} \leq A/H + BH/2$, and $(ii)$ is due to the truncated value function is at most 1 and $\mathrm{Var}(X) \leq \mathbb{E}[X]$ if $X \in [0,1]$.

Therefore, by combining the above two inequalities and taking expectation, we have,

$$\bar{V}_{h,\hat{P}^{(n)},\hat{b}^{(n)}}^{\alpha_H,\pi}(s_h) \leq \mathbb{E}_\pi \left[ \bar{Q}_{h,\hat{P}^{(n)},\hat{b}^{(n)}}^{\alpha_H,\pi}(s_h, a_h) | s_h \right]$$

$$\leq \mathbb{E}_\pi \left[ \min \left\{ \hat{b}_h^{(n)}(s_h, a_h), 1 \right\} \Big| s_h \right] + \alpha_H \mathbb{E}_\pi \left[ \min \left\{ \hat{b}_h^{(n)}(s_h, a_h)/8, 1 \right\} \Big| s_h \right]$$

$$+ \mathbb{E}_\pi \left[ (\alpha_H + \frac{\alpha_H}{H}) P_h^* \bar{V}_{h+1,\hat{P}^{(n)},\hat{b}^{(n)}}^{\alpha_H,\pi}(s_h, a_h) \Big| s_h \right]$$

$$\leq \mathbb{E}_\pi \left[ 2\min \left\{ \hat{b}_h^{(n)}(s_h, a_h), 1 \right\} + \left( 1 + \frac{3}{H} \right) P_h^* \bar{V}_{h+1,\hat{P}^{(n)},\hat{b}^{(n)}}^{\alpha_H,\pi}(s_h, a_h) \Big| s_h \right].$$

Telescoping the above inequality from $h = 1$ to $H$ and defining $b_h^{(n)}(s_h, a_h) = \min \left\{ \hat{b}_h^{(n)}(s_h, a_h), 1 \right\}$, we get

$$\bar{V}_{\hat{P}^{(n)},\hat{b}^{(n)}}^{\alpha_H,\pi} \leq V_{P^*,2b^{(n)}}^{1+3/H,\pi} \leq 2e^3 V_{P^*,\hat{b}^{(n)}}^\pi.$$

Therefore, if $\rho_{*,h}^{\pi^{(n)}}(s_h, a_h)$ is the marginal distribution over state-action pairs induced by exploration policy $\pi^{(n)}$ under the true model $P^*$, we have

$$\sum_{n \in \mathcal{N}} \bar{V}_{\hat{P}^{(n)},\hat{b}^{(n)}}^{\alpha_H,\pi^{(n)}} \leq \sum_{n \in \mathcal{N}} 2e^3 V_{P^*,b^n}^{\pi^{(n)}}$$

$$\leq 2e^3 \sum_{n\in\mathcal{N}} \sum_{h=1}^{H} \mathop{\mathbb{E}}_{P^*,\pi^{(n)}} \left[\min\left\{\frac{8H\beta}{N^{(n)}(s_h,a_h)},1\right\}\right]$$

$$= 2e^3 \sum_{n\in\mathcal{N}} \sum_{h=1}^{H} \sum_{s_h,a_h} \rho_{*,h}^{\pi^{(n)}}(s_h,a_h)\min\left\{\frac{8H\beta}{N^{(n)}(s_h,a_h)},1\right\}$$

$$\overset{(i)}{\leq} 2e^3 \sum_{h=1}^{H} \sum_{s_h,a_h} \sum_{n\in\mathcal{N}} \rho_{*,h}^{\pi^{(n)}}(s_h,a_h)\frac{8H\beta}{\max\left\{1,\sum_{m=0}^{n-1}\rho_{*,h}^{\pi_m}(s_h,a_h)\right\}}$$

$$\leq 16e^3 H\beta \sum_{h=1}^{H} \sum_{s_h,a_h} \sum_{n\in\mathcal{N}} \frac{\rho_{*,h}^{\pi^{(n)}}(s_h,a_h)}{\max\left\{1,\sum_{m\in\mathcal{N},m<n}\rho_{*,h}^{\pi_m}(s_h,a_h)\right\}}$$

$$\overset{(ii)}{\leq} 64e^3 H\beta \sum_{h=1}^{H} \sum_{s_h,a_h} \log\left(1+\sum_{n\in\mathcal{N}}\rho_{*,h}^{\pi^{(n)}}(s_h,a_h)\right)$$

$$\overset{(iii)}{\leq} 64e^3 \beta H S A \log(1+|\mathcal{N}|),$$

where $(i)$ is due to Lemma 5, $(ii)$ follows from Lemma 18, and $(iii)$ follows the fact that $\rho_{*,h}^{\pi^{(m)}}(s_h,a_h) \leq 1$. Therefore,

$$\sum_{n\in\mathcal{N}} \bar{V}_{\hat{P}^{(n)},\hat{b}^{(n)}}^{\alpha_H,\pi^{(n)}} \leq 64e^3 \beta H S A \log(1+|\mathcal{N}|).$$

$\square$

### B.3 PROOF OF THEOREM 2

**Theorem 5** (Complete version of Theorem 2). *Given $\epsilon,\delta \in (0,1)$, and safety constraint $(c,\tau)$, let $\mathfrak{U} = \min\left\{\frac{\epsilon}{2},\frac{\Delta_{\min}}{2},\frac{\epsilon\Delta_{\min}}{5},\frac{\tau}{4},\frac{\kappa}{16}\right\}$, and $\mathtt{T} = \Delta(c,\tau)\mathfrak{U}/2$ be the termination condition of Tabular-SWEET. Then, with probability at least $1-\delta$, Tabular-SWEET achieves the learning objective of safe reward-free exploration (Equations (1) and (2)), and the number of trajectories collected in the exploration phase is at most*

$$O\left(\frac{\beta H S A \iota}{\Delta(c,\tau)^2 \mathfrak{U}^2} + \frac{\beta H S A \iota}{\kappa^2}\right),$$

*where $\iota = \log\left(\frac{\beta H S A}{\Delta(c,\tau)^2 \mathfrak{U}^2} + \frac{\beta H S A}{\kappa^2}\right)$, and $\beta = \log(2SAH/\delta) + S\log(e(1+N))$.*

*Proof.* The proof of Theorem 2 mainly instantiates Theorem 1 by verifying that (a) $\mathtt{U}^{(n)}(\pi) = 4\sqrt{\bar{V}_{\hat{P}^{(n)},\hat{b}^{(n)}}^{\alpha_H,\pi}}$ is a valid approximation error bound for $V_{\hat{P}^{(n)},u}^{\pi}$, and (b) Tabular-SWEET satisfies the termination condition within $N$ episodes. The proof consists of three steps with the first two steps verifying the above two conditions and the last step characterizes the sample complexity.

**Step 1:** This step establishes the following lemma, which shows that $\mathtt{U}^{(n)}(\pi) = 4\sqrt{\bar{V}_{\hat{P}^{(n)},\hat{b}^{(n)}}^{\alpha_H,\pi}}$ is a valid approximation error bound.

**Lemma 8.** *With $\alpha_H = 1 + 1/H$ defined in Tabular-SWEET (Algorithm 2), on the event $\mathcal{E}$, for any policy $\pi$ and any utility normalized function $u$,*

$$\left|V_{\hat{P}^{(n)},u}^{\pi} - V_{P^*,u}^{\pi}\right| \leq 4\sqrt{\bar{V}_{\hat{P}^{(n)},\hat{b}^{(n)}}^{\alpha_H,\pi}}.$$

*Proof.* Recall that

$$\hat{b}_h^{(n)} = \frac{\beta_0 H}{N_h^{(n)}(s_h,a_h)},$$

where $\beta_0 = 8\beta$.

Define utility function $u^v$ as

$$u_h^v(s_h, a_h) = \sqrt{\mathrm{Var}_{\hat{P}_h^{(n)}} V_{h+1,\hat{P}^{(n)},u}^{\pi}(s_h, a_h) \min\left\{\frac{8\beta}{N_h^{(n)}(s_h, a_h)}, \frac{1}{H}\right\}}.$$

Following Step 1 of Lemma 1 in Ménard et al. (2021), we get

$$\left|V_{\hat{P}^{(n)},u}^{\pi} - V_{P^*,u}^{\pi}\right| \leq \bar{V}_{\hat{P}^{(n)},u^v}^{\alpha_H,\pi} + \bar{V}_{\hat{P}^{(n)},\hat{b}^{(n)}}^{\alpha_H,\pi}. \tag{25}$$

Next, we aim to show that

$$\bar{V}_{\hat{P}^{(n)},u^v}^{\alpha_H,\pi} \leq e\sqrt{\bar{V}_{\hat{P}^{(n)},\hat{b}^{(n)}}^{\alpha_H,\pi}}. \tag{26}$$

For that, let $\hat{\rho}^{\pi}(s_h, a_h)$ be the marginal distribution over state-action pair $(s_h, a_h)$ induced by model $\hat{P}^{(n)}$ and policy $\pi$. Note that the truncated value function is a lower bound of the corresponding value function. Thus, we can expand $\bar{V}_{\hat{P}^{(n)},u^v}^{\alpha_H,\pi}$ as follows:

$$\bar{V}_{\hat{P}^{(n)},u^v}^{\alpha_H,\pi} = \sum_{h=1}^{H} \sum_{s_h,a_h} \alpha_H^{h-1} \hat{\rho}^{\pi}(s_h, a_h) u_h^v(s_h, a_h)$$

$$\overset{(i)}{\leq} e \sum_{h=1}^{H} \sum_{s_h,a_h} \hat{\rho}^{\pi}(s_h, a_h) \sqrt{\mathrm{Var}_{\hat{P}_h^{(n)}} V_{h+1,\hat{P}^{(n)},u}^{\pi}(s_h, a_h) \min\left\{\frac{8\beta}{N_h^{(n)}(s_h, a_h)}, \frac{1}{H}\right\}}$$

$$\overset{(ii)}{\leq} e \sqrt{\sum_{h=1}^{H} \sum_{s_h,a_h} \hat{\rho}^{\pi}(s_h, a_h) \mathrm{Var}_{\hat{P}_h^{(n)}} V_{h+1,\hat{P}^{(n)},u}^{\pi}(s_h, a_h)} \sqrt{\sum_{h=1}^{H} \sum_{s_h,a_h} \hat{\rho}^{\pi}(s_h, a_h) \min\left\{\frac{8\beta}{N_h^{(n)}(s_h, a_h)}, \frac{1}{H}\right\}},$$

where $(i)$ follows from the fact that $(1 + 1/H)^H \leq e$ and $(ii)$ follows from Cauchy-Schwarz inequality.

Note that in contrast to the optimistic policy, $\pi$ could be a randomized policy in general. By Lemma 6, we have

$$\sum_{h=1}^{H} \sum_{s_h,a_h} \hat{\rho}^{\pi}(s_h, a_h) \mathrm{Var}_{\hat{P}_h^{(n)}} V_{h+1,\hat{P}^{(n)},u}^{\pi}(s_h, a_h) \leq 1.$$

Meanwhile, if we define $u_h^b(s_h, a_h) = \min\left\{\frac{8\beta}{N_h^{(n)}(s_h,a_h)}, \frac{1}{H}\right\}$, which is obviously a normalized utility function, then, we have

$$\sum_{h=1}^{H} \sum_{s_h,a_h} \hat{\rho}^{\pi}(s_h, a_h) \min\left\{\frac{8\beta}{N_h^{(n)}(s_h, a_h)}, \frac{1}{H}\right\} = \bar{V}_{\hat{P}^{(n)},u^b}^{\pi} \leq \bar{V}_{\hat{P}^{(n)},\hat{b}^{(n)}}^{\alpha_H,\pi},$$

where the last inequality follows from the facts that $u_h^b(s_h, a_h) \leq \hat{b}_h^{(n)}(s_h, a_h)$ and $\alpha_H > 1$. Thus, we have Equation (26) established.

Combining Equations (25) and (26), we conclude that

$$\left|V_{\hat{P}^{(n)},u}^{\pi} - V_{P^*,u}^{\pi}\right| \leq e\sqrt{\bar{V}_{\hat{P}^{(n)},\hat{b}^{(n)}}^{\alpha_H,\pi}} + \bar{V}_{\hat{P}^{(n)},\hat{b}^{(n)}}^{\alpha_H,\pi} \overset{(i)}{\leq} (1+e)\sqrt{\bar{V}_{\hat{P}^{(n)},\hat{b}^{(n)}}^{\alpha_H,\pi}} \leq 4\sqrt{\bar{V}_{\hat{P}^{(n)},\hat{b}^{(n)}}^{\alpha_H,\pi}},$$

where $(i)$ is due to the fact that the truncated value function is at most 1. $\qquad\square$

**Step 2:** This step establishes the following lemma, which shows that Tabular-SWEET will terminate within $N$ episodes.

**Lemma 9.** *On the event $\mathcal{E}$, there exists $n_\epsilon \in [N]$ such that $\left|\mathcal{C}^{(n_\epsilon)}\right| > 1$ and $\bar{V}_{\hat{P}^{(n_\epsilon)},\hat{b}^{(n_\epsilon)}}^{\alpha_H,\pi^{(n_\epsilon)}} \leq \mathtt{T}^2/16$, where $N$ is defined in Equation* (19)*, and $\mathtt{T}$ is defined in Tabular-SWEET (Algorithm 2).*

*Proof.* Denote $\mathcal{N}_0 = \{n \in [N] : \pi^{(n)} = \pi^0\}$. We first prove $\mathcal{N}_0$ is finite. Note that for all $n \in \mathcal{N}_0$, $V_{\hat{P}^{(n)},c}^{\pi^0} + 4\sqrt{\bar{V}_{\hat{P}^{(n)},\hat{b}^{(n)}}^{\alpha_H,\pi^0}} \geq \tau - \kappa/2$.

By Lemmas 7 and 8, we have

$$
\begin{aligned}
|\mathcal{N}_0|\kappa/2 &\leq \sum_{n \in \mathcal{N}_0} \left( V_{\hat{P}^{(n)},c}^{\pi^0} + 4\sqrt{\bar{V}_{\hat{P}^{(n)},\hat{b}^{(n)}}^{\alpha_H,\pi^0}} - V_{P^*,c}^{\pi^0} \right) \\
&\leq \sum_{n \in \mathcal{N}_0} 8\sqrt{\bar{V}_{\hat{P}^{(n)},\hat{b}^{(n)}}^{\alpha_H,\pi^0}} \\
&\leq 64\sqrt{e^3|\mathcal{N}_0|\beta HSA \log(1+N)},
\end{aligned}
$$

where the last inequality is due to Cauchy-Schwarz inequality. Therefore, $|\mathcal{N}_0| \leq \frac{2^{14}e^3\beta HSA \log(N+1)}{\kappa^2}$.

Then, we prove Lemma 9 by contradiction. Assume $\bar{V}_{\hat{P}^{(n)},\hat{b}^{(n)}}^{\pi^{(n)}} > \mathtt{T}^2/16, \forall n \in [N]\backslash\mathcal{N}_0$. According to Lemma 7, we have

$$
\begin{aligned}
(N - |\mathcal{N}_0|)\mathtt{T}^2/16 &< \sum_{n \in [N]/\mathcal{N}_0} \bar{V}_{\hat{P}^{(n)},\hat{b}^{(n)}}^{\alpha_H,\pi^{(n)}} \\
&\leq 64e^3\beta HSA \log(N - |\mathcal{N}_0| + 1),
\end{aligned}
$$

which implies that

$$
N < \frac{2^{10}e^3\beta HSA \log(N+1)}{\mathtt{T}^2} + \frac{2^{14}e^3\beta HSA \log(N+1)}{\kappa^2}.
$$

This contradicts with the condition that $N = \frac{2^{10}e^3\beta HSA \log(N+1)}{\mathtt{T}^2} + \frac{2^{14}e^3\beta HSA \log(N+1)}{\kappa^2}$. Therefore, by noting that $\mathtt{U}^{(n)}(\pi) = 4\sqrt{\bar{V}_{\hat{P}^{(n)},\hat{b}^{(n)}}^{\alpha_H,\pi^{(n)}}}$, there exists $n_\epsilon \in [N]$ such that the exploration phase under Tabular-SWEET terminates. $\qquad\square$

**Step 3:** This step analyzes the sample complexity as follows.

On the event $\mathcal{E}$, since $\mathtt{T} = \Delta(c,\tau)\mathfrak{U}/2$, by Lemma 9, the sample complexity is at most

$$
N = \frac{2^8 e^3 30^2 \beta HSA \log(N+1)}{\Delta(c,\tau)^2\mathfrak{U}^2} + \frac{2^{15}e^3\beta HSA \log(N+1)}{\kappa^2}.
$$

Note that $n = c_0 \log(c_1 n)$ implies $n \leq 2c_0 \log(c_0 c_1)$. Thus,

$$
N = O\left( \frac{\beta HSA\iota}{\Delta(c,\tau)^2\mathfrak{U}^2} + \frac{\beta HSA\iota}{\kappa^2} \right),
$$

where

$$
\iota = \log\left( \frac{\beta HSA}{\Delta(c,\tau)^2\mathfrak{U}^2} + \frac{\beta HSA}{\kappa^2} \right).
$$

Therefore, Tabular-SWEET terminates in finite episodes.

Besides, $\mathtt{U}(\pi) = 4\sqrt{\bar{V}_{\hat{P}^\epsilon,\hat{b}^\epsilon}^{\alpha_H,\pi}}$. On the event $\mathcal{E}$, by Lemma 3, Lemma 8 and the concavity of $\sqrt{x}$, $\mathtt{U}(\pi)$ is an approximation error function under $\hat{P}^\epsilon$, and is concave and continuous on $\mathcal{X}$.

We further note that $\frac{\kappa/2(\Delta(c,\tau)-\kappa/2)}{4\Delta(c,\tau)} \geq \frac{\kappa}{16}$ due to the condition $\Delta(c,\tau) \geq \kappa$, which indicates that $\mathtt{T} = \Delta(c,\tau)\mathfrak{U}/2$ satisfies the requirement in Theorem 4.

Therefore, by Theorem 4, we conclude that with probability at least $1 - \delta$, the exploration phase of Tabular-SWEET is safe and $\bar{\pi}$ is an $\epsilon$-optimal policy subject to the safety constraint $(c^*,\tau^*)$.

$\qquad\square$

# C ANALYSIS OF LOW-RANK-SWEET

In this section, we first elaborate the Low-rank-SWEET algorithm in Appendix C.1. To provide the analysis for this algorithm, we first provide several supporting lemmas in Appendix C.2, and then prove Theorem 3 in Appendix C.3.

## C.1 THE LOW-RANK-SWEET ALGORITHM

We first specify the parameters adopted in Low-rank-SWEET, which is presented in Algorithm 3.

Let $\mathfrak{U} = \min\left\{\frac{\epsilon}{2}, \frac{\Delta_{\min}}{2}, \frac{\epsilon\Delta_{\min}}{5}, \frac{\tau}{6}, \frac{\kappa}{24}\right\}$, and $\mathtt{T} = \Delta(c, \tau)\mathfrak{U}/3$ be the termination condition of Low-rank-SWEET. Recall that we set $\epsilon_0 = \kappa/6$, $t = 2$, and $\tilde{\kappa} = \kappa/3$.

We define the maximum number of iterations $N$ as

$$N = \frac{2^{10}\beta_3 H^2 d^4 A^2 \zeta^2}{\kappa^2 \mathtt{T}^2} + \frac{2^{12} \cdot 3^2 \beta_3 H^2 d^4 A^2 \zeta^2}{\kappa^4}, \tag{27}$$

where $\zeta = \log\left(2|\Phi||\Psi|NH/\delta\right)$, and $\beta_3$ is defined in Lemma 10.

Besides, we set $\tilde{A} = A/\epsilon_0$ and $\hat{\alpha} = 5\sqrt{\beta_3\zeta(\tilde{A} + d^2)}$.

For ease of exposition, we introduce the following notation for an $(\epsilon_0, t)$-greedy version of policy $\pi$, denoted as $G_{\mathcal{H}}^{\epsilon_0}\pi$, as follows:

$$G_{\mathcal{H}}^{\epsilon_0}\pi(a_h|s_h) = \begin{cases} \frac{\epsilon_0}{|\mathcal{A}|} + (1 - \epsilon_0)\pi(a_h|s_h), & \text{if } h \in \mathcal{H}, \\ \pi(a_h, a_h), & \text{if } h \notin \mathcal{H}. \end{cases} \tag{28}$$

where $|\mathcal{H}| = t$. Intuitively, $G_{\mathcal{H}}^{\epsilon_0}\pi$ follows $\pi$ at time step $h \in \mathcal{H}$ with probability $1 - \epsilon_0$ and takes uniformly action selection with the probability $\epsilon_0$.

We also define

$$\Pi_n = \text{Unif}\{\pi^{(m)}\}_{m=0}^{n-1}, \tag{29}$$

where $\text{Unif}(\mathcal{X}_0)$ is a mixture policy that uniformly chooses one policy from the policy set $\mathcal{X}_0 \subset \mathcal{X}$. We use $G_{\mathcal{H}}^{\epsilon_0}\Pi_n$ to denote the $(\epsilon_0, |\mathcal{H}|)$-greedy version of $\Pi_n$.

## C.2 SUPPORTING LEMMAS

We first characterize the following high probability event.

**Lemma 10.** *Denote*

$$f_h^{(n)}(s_h, a_h) = \left\|P_h^*(\cdot|s_h, a_h) - \hat{P}_h^{(n)}(\cdot|s_h, a_h)\right\|_1, \tag{31}$$

$$U_{h,\phi}^{(n)} = n \mathop{\mathbb{E}}_{\substack{s_h \sim (P^*, \Pi_n) \\ a_h \sim G_h^{\epsilon_0}\Pi_n}} \left[\phi(s_h, a_h)(\phi(s_h, a_h))^\top\right] + \lambda I, \tag{32}$$

*where $\lambda = \beta_3 d \log(2NH|\Phi|/\delta))$ and $\beta_3 = O(1)$.*

*Define events $\mathcal{E}_0$ and $\mathcal{E}_1$ as*

$$\mathcal{E}_0 = \left\{\forall n \in [N], h \in [H], s_h \in \mathcal{S}, a_h \in \mathcal{A}, \mathop{\mathbb{E}}_{\substack{s_h \sim (P^*, G_{h-1}^{\epsilon_0}\Pi_n) \\ a_h \sim G_h^{\epsilon_0}\Pi_n}} \left[f_h^{(n)}(s_h, a_h)^2\right] \leq \zeta/n\right\},$$

$$\mathcal{E}_1 = \left\{\forall n \in [N], h \in [H], s_h \in \mathcal{S}, a_h \in \mathcal{A}, \right.$$

$$\left. \frac{1}{5}\left\|\hat{\phi}_{h-1}^{(n)}(s, a)\right\|_{(U_{h-1,\hat{\phi}}^{(n)})^{-1}} \leq \left\|\hat{\phi}_{h-1}^{(n)}(s, a)\right\|_{(\hat{U}_{h-1}^{(n)})^{-1}} \leq 3\left\|\hat{\phi}_{h-1}^{(n)}(s, a)\right\|_{(U_{h-1,\hat{\phi}}^{(n)})^{-1}}\right\},$$

*where $\zeta = \log\left(2|\Phi||\Psi|NH/\delta\right)$.*

*Denote $\mathcal{E} := \mathcal{E}_0 \cap \mathcal{E}_1$. Then, $\mathbb{P}[\mathcal{E}] \geq 1 - \delta$.*

---

**Algorithm 3** Low-rank-SWEET

---

1: **Input:** Constants $\epsilon_0 = \kappa/6$, $\tilde{A} = A/\epsilon_0$, and termination condition T.
2: // Exploration:
3: **for** $n = 1, \ldots, N$ **do**
4:     **for** $h = 1, \ldots, H$ **do**
5:         Execute policy $G^{\epsilon_0}_{h-1,h}\pi^{(n-1)}$ and collect data $s_1^{(n,h)}, a_1^{(n,h)}, \ldots, s_H^{(n,h)}, a_H^{(n,h)}$
6:         $\mathcal{D}_h^n \leftarrow D_h^n \cup \{s_h^{(n,h)}, a_h^{(n,h)}, s_{h+1}^{(n,h)}\}$
7:     **end for**
8:     Learn $(\hat{\phi}_h^{(n)}, \hat{\mu}_h^{(n)}) = \text{MLE}(\mathcal{D}_h^n)$, and update $\hat{P}^{(n)}$ according to Equation (5)
9:     Update empirical covariance matrix $\hat{U}_h^{(n)}$ according to Equation (6)
10:    Define exploration-driven reward function $\hat{b}_h^{(n)}(\cdot, \cdot) = \min\left\{\hat{\alpha}\|\hat{\phi}_h^{(n)}(\cdot, \cdot)\|_{(\hat{U}_h^{(n)})^{-1}}, 1\right\}$.
11:    Define $\mathtt{U}^{(n)}(\pi) = \bar{V}^{\pi^{(n)}}_{\hat{P}^{(n)}, \hat{b}^{(n)}} + \sqrt{\tilde{A}\zeta/n}$

$$
\mathcal{C}_L^{(n)} = \begin{cases} \{\pi^0\}, & \text{if } V^{\pi^0}_{\hat{P}, c} + \mathtt{U}^{(n)}(\pi^0) \geq \tau - 2\kappa/3, \\ \left\{\pi : V^{\pi}_{\hat{P}, c} + \mathtt{U}^{(n)}(\pi) \leq \tau - \kappa/3\right\}, & \text{otherwise.} \end{cases} \tag{30}
$$

12:    Solve $\pi^{(n)} = \arg\max_{\pi \in \mathcal{C}_L^{(n)}} \mathtt{U}^{(n)}(\pi)$, where $\mathcal{C}_L^{(n)}$ is defined in Equation (30).
13:    **if** $\left|\mathcal{C}_L^{(n)}\right| > 1$ and $\mathtt{U}^{(n)}(\pi^{(n)}) \leq \mathtt{T}$ **then**
14:       $\left(n_\epsilon, \hat{P}^\epsilon, \hat{b}^\epsilon\right) \leftarrow \left(n, \hat{P}^{(n)}, \hat{b}^{(n)}\right)$, **break**
15:    **end if**
16: **end for**
17: // Planning:
18: Receive reward function $r^*$ and constraint $(c^*, \tau^*)$,
19: **Output:** $\bar{\pi} = \arg\max_\pi V^{\pi}_{\hat{P}^\epsilon, r^*}$    s.t.    $V^{\pi}_{\hat{P}^\epsilon, c^*} + \bar{V}^{\pi}_{\hat{P}^\epsilon, \hat{b}^\epsilon} + \sqrt{\tilde{A}\zeta_{n_\epsilon}} \leq \tau^*$.

---

*Proof.* By Corollary 2 in Appendix D, we have $\mathbb{P}[\mathcal{E}_0] \geq 1 - \delta/2$. Further, by Lemma 39 in Zanette et al. (2020a) for the version of fixed $\phi$ and Lemma 11 in Uehara et al. (2021), we have $\mathbb{P}[\mathcal{E}_1] \geq 1 - \delta/2$. Therefore, $\mathbb{P}[\mathcal{E}] \geq 1 - \delta$.    □

Based on Lemma 10, we can bound the exploration-driven virtual reward in Low-rank-SWEET as follows.

**Corollary 1.** *Given that the event $\mathcal{E}$ occurs, the following inequality holds for any $n \in [N], h \in [H], s_h \in \mathcal{S}, a_h \in \mathcal{A}$:*

$$
\min\left\{\frac{\hat{\alpha}}{5}\left\|\hat{\phi}_h^{(n)}(s_h, a_h)\right\|_{(U_{h,\hat{\phi}}^{(n)})^{-1}}, 1\right\} \leq \hat{b}_h^{(n)}(s_h, a_h) \leq 3\hat{\alpha}\left\|\hat{\phi}_h^{(n)}(s_h, a_h)\right\|_{(U_{h,\hat{\phi}}^{(n)})^{-1}},
$$

*where $\hat{\alpha} = 5\sqrt{\beta_3\zeta(\tilde{A} + d^2)}$.*

*Proof.* Recall $\hat{b}_h^{(n)}(s_h, a_h) = \min\left\{\hat{\alpha}\left\|\hat{\phi}_h^{(n)}(s, a)\right\|_{(\hat{U}_h^{(n)})^{-1}}, 1\right\}$. Applying Lemma 10, we can immediately obtain the result.    □

The following lemma summarizes Lemmas 12 and 13 in Uehara et al. (2021) and generalizes them to $\epsilon_0$-greedy policies. We provide the proof for completeness.

**Lemma 11.** *Let $P_{h-1} = \langle\phi_{h-1}, \mu_{h-1}\rangle$ be a low-rank MDP model, and $\Pi$ be an arbitrary and possibly a mixture policy. Define an expected Gram matrix as follows:*

$$
M_{h-1, \phi} = \lambda I + n\mathop{\mathbb{E}}_{\substack{s_{h-1} \sim (P^*, \Pi) \\ a_{h-1} \sim \Pi}}\left[\phi_{h-1}(s_{h-1}, a_{h-1})\left(\phi_{h-1}(s_{h-1}, a_{h-1})\right)^\top\right].
$$

*Further, let $f_{h-1}(s_{h-1}, a_{h-1})$ be the total variation distance between $P_{h-1}^*$ and $P_{h-1}$ at time step $h - 1$. Suppose $g \in \mathcal{S} \times \mathcal{A} \to \mathbb{R}$ is bounded by $B \in (0, \infty)$, i.e., $\|g\|_\infty \leq B$. Then, for $h \geq 2$, any policy $\pi_h$,*

$$\mathop{\mathbb{E}}_{\substack{s_h \sim P_{h-1} \\ a_h \sim \pi_h}} \left[ g(s_h, a_h) | s_{h-1}, a_{h-1} \right]$$

$$\leq \left\| \phi_{h-1}(s_{h-1}, a_{h-1}) \right\|_{(M_{h-1,\phi})^{-1}} \times$$

$$\sqrt{n\tilde{A} \mathop{\mathbb{E}}_{\substack{s_h \sim (P^*, \Pi) \\ a_h \sim G_h^{\epsilon_0} \Pi}} [g^2(s_h, a_h)] + \lambda dB^2 + nB^2 \mathop{\mathbb{E}}_{\substack{s_{h-1} \sim (P^*, \Pi) \\ a_{h-1} \sim \Pi}} \left[ f_{h-1}(s_{h-1}, a_{h-1})^2 \right].}$$

*Proof.* We first derive the following bound:

$$\mathop{\mathbb{E}}_{\substack{s_h \sim P_{h-1} \\ a_h \sim \pi_h}} \left[ g(s_h, a_h) | s_{h-1}, a_{h-1} \right]$$

$$= \int_{s_h} \sum_{a_h} g(s_h, a_h) \pi(a_h | s_h) \langle \phi_{h-1}(s_{h-1}, a_{h-1}), \mu_{h-1}(s_h) \rangle ds_h$$

$$\leq \left\| \phi_{h-1}(s_{h-1}, a_{h-1}) \right\|_{(M_{h-1,\phi})^{-1}} \left\| \int \sum_{a_h} g(s_h, a_h) \pi(a_h | s_h) \mu_{h-1}(s_h) ds_h \right\|_{M_{h-1,\phi}},$$

where the inequality follows from Cauchy-Schwarz inequality. We further expand the second term in the RHS of the above inequality as follows.

$$\left\| \int \sum_{a_h} g(s_h, a_h) \pi(a_h | s_h) \mu_{h-1}(s_h) ds_h \right\|^2_{M_{h-1,\phi}}$$

$$\overset{(i)}{\leq} n \mathop{\mathbb{E}}_{\substack{s_{h-1} \sim (P^*, \Pi) \\ a_{h-1} \sim \Pi}} \left[ \left( \int_{s_h} \sum_{a_h} g(s_h, a_h) \pi_h(a_h | s_h) \mu(s_h)^\top \phi(s_{h-1}, a_{h-1}) ds_h \right)^2 \right] + \lambda dB^2$$

$$= n \mathop{\mathbb{E}}_{\substack{s_{h-1} \sim (P^*, \Pi) \\ a_{h-1} \sim \Pi}} \left[ \left( \mathop{\mathbb{E}}_{\substack{s_h \sim P_{h-1} \\ a_h \sim \pi_h}} \left[ g(s_h, a_h) \Big| s_{h-1}, a_{h-1} \right] \right)^2 \right] + \lambda dB^2$$

$$\overset{(ii)}{\leq} 2n \mathop{\mathbb{E}}_{\substack{s_{h-1} \sim (P^*, \Pi) \\ a_{h-1} \sim \Pi}} \left[ \mathop{\mathbb{E}}_{\substack{s_h \sim P_{h-1}^* \\ a_h \sim \pi_h}} \left[ g(s_h, a_h)^2 \Big| s_{h-1}, a_{h-1} \right] \right] + \lambda dB^2$$

$$+ 2nB^2 \mathop{\mathbb{E}}_{\substack{s_{h-1} \sim (P^*, \Pi) \\ a_{h-1} \sim \Pi}} \left[ f_{h-1}(s_{h-1}, a_{h-1})^2 \right]$$

$$\overset{(iii)}{\leq} n\tilde{A} \mathop{\mathbb{E}}_{\substack{s_h \sim (P^*, \Pi) \\ a_h \sim G_h^{\epsilon_0} \Pi}} \left[ g(s_h, a_h)^2 \right] + \lambda dB^2 + nB^2 \mathop{\mathbb{E}}_{\substack{s_{h-1} \sim (P^*, \Pi) \\ a_{h-1} \sim \Pi}} \left[ f_{h-1}(s_{h-1}, a_{h-1})^2 \right],$$

where $(i)$ follows from the assumption that $\|g\|_\infty \leq B$, $(ii)$ follows from Jensen's inequality, and that $f_{h-1}(s_{h-1}, a_{h-1})$ is the total variation between $P_{h-1}^*$ and $P_{h-1}$ at time step $h - 1$. For $(iii)$, note that $\tilde{G}_h^{\epsilon_0} \Pi(\cdot | s_h) \geq \epsilon_0 / A = 1/\tilde{A}$, which implies that $\pi_h(\cdot | s_h) \leq 1 \leq \tilde{A} G_h^{\epsilon_0} \Pi(\cdot | s_h)$. This finishes the proof. $\square$

Based on Lemma 11, we summarize three useful inequalities in the following lemma, which bridges the total variation $f_h^{(n)}$ and the exploration-driven reward $\hat{b}_h^{(n)}$.

**Lemma 12.** *Define*

$$W_{h,\phi}^{(n)} = n \mathop{\mathbb{E}}_{\substack{s_h \sim (P^*, \Pi_n) \\ a_h \sim \Pi_n}} \left[ \phi(s_h, a_h)(\phi(s_h, a_h))^\top \right] + \lambda I, \tag{33}$$

where $\lambda = \beta_3 d \log(2NH|\Phi|/\delta)$. *Given that the event $\mathcal{E}$ occurs, the following inequalities hold for any iteration $n$: When $h \geq 2$,*

$$\underset{\substack{s_h \sim \hat{P}_{h-1}^{(n)} \\ a_h \sim \pi}}{\mathbb{E}} \left[ f_h^{(n)}(s_h, a_h) \Big| s_{h-1}, a_{h-1} \right] \leq \alpha \left\| \hat{\phi}_{h-1}^{(n)}(s_{h-1}, a_{h-1}) \right\|_{(U_{h-1,\hat{\phi}}^{(n)})^{-1}}, \tag{34}$$

$$\underset{\substack{s_h \sim P_{h-1}^{*} \\ a_h \sim \pi}}{\mathbb{E}} \left[ f_h^{(n)}(s_h, a_h) \Big| s_{h-1}, a_{h-1} \right] \leq \alpha \left\| \phi_{h-1}^{*}(s_{h-1}, a_{h-1}) \right\|_{(U_{h-1,\phi^*}^{(n)})^{-1}}, \tag{35}$$

$$\underset{\substack{s_h \sim P_{h-1}^{*} \\ a_h \sim \pi}}{\mathbb{E}} \left[ \hat{b}_h^{(n)}(s_h, a_h) \Big| s_{h-1}, a_{h-1} \right] \leq \gamma \left\| \phi_{h-1}^{*}(s_{h-1}, a_{h-1}) \right\|_{(W_{h-1,\phi^*}^{(n)})^{-1}}, \tag{36}$$

*where*

$$\alpha = \sqrt{\beta_3 \zeta(\tilde{A} + d^2)}, \quad \gamma = \sqrt{45 \beta_3 \zeta \tilde{A} d (\tilde{A} + d^2)}.$$

*When $h = 1$,*

$$\underset{a_1 \sim \pi}{\mathbb{E}} \left[ f_1^{(n)}(s_1, a_1) \right] \leq \sqrt{\tilde{A}\zeta/n}, \qquad \underset{a_1 \sim \pi}{\mathbb{E}} \left[ \hat{b}(s_1, a_1) \right] \leq 15\alpha\sqrt{\frac{d\tilde{A}}{n}}. \tag{37}$$

*Proof.* We start by showing Equation (34) as follows. Given that the event $\mathcal{E}$ occurs, we have

$$\underset{\substack{s_h \sim \hat{P}_{h-1}^{(n)} \\ a_h \sim \pi}}{\mathbb{E}} \left[ f_h^{(n)}(s_h, a_h) \Big| s_{h-1}, a_{h-1} \right]$$

$$\overset{(i)}{\leq} \left\| \hat{\phi}_{h-1}^{(n)}(s_{h-1}, a_{h-1}) \right\|_{(U_{h-1,\hat{\phi}}^{(n)})^{-1}} \times$$

$$\sqrt{n\tilde{A} \underset{\substack{s_h \sim (P^*, G_{h-1}^{\epsilon_0}\Pi_n) \\ a_h \sim G_h^{\epsilon_0}\Pi_n}}{\mathbb{E}} [f_h^{(n)}(s_h, a_h)^2] + \lambda d + n \underset{\substack{s_{h-1} \sim (P^*, G_{h-1}^{\epsilon_0}\Pi_n) \\ (a_{h-1}) \sim G_{h-1}^{\epsilon_0}\Pi_n}}{\mathbb{E}} \left[ f_{h-1}^{(n)}(s_{h-1}, a_{h-1})^2 \right]}$$

$$\overset{(ii)}{\leq} \left\| \hat{\phi}_{h-1}^{(n)}(s_{h-1}, a_{h-1}) \right\|_{(U_{h-1,\hat{\phi}}^{(n)})^{-1}} \times$$

$$\sqrt{n\tilde{A} \underset{\substack{s_h \sim (P^*, G_{h-1}^{\epsilon_0}\Pi_n) \\ a_h \sim G_h^{\epsilon_0}\Pi_n}}{\mathbb{E}} [f_h^{(n)}(s_h, a_h)^2] + \lambda d + n\tilde{A} \underset{\substack{s_{h-1} \sim (P^*, G_{h-2}^{\epsilon_0}\Pi_n) \\ (a_{h-1}) \sim G_{h-1}^{\epsilon_0}\Pi_n}}{\mathbb{E}} \left[ f_{h-1}^{(n)}(s_{h-1}, a_{h-1})^2 \right]}$$

$$\overset{(iii)}{\leq} \left\| \hat{\phi}_{h-1}^{(n)}(s_{h-1}, a_{h-1}) \right\|_{(U_{h-1,\hat{\phi}}^{(n)})^{-1}} \sqrt{2\zeta\tilde{A} + \beta_3 \zeta d^2}$$

$$\leq \alpha \left\| \hat{\phi}_{h-1}^{(n)}(s_{h-1}, a_{h-1}) \right\|_{(U_{h-1,\hat{\phi}}^{(n)})^{-1}},$$

where $(i)$ follows from Lemma 11 and the fact that $f_h^{(n)}(s_h, a_h) \leq 1$, $(ii)$ follows from importance sampling at time step $h - 2$, and $(iii)$ follows from Lemma 10.

Equation (35) follows from the arguments similar to the above.

To obtain Equation (36), we first apply Lemma 11 and obtain

$$\underset{\substack{s_h \sim P_{h-1}^{*} \\ a_h \sim \pi^{(n)}}}{\mathbb{E}} \left[ \hat{b}_h^{(n)}(s_h, a_h) \Big| s_{h-1}, a_{h-1} \right]$$

$$\leq \left\| \phi_{h-1}^{*}(s_{h-1}, a_{h-1}) \right\|_{(W_{h-1,\phi^*}^{(n)})^{-1}} \sqrt{n\tilde{A} \underset{\substack{s_h \sim (P^*, \Pi_n) \\ a_h \sim G_h^{\epsilon_0}\Pi_n}}{\mathbb{E}} [\{\hat{b}_h^{(n)}(s_h, a_h)\}^2] + \lambda d},$$

where we use the fact that $\hat{b}_h^{(n)}(s_h, a_h) \leq 1$. We further bound the term $n \mathbb{E}_{\substack{s_h \sim (P^*, \Pi_n) \\ a_h \sim G_h^{\epsilon_0} \Pi_n}} [(\hat{b}_h^{(n)}(s_h, a_h))^2]$ as follows:

$$
n \mathbb{E}_{\substack{s_h \sim (P^*, \Pi_n) \\ a_h \sim G_h^{\epsilon_0} \Pi_n}} \left[ \left( \hat{b}_h^{(n)}(s_h, a_h) \right)^2 \right]
$$

$$
\leq n \mathbb{E}_{\substack{s_h \sim (P^*, \Pi_n) \\ a_h \sim G_h^{\epsilon_0} \Pi_n}} \left[ \hat{\alpha}^2 \left\| \hat{\phi}_h^{(n)}(s_h, a_h) \right\|_{(\hat{U}_{h,\hat{\phi}}^{(n)})^{-1}}^2 \right]
$$

$$
\stackrel{(i)}{\leq} n \mathbb{E}_{\substack{s_h \sim (P^*, \Pi_n) \\ a_h \sim G_h^{\epsilon_0} \Pi_n}} \left[ 9\hat{\alpha}^2 \left\| \hat{\phi}_h^{(n)}(s_h, a_h) \right\|_{(U_{h,\hat{\phi}}^{(n)})^{-1}}^2 \right]
$$

$$
= 9\hat{\alpha}^2 \mathrm{tr} \left\{ n \mathbb{E}_{\substack{s_h \sim (P^*, \Pi_n) \\ a_h \sim G_h^{\epsilon_0} \Pi_n}} \left[ \hat{\phi}_h^{(n)}(s_h, a_h) \hat{\phi}_h^{(n)}(s_h, a_h)^\top \left( n \mathbb{E}_{\substack{s_h \sim (P^*, \Pi_n) \\ a_h \sim G_h^{\epsilon_0} \Pi_n}} \left[ \hat{\phi}_h(s_h, a_h) \hat{\phi}_h^{(n)}(s_h, a_h)^\top \right] + \lambda I \right)^{-1} \right] \right\}
$$

$$
\leq 9\hat{\alpha}^2 \mathrm{tr}(I) = 9\hat{\alpha}^2 d,
$$

where $(i)$ follows from Lemma 10, and we use $\mathrm{tr}(A)$ to denote the trace of any matrix $A$.

Thus,

$$
\mathbb{E}_{\substack{s_h \sim P_{h-1}^* \\ a_h \sim \pi}} \left[ \hat{b}_h^{(n)}(s_h, a_h) \Big| s_{h-1}, a_{h-1} \right] \leq \left\| \phi_{h-1}^*(s_{h-1}, a_{h-1}) \right\|_{(W_{h-1,\phi^*}^{(n)})^{-1}} \sqrt{9\tilde{A}\hat{\alpha}^2 d + \lambda d}
$$

$$
\leq \gamma \left\| \phi_{h-1}^*(s_{h-1}, a_{h-1}) \right\|_{W_{h-1,\phi^*}^{(n)})^{-1}}.
$$

In addition, for $h = 1$, we have

$$
\mathbb{E}_{a_1 \sim \pi^{(n)}} \left[ f_1^{(n)}(s_1, a_1) \right] \stackrel{(i)}{\leq} \sqrt{\tilde{A} \mathbb{E}_{a_1 \sim G_1^{\epsilon_0} \Pi_n} \left[ f_1^{(n)}(s_1, a_1)^2 \right]} \leq \sqrt{\tilde{A}\zeta/n},
$$

and

$$
\mathbb{E}_{a_1 \sim \pi^{(n)}} \left[ \hat{b}(s_1, a_1) \right] \stackrel{(ii)}{\leq} \hat{\alpha} \sqrt{\tilde{A} \mathbb{E}_{a_1 \sim G_1^{\epsilon_0} \Pi_n} \left[ \|\hat{\phi}_1(s_1, a_1)\|_{(\hat{U}_{1,\hat{\phi}}^{(n)})^{-1}}^2 \right]}
$$

$$
\leq 3\hat{\alpha} \sqrt{\tilde{A} \mathbb{E}_{a_1 \sim G_1^{\epsilon_0} \Pi_n} \left[ \|\hat{\phi}_1(s_1, a_1)\|_{(U_{1,\hat{\phi}}^{(n)})^{-1}}^2 \right]}
$$

$$
\leq 3 \sqrt{\frac{25\tilde{A}\alpha^2 d}{n}} = 15\alpha \sqrt{\tilde{A}\zeta/n},
$$

where both $(i)$ and $(ii)$ follow from Jensen's inequality and importance sampling. $\qquad\square$

The following lemma is key to ensure that Low-Rank-SWEET terminates in finite episodes.

**Lemma 13.** *Given that the event $\mathcal{E}$ occurs, the summation of the truncated value functions $\bar{V}_{\hat{P}^{(n)}, \hat{b}^{(n)}}^{\pi^{(n)}}$ under exploration policies $\{\pi^{(n)}\}_{n \in \mathcal{N}}$ is sublinear with respect to $|\mathcal{N}|$ for any $\mathcal{N} \subset [N]$, i.e., the following bound holds:*

$$
\sum_{n \in \mathcal{N}} \bar{V}_{\hat{P}^{(n)}, \hat{b}^{(n)}}^{\pi^{(n)}} + \sqrt{\tilde{A}\zeta/n} \leq 32\zeta H d^2 \tilde{A} \sqrt{\beta_3 |\mathcal{N}|}.
$$

*Proof.* Note that $\bar{V}^{\pi}_{h,\hat{P}^{(n)},\hat{b}^{(n)}} \leq 1$ holds for any policy $\pi$ and $h \in [H]$. We first have

$$
\bar{V}^{\pi^{(n)}}_{\hat{P}^{(n)},\hat{b}^{(n)}} - V^{\pi^{(n)}}_{P^*,\hat{b}^{(n)}} \leq \mathop{\mathbb{E}}_{\pi^{(n)}} \left[ \hat{P}^{(n)}_1 \bar{V}^{\pi^{(n)}}_{2,\hat{P}^{(n)},\hat{b}^{(n)}}(s_1,a_1) - P^*_1 V^{\pi^{(n)}}_{2,P^*,\hat{b}^{(n)}}(s_1,a_1) \right]
$$

$$
= \mathop{\mathbb{E}}_{\pi^{(n)}} \left[ \left( \hat{P}^{(n)}_1 - P^*_1 \right) \bar{V}^{\pi^{(n)}}_{2,\hat{P}^{(n)},\hat{b}^{(n)}}(s_1,a_1) + P^*_1 \left( \bar{V}^{\pi^{(n)}}_{2,\hat{P}^{(n)},\hat{b}^{(n)}} - V^{\pi^{(n)}}_{2,P^*,\hat{b}^{(n)}} \right)(s_1,a_1) \right]
$$

$$
\leq \mathop{\mathbb{E}}_{\pi^{(n)}} \left[ f^{(n)}_1(s_1,a_1) + P^*_1 \left( \bar{V}^{\pi^{(n)}}_{2,\hat{P}^{(n)},\hat{b}^{(n)}} - V^{\pi^{(n)}}_{2,P^*,\hat{b}^{(n)}} \right) \right]
$$

$$
\leq \cdots
$$

$$
\leq \mathop{\mathbb{E}}_{P^*,\pi^{(n)}} \left[ \sum_{h=1}^{H} f^{(n)}(s_h,a_h) \right] = V^{\pi^{(n)}}_{P^*,f^{(n)}},
$$

which implies $\bar{V}^{\pi^{(n)}}_{\hat{P}^{(n)},\hat{b}^{(n)}} \leq V^{\pi^{(n)}}_{P^*,\hat{b}^{(n)}} + V^{\pi^{(n)}}_{P^*,f^{(n)}}$.

Applying the Equation (36) and Equation (37), we obtain the following bound on the value function $V^{\pi_n}_{P^*,\hat{b}^{(n)}}$:

$$
V^{\pi_n}_{P^*,\hat{b}^{(n)}} = \sum_{h=1}^{H} \mathop{\mathbb{E}}_{\substack{s_h \sim (P^*,\pi^{(n)}) \\ a_h \sim \pi^{(n)}}} \left[ \hat{b}_n(s_h,a_h) \right]
$$

$$
\leq \sum_{h=2}^{H} \mathop{\mathbb{E}}_{\substack{s_{h-1} \sim (P^*,\pi^{(n)}) \\ a_{h-1} \sim \pi^{(n)}}} \left[ \gamma \left\| \phi^*_{h-1}(s_{h-1},a_{h-1}) \right\|_{(W^{(n)}_{h-1,\phi^*})^{-1}} \right] + 15\alpha \sqrt{\frac{d\tilde{A}}{n}}
$$

$$
\leq \sum_{h=1}^{H} \mathop{\mathbb{E}}_{\substack{s_h \sim (P^*,\pi^{(n)}) \\ a_h \sim \pi^{(n)}}} \left[ \gamma \left\| \phi^*_h(s_h,a_h) \right\|_{(W^{(n)}_{h,\phi^*})^{-1}} \right] + 15\alpha \sqrt{\frac{d\tilde{A}}{n}}.
$$

Similarly, we obtain

$$
V^{\pi_n}_{P^*,f^{(n)}} = \sum_{h=1}^{H} \mathop{\mathbb{E}}_{\substack{s_h \sim (P^*,\pi^{(n)}) \\ a_h \sim \pi^{(n)}}} \left[ \hat{b}_n(s_h,a_h) \right]
$$

$$
\leq \sum_{h=2}^{H} \mathop{\mathbb{E}}_{\substack{s_{h-1} \sim (P^*,\pi^{(n)}) \\ a_{h-1} \sim \pi^{(n)}}} \left[ \alpha \left\| \phi^*_{h-1}(s_{h-1},a_{h-1}) \right\|_{(U^{(n)}_{h-1,\phi^*})^{-1}} \right] + \sqrt{\frac{\zeta\tilde{A}}{n}}
$$

$$
\leq \sum_{h=1}^{H} \mathop{\mathbb{E}}_{\substack{s_h \sim (P^*,\pi^{(n)}) \\ a_h \sim \pi^{(n)}}} \left[ \alpha \left\| \phi^*_h(s_h,a_h) \right\|_{(U^{(n)}_{h,\phi^*})^{-1}} \right] + \sqrt{\frac{\zeta\tilde{A}}{n}}.
$$

Then, taking the summation of $V^{\pi_n}_{P^*,\hat{b}^{(n)}+f^{(n)}}$ over $n \in \mathcal{N}$, we have

$$
\sum_{n \in \mathcal{N}} V^{\pi_n}_{P^*,f^{(n)}+\hat{b}^{(n)}} + \sqrt{\tilde{A}\zeta/n}
$$

$$
\leq \sum_{n \in \mathcal{N}} 15\alpha \sqrt{\frac{d\tilde{A}}{n}} + 2 \sum_{n \in \mathcal{N}} \sqrt{\frac{\tilde{A}\zeta}{n}} + \sum_{n \in \mathcal{N}} \sum_{h=1}^{H} \mathop{\mathbb{E}}_{\substack{s_h \sim (P^*,\pi^{(n)}) \\ a_h \sim \pi^{(n)}}} \left[ \gamma_n \left\| \phi^*_h(s_h,a_h) \right\|_{(W^{(n)}_{h,\phi^*})^{-1}} \right]
$$

$$
+ \sum_{n \in \mathcal{N}} \sum_{h=1}^{H} \mathop{\mathbb{E}}_{\substack{s_h \sim (P^*,\pi^{(n)}) \\ a_h \sim \pi^{(n)}}} \left[ \alpha \left\| \phi^*_h(s_h,a_h) \right\|_{(U^{(n)}_{h,\phi^*})^{-1}} \right]
$$

$$\overset{(i)}{\leq} 17\alpha\sqrt{\zeta d\tilde{A}|\mathcal{N}|} + \gamma\sum_{h=1}^{H}\sqrt{|\mathcal{N}|\sum_{n\in\mathcal{N}}\underset{\substack{s_h\sim(P^*,\pi^{(n)})\\a_h\sim\pi^{(n)}}}{\mathbb{E}}\left[\left\|\phi_h^*(s_h,a_h)\right\|_{(W_{h,\phi^*}^{(n)})^{-1}}^2\right]}$$

$$+ \alpha\sum_{h=1}^{H}\sqrt{\tilde{A}|\mathcal{N}|\sum_{n\in\mathcal{N}}\underset{\substack{s_h\sim(P^*,\pi^{(n)})\\a_h\sim G_h^{\epsilon_0}\pi^{(n)}}}{\mathbb{E}}\left[\left\|\phi_h^*(s_h,a_h)\right\|_{(U_{h,\phi^*}^{(n)})^{-1}}^2\right]}$$

$$\overset{(ii)}{\leq} 17\zeta\sqrt{2\beta_3 d\tilde{A}(\tilde{A}+d^2)|\mathcal{N}|} + H\sqrt{45\beta_3\zeta d\tilde{A}(\tilde{A}+d^2)}\sqrt{d|\mathcal{N}|\zeta}$$

$$+ H\sqrt{\beta_3\zeta(\tilde{A}+d^2)}\sqrt{d\tilde{A}|\mathcal{N}|\zeta}$$

$$\leq 32\zeta Hd\sqrt{\beta_3\tilde{A}(d^2+\tilde{A})|\mathcal{N}|}$$

$$\leq 32\zeta Hd^2\tilde{A}\sqrt{\beta_3|\mathcal{N}|},$$

where $(i)$ follows from Cauchy-Schwarz inequality and importance sampling, and $(ii)$ follows from Lemma 19. Hence, the statement of Lemma 13 is verified. □

## C.3 PROOF OF THEOREM 3

**Theorem 6** (Restatement of Theorem 3). *Given $\epsilon, \delta \in (0,1)$, and safety constraint $(c,\tau)$, let $\mathfrak{U} = \min\left\{\frac{\epsilon}{2}, \frac{\Delta_{\min}}{2}, \frac{\epsilon\Delta_{\min}}{5}, \frac{\tau}{6}, \frac{\kappa}{24}\right\}$, and $\mathtt{T} = \Delta(c,\tau)\mathfrak{U}/3$ be the termination condition of Low-rank-SWEET. Then, with probability at least $1-\delta$, Low-rank-SWEET achieves the learning objective of safe reward-free exploration (Equations (1) and (2)) and the number of trajectories collected in the exploration phase is at most*

$$O\left(\frac{H^3 d^4 A^2 \iota}{\kappa^2\Delta(c,\tau)^2\mathfrak{U}^2} + \frac{H^3 d^4 A^2 \iota}{\kappa^4}\right),$$

*where $\iota = \log^2\left[\left(\frac{H^2 d^4 A^2}{\kappa^2\Delta(c,\tau)^2\mathfrak{U}^2} + \frac{H^2 d^4 A^2}{\kappa^4}\right)|\Phi||\Psi|H/\delta\right]$.*

*Proof.* The proof of Theorem 3 mainly instantiates Theorem 1 by verifying that (a) $\mathtt{U}^{(n)}(\pi) = \bar{V}_{\hat{P}^{(n)},\hat{b}^{(n)}}^\pi + \sqrt{\tilde{A}\zeta/n}$ is a valid approximation error bound for $V_{\hat{P}^{(n)},u}^\pi$, and (b) Low-rank-SWEET satisfies the termination condition within $N$ iterations. The proof consists of three steps with the first two steps verifying the above two conditions and the last step characterizes the sample complexity.

**Step 1:** This step establishes that $\mathtt{U}^{(n)}(\pi) = \bar{V}_{\hat{P}^{(n)},\hat{b}^{(n)}}^\pi + \sqrt{\tilde{A}\zeta/n}$ is a valid approximation error bound in Low-rank-SWEET.

**Lemma 14.** *For all $n \in [N]$, policy $\pi$ and the normalized utility function $u$, given that the event $\mathcal{E}$ occurs, we have*

$$\left|V_{P^*,u}^\pi - V_{\hat{P}^{(n)},u}^\pi\right| \leq \bar{V}_{\hat{P}^{(n)},\hat{b}^{(n)}}^\pi + \sqrt{\tilde{A}\zeta/n}.$$

*Proof.* We first show that $\left|V_{P^*,u}^\pi - V_{\hat{P}^{(n)},u}^\pi\right| \leq \bar{V}_{\hat{P}^{(n)},f^{(n)}}^\pi$.

Recall the definition of the truncated value functions $\bar{V}_{h,\hat{P}^{(n)},u}(s_h)$ and $\bar{Q}_{h,\hat{P}^{(n)},u}(s_h,a_h)$:

$$\bar{Q}_{h,\hat{P}^{(n)},u}^\pi(s_h,a_h) = u(s,a) + \hat{P}_h^{(n)}\bar{V}_{h+1,\hat{P}^{(n)},u}^\pi(s_h,a_h),$$

$$\bar{V}_{h,\hat{P}^{(n)},u}^\pi(s_h) = \min\left\{1, \underset{\pi}{\mathbb{E}}\left[\hat{Q}_{h,\hat{P}^{(n)},u}^\pi(s_h,a_h)\right]\right\}.$$

We develop the proof by induction. For the base case $h = H + 1$, we have $\left|V_{H+1,\hat{P}^{(n)},u}^\pi(s_{H+1}) - V_{H+1,P^*,u}^\pi(s_{H+1})\right| = 0 = \bar{V}_{H+1,\hat{P}^{(n)},\hat{b}^{(n)}}^\pi(s_{H+1})$.

Assume that $\left| V^\pi_{h+1,\hat{P}^{(n)},u}(s_{h+1}) - V^\pi_{h+1,P^*,u}(s_{h+1}) \right| \le \bar{V}^\pi_{h+1,\hat{P}^{(n)},\hat{b}^{(n)}}(s_{h+1})$ holds for any $s_{h+1}$.

Then, from Bellman equation, we have,

$$\left| Q^\pi_{h,\hat{P}^{(n)},u}(s_h,a_h) - Q^\pi_{P^*,u}(s_h,a_h) \right|$$

$$= \left| \hat{P}^{(n)}_h V^\pi_{h,\hat{P}^{(n)},u}(s_h,a_h) - P^*_h V^\pi_{h+1,P^*,u}(s_h,a_h) \right|$$

$$= \left| \hat{P}^{(n)}_h \left( V^\pi_{h+1,\hat{P}^{(n)},u} - V^\pi_{h+1,P^*,u} \right)(s_h,a_h) + \left( \hat{P}^{(n)}_h - P^*_h \right) V^\pi_{h,P^*,u}(s_h,a_h) \right|$$

$$\overset{(i)}{\le} f^{(n)}_h(s_h,a_h) + \hat{P}^{(n)}_h \left| V^\pi_{h+1,\hat{P}^{(n)},u} - V^\pi_{h+1,P^*,u} \right|(s_h,a_h)$$

$$\overset{(ii)}{\le} f^{(n)}_h(s_h,a_h) + \hat{P}^{(n)}_h \bar{V}^\pi_{h+1,\hat{P}^{(n)},f^{(n)}}(s_h,a_h)$$

$$= \bar{Q}^\pi_{h,\hat{P}^{(n)},f^{(n)}}(s_h,a_h), \tag{38}$$

where $(i)$ follows from $\|\hat{P}^{(n)}_h(\cdot|s_h,a_h) - P^*_h(\cdot|s_h,a_h)\|_1 = f^{(n)}_h(s_h,a_h)$ and the assumption that $u$ is normalized, and $(ii)$ follows from the induction hypothesis.

Then, by the definition of $\bar{V}^\pi_{h,\hat{P}^{(n)},u}(s_h)$, we have

$$\left| V^\pi_{h,\hat{P}^{(n)},u}(s_h) - V^\pi_{h,P^*,u}(s_h) \right|$$

$$= \left| \min\left\{ 1 - V^\pi_{h,P^*,u}(s_h), \underset{\pi}{\mathbb{E}}\left[ Q^\pi_{h,\hat{P}^{(n)},u}(s_h,a_h) \right] - \underset{\pi}{\mathbb{E}}\left[ Q^\pi_{h,P^*,u}(s_h,a_h) \right] \right\} \right|$$

$$\overset{(i)}{\le} \min\left\{ 1, \left| \underset{\pi}{\mathbb{E}}\left[ Q^\pi_{h,\hat{P}^{(n)},u}(s_h,a_h) - Q^\pi_{h,P^*,u}(s_h,a_h) \right] \right| \right\}$$

$$\overset{(ii)}{\le} \min\left\{ 1, \underset{\pi}{\mathbb{E}}\left[ \hat{Q}^\pi_{h,\hat{P}^{(n)},f^{(n)}}(s_h,a_h) \right] \right\}$$

$$= \bar{V}^\pi_{h,\hat{P}^{(n)},f^{(n)}}(s_h),$$

where $(i)$ follows because $\hat{Q}^\pi_{h,\hat{P}^{(n)},u}(s_h,a_h) - Q^\pi_{h,P^*,u}(s_h,a_h) > -1$, and $(ii)$ follows from Equation (38).

Therefore, by induction, we have

$$\left| V^\pi_{P^*,u} - \bar{V}^\pi_{\hat{P}^{(n)},u} \right| \le \bar{V}^\pi_{\hat{P}^{(n)},f^{(n)}}.$$

Then, we show that $\bar{V}^\pi_{\hat{P}^{(n)},f^{(n)}} \le \bar{V}^\pi_{\hat{P}^{(n)},\hat{b}^{(n)}} + \sqrt{\tilde{A}\zeta_n}$.

By Equation (34) and the fact that the total variation distance is upper bounded by 1, with probability at least $1 - \delta/2$, we have

$$\underset{\hat{P}^{(n)},\pi}{\mathbb{E}}\left[ f^{(n)}_h(s_h,a_h) \middle| s_{h-1} \right] \le \underset{\pi}{\mathbb{E}}\left[ \min\left( \alpha \left\| \hat{\phi}^{(n)}_{h-1} \right\|_{(U^{(n)}_{h-1,\hat{\phi}})^{-1}}, 1 \right) \right], \forall h \ge 2. \tag{39}$$

Similarly, when $h = 1$,

$$\underset{a_1\sim\pi}{\mathbb{E}}\left[ f^{(n)}_1(s_1,a_1) \right] \le \sqrt{\tilde{A} \underset{a\sim G^{\epsilon_0}_1 \Pi_n}{\mathbb{E}}\left[ \left( f^{(n)}_1(s_1,a_1) \right)^2 \right]} \le \sqrt{\tilde{A}\zeta_n}. \tag{40}$$

Based on Corollary 1, Equation (39) and $\alpha = 5\hat{\alpha}$, we have

$$\underset{\pi}{\mathbb{E}}\left[ \hat{b}^{(n)}_h(s_h,a_h) \middle| s_h \right] \ge \underset{\pi}{\mathbb{E}}\left[ \min\left( \alpha \left\| \hat{\phi}^{(n)}_h \right\|_{(U^{(n)}_{h,\hat{\phi}})^{-1}}, 1 \right) \right] \ge \underset{\hat{P}^{(n)},\pi}{\mathbb{E}}\left[ f^{(n)}_{h+1}(s_{h+1},a_{h+1}) \middle| s_h \right].$$

$$\tag{41}$$

For the base case $h = H$, we have

$$
\begin{aligned}
\mathop{\mathbb{E}}_{\hat{P}^{(n)},\pi} \left[ \bar{V}^\pi_{H,\hat{P}^{(n)},f^{(n)}}(s_H) \middle| s_{H-1} \right] &= \mathop{\mathbb{E}}_{\hat{P}^{(n)},\pi} \left[ f^{(n)}_H(s_H, a_H) \middle| s_{H-1} \right] \\
&\leq \mathop{\mathbb{E}}_\pi \left[ b^{(n)}_{H-1}(s_{H-1}, a_{H-1}) | s_{H-1} \right] \\
&\leq \min \left\{ 1, \mathop{\mathbb{E}}_\pi \left[ \bar{Q}^\pi_{H-1,\hat{P}^{(n)},\hat{b}^{(n)}}(s_{H-1}, a_{H-1}) \middle| s_{H-1} \right] \right\} \\
&= \bar{V}^\pi_{H-1,\hat{P}^{(n)},\hat{b}^{(n)}}(s_{H-1}).
\end{aligned}
$$

Assume that $\mathbb{E}_{\hat{P}^{(n)},\pi} \left[ \bar{V}^\pi_{h+1,\hat{P}^{(n)},f^{(n)}}(s_{h+1}) \middle| s_h \right] \leq \bar{V}^\pi_{h,\hat{P}^{(n)},\hat{b}^{(n)}}(s_h)$ holds for step $h+1$. Then, by Jensen's inequality, we obtain

$$
\mathop{\mathbb{E}}_{\hat{P}^{(n)},\pi} \left[ \bar{V}^\pi_{h,\hat{P}^{(n)},f^{(n)}}(s_h) \middle| s_{h-1} \right]
$$

$$
\leq \min \left\{ 1, \mathop{\mathbb{E}}_{\hat{P}^{(n)},\pi} \left[ f^{(n)}_h(s_h, a_h) + \hat{P}^{(n)}_h \bar{V}^\pi_{h+1,\hat{P}^{(n)},f^{(n)}}(s_h, a_h) \middle| s_{h-1} \right] \right\}
$$

$$
\overset{(i)}{\leq} \min \left\{ 1, \mathop{\mathbb{E}}_\pi \left[ \hat{b}^{(n)}_{h-1}(s_{h-1}, a_{h-1}) \right] + \mathop{\mathbb{E}}_{\hat{P}^{(n)},\pi} \left[ \mathop{\mathbb{E}}_{\hat{P}^{(n)},\pi} \left[ \bar{V}^\pi_{h+1,\hat{P}^{(n)},f^{(n)}}(s_{h+1}) \middle| s_h \right] \middle| s_{h-1} \right] \right\}
$$

$$
\overset{(ii)}{\leq} \min \left\{ 1, \mathop{\mathbb{E}}_\pi \left[ b^{(n)}_{h-1}(s_{h-1}, a_{h-1}) \right] + \mathop{\mathbb{E}}_{\hat{P}^{(n)},\pi} \left[ \bar{V}^\pi_{h,\hat{P}^{(n)},\hat{b}^{(n)}}(s_h) \middle| s_{h-1} \right] \right\}
$$

$$
= \min \left\{ 1, \mathop{\mathbb{E}}_\pi \left[ \bar{Q}^\pi_{h-1,\hat{P}^{(n)},\hat{b}^{(n)}}(s_{h-1}, a_{h-1}) \right] \right\}
$$

$$
= \bar{V}^\pi_{h-1,\hat{P}^{(n)},\hat{b}^{(n)}}(s_{h-1}),
$$

where $(i)$ follows from Equation (41), and $(ii)$ is due to the induction hypothesis.

By induction, we conclude that

$$
\begin{aligned}
\bar{V}^\pi_{\hat{P}^{(n)},f^{(n)}} &= \mathop{\mathbb{E}}_\pi \left[ f^{(s)}_1(s_1, a_1) \right] + \mathop{\mathbb{E}}_{\hat{P}^{(n)},\pi} \left[ \bar{V}^\pi_{2,\hat{P}^{(n)},f^{(n)}}(s_2) \middle| s_1 \right] \\
&\leq \sqrt{\tilde{A}\zeta/n} + \bar{V}^\pi_{\hat{P}^{(n)},\hat{b}^{(n)}}.
\end{aligned}
$$

Combining Step 1 and Step 2, we conclude that

$$
\left| V^\pi_{P^*,u} - V^\pi_{\hat{P}^{(n)},u} \right| \leq \sqrt{\tilde{A}\zeta/n} + \bar{V}^\pi_{\hat{P}^{(n)},\hat{b}^{(n)}}.
$$

$\square$

**Step 2:** This step shows that Low-rank-SWEET terminates in finite episodes.

**Lemma 15.** *On the event $\mathcal{E}$, there exists $n_\epsilon \in [N]$ such that $\bar{V}^{\pi^{n_\epsilon}}_{\hat{P}^{(n_\epsilon)},\hat{b}(n_\epsilon)} + \sqrt{\tilde{A}\zeta/n_\epsilon} \leq \mathtt{T}$, where $N$ is defined in Equation (27).*

Let $\mathcal{N}_0 = \left\{ n : \left| \mathcal{C}_L^{(n)} \right| = 1 \right\}$. We first show that $\mathcal{N}_0$ is a finite set.

Note that, $n \in \mathcal{N}_0$ implies that $\bar{V}_{\hat{P}^{(n)}, c}^{\pi^0} + \bar{V}_{\hat{P}^{(n)}, \hat{b}^{(n)}}^{\pi^0} + \sqrt{\tilde{A}\zeta/n} > \tau - 2\kappa/3$, and $\pi^{(n)} = \pi^0$. Then, we have,

$$
\begin{aligned}
|\mathcal{N}_0|\kappa/3 &< \sum_{n \in \mathcal{N}_0} \left( \bar{V}_{\hat{P}^{(n)}, c}^{\pi^0} + \bar{V}_{\hat{P}^{(n)}, \hat{b}^{(n)}}^{\pi^0} + \sqrt{\tilde{A}\zeta_n} - V_{P^*, c}^{\pi^0} \right) \\
&\overset{(i)}{\leq} \sum_{n \in \mathcal{N}_0} 2\bar{V}_{\hat{P}^{(n)}, \hat{b}^{(n)}}^{\pi^0} + 2\sqrt{\tilde{A}\zeta_n} \\
&\overset{(ii)}{\leq} 64\zeta H d^2 \tilde{A} \sqrt{\beta_3 |\mathcal{N}|},
\end{aligned}
$$

where $(i)$ is due to Lemma 14 and the $(ii)$ follows from Lemma 13. Therefore, we have

$$
|\mathcal{N}_0| \leq \frac{2^{12} \cdot 3^2 \beta_3 H^2 d^4 \tilde{A}^2 \zeta^2}{\kappa^2}.
$$

Next, we prove the existence of $n_\epsilon$ via contradiction. Assume $\bar{V}_{\hat{P}^{(n)}, \hat{b}^{(n)}}^{\pi^{(n)}} + \sqrt{\tilde{A}\zeta_n} > \text{T}, \forall n \in [N]/\mathcal{N}_0$. By Lemma 13, we have

$$
\begin{aligned}
(N - |\mathcal{N}_0|)\text{T} &< \sum_{n \in \mathcal{N}} V_{\hat{P}^{(n)}, \hat{b}^{(n)}}^{\pi^{(n)}} + \sqrt{\tilde{A}\zeta_n} \\
&\leq 32\zeta H d^2 \tilde{A} \sqrt{\beta_3 |\mathcal{N}|},
\end{aligned}
$$

which implies

$$
N < |\mathcal{N}_0| + \frac{2^{10} \beta_3 H^2 d^4 \tilde{A}^2 \zeta^2}{\text{T}^2} \leq \frac{2^{10} \beta_3 H^2 d^4 \tilde{A}^2 \zeta^2}{\text{T}^2} + \frac{2^{12} \cdot 3^2 \beta_3 H^2 d^4 \tilde{A}^2 \zeta^2}{\kappa^2}.
$$

This contradicts with the fact that $N = \frac{2^{10} \beta_3 H^2 d^4 \tilde{A}^2 \zeta^2}{\text{T}^2} + \frac{2^{12} \cdot 3^2 \beta_3 H^2 d^4 \tilde{A}^2 \zeta^2}{\kappa^2}$. $\qquad\square$

**Step 3:** This step analyzes the sample complexity of Low-rank-SWEET as follows.

Given that the event $\mathcal{E}$ occurs, since $\text{T} = \Delta(c, \tau)\mathfrak{U}/3$ and $\tilde{A} = A/\epsilon_0 = 6A/\kappa$, by Lemma 15, the number of iterations is at most

$$
N = \frac{2^{12} 3^4 \beta_3 H^2 d^4 A^2 \zeta^2}{\kappa^2 \Delta(c, \tau)^2 \mathfrak{U}^2} + \frac{2^{14} 3^4 \beta_3 H^2 d^4 A^2 \zeta^2}{\kappa^4}.
$$

Note that $n = c_0 \log^2(c_1 n)$ implies $n \leq 4c_0 \log^2(c_0 c_1)$. Thus,

$$
N = O\left( \frac{H^2 d^4 A^2 \iota}{\kappa^2 \Delta(c, \tau)^2 \mathfrak{U}^2} + \frac{H^2 d^4 A^2 \iota}{\kappa^4} \right),
$$

where

$$
\iota = \log^2 \left[ \left( \frac{H^2 d^4 A^2}{\kappa^2 \Delta(c, \tau)^2 \mathfrak{U}^2} + \frac{H^2 d^4 A^2}{\kappa^4} \right) |\Phi||\Psi| H / \delta \right].
$$

Since there are $H$ episodes in each iteration, the sample complexity is at most

$$
O\left( \frac{H^3 d^4 A^2 \iota}{\kappa^2 \Delta(c, \tau)^2 \mathfrak{U}^2} + \frac{H^3 d^4 A^2 \iota}{\kappa^4} \right).
$$

Thus, Low-rank-SWEET terminates in finite episodes.

Note that $\text{U}(\pi) = \bar{V}_{\hat{P}^\epsilon, \hat{b}^\epsilon}^\pi + \sqrt{\tilde{A}\zeta/n_\epsilon}$. Given that the event $\mathcal{E}$ occurs, by Lemma 3 and Lemma 14, $\text{U}(\pi)$ is an approximation error function under $\hat{P}^\epsilon$, and is concave and continuous on $\mathcal{X}$.

We further note that $\frac{\kappa/3(\Delta(c,\tau)-2\kappa/3)}{4(\Delta(c,\tau)-\kappa/3)} \geq \frac{\kappa}{24}$ , and $\Delta(c,\tau) - \kappa/3 \geq 2\Delta(c,\tau)/3$ due to the condition $\Delta(c,\tau) \geq \kappa$, which implies that $\mathtt{T} = \Delta(c,\tau)\mathfrak{U}/3$ satisfies the requirement in Theorem 4.

Therefore, using Theorem 4, we conclude that with probability at least $1 - \delta$, the exploration phase of Low-rank-SWEET is safe and $\bar{\pi}$ is an $\epsilon$-optimal policy subject to the constraint $(c^*, \tau^*)$.

## D    AUXILIARY LEMMAS

We first provide the following property of a mixture policy and its equivalent Markov policy for completeness.

**Lemma 16** (Theorem 6.1 in Altman (1999)). *Given a model $P$, any Markov policies $\pi, \pi' \in \mathcal{X}$, and $\gamma \in [0,1]$, there exists $\pi^\gamma \in \mathcal{X}$ that is Markov and equivalent to the mixture policy $\gamma\pi \oplus (1-\gamma)\pi'$. Let $\rho_h^\pi(s_h)$ and $\rho_h^\pi(s_h, a_h)$ be the marginal distributions over the state and the state-action pairs induced by $\pi$ under $P$, respectively. Then, the following statements hold:*

- $\rho_h^{\pi^\gamma}(s_h) = \gamma\rho_h^\pi(s_h) + (1-\gamma)\rho_h^{\pi'}(s_h)$,

- $\rho_h^{\pi^\gamma}(s_h, a_h) = \gamma\rho_h^\pi(s_h, a_h) + (1-\gamma)\rho_h^{\pi'}(s_h, a_h)$,

- $\pi^\gamma(a_h|s_h) = \rho_h^{\pi^\gamma}(s_h, a_h)/\rho_h^{\pi^\gamma}(s_h)$,

- $V_{P,u}^{\pi^\gamma} = \gamma V_{P,u}^\pi + (1-\gamma)V_{P,u}^{\pi'}$ *holds for any utility function $u$.*

Next, we present the estimation error of MLE in the $n$-th iteration at step $h$, given state $s$ and action $a$, in terms of the total variation distance, i.e. $f_h^{(n)}(s,a) = \left\|\hat{P}_h^{(n)}(\cdot|s,a) - P_h^*(\cdot|s,a)\right\|_1$. By Theorem 21 in Agarwal et al. (2020), we are able to guarantee that under all exploration policies, the estimation error can be bounded with high probability.

**Lemma 17.** (MLE guarantee). *Given $\delta \in (0,1)$, we have the following inequality holds for any $n \in [N], h \in [H]$ with probability at least $1 - \delta/2$:*

$$\sum_{m=0}^{n-1} \underset{\substack{s_h \sim \left(P^*, G_{h-1}^{\epsilon_0}\pi^{(m)}\right) \\ a_h \sim G_h^{\epsilon_0}\pi^{(m)}}}{\mathbb{E}} \left[ f_h^n(s_h, a_h)^2 \right] \leq \zeta, \quad \text{where } \zeta := \log\left(2|\Phi||\Psi|NH/\delta\right).$$

Dividing both sides of the inequality in Lemma 17 by $n$, we have the following corollary hold, which is intensively used in the analysis.

**Corollary 2.** *Given $\delta \in (0,1)$, the following inequality holds for any $n, h \geq 1$ with probability at least $1 - \delta/2$:*

$$\underset{\substack{s_h \sim \left(P^*, G_{h-1}^{\epsilon_0}\Pi_n\right) \\ a_h \sim G_h^{\epsilon_0}\Pi_n}}{\mathbb{E}} \left[ f_h^n(s_h, a_h)^2 \right] \leq \zeta/n,$$

*where $\Pi_n$ and $G_h^{\epsilon_0}\Pi_n$ are defined in Equation (29) and Equation (28), respectively.*

Then, we present two critical lemmas which ensure the summation of the approximation errors grows sublinearly in Tabular-SWEET and Low-rank-SWEET.

**Lemma 18** (Lemma 9 in Ménard et al. (2021)). *Suppose $\{a_n\}_{n=0}^\infty$ is a sequence with $a_n \in [0,1]$, $\forall n$. Let $S_n = \max\{1, \sum_{m=0}^n a_m\}$. Then, the following inequality holds:*

$$\sum_{n=1}^N \frac{a_n}{S_{n-1}} \leq 4\log(S_N + 1).$$

**Lemma 19** (Elliptical potential lemma: Lemma B.3 in He et al. (2021)). *Consider a sequence of $d \times d$ positive semidefinite matrices $X_1, \ldots, X_N$ with $\mathrm{tr}(X_n) \leq 1$ for all $n \in [N]$. Define $M_0 = \lambda_0 I$ and $M_n = M_{n-1} + X_n$. Then, $\forall \mathcal{N} \subset [N]$,*

$$\sum_{n \in \mathcal{N}} \mathrm{tr}(X_n M_{n-1}^{-1}) \leq 2d\log\left(1 + \frac{|\mathcal{N}|}{d\lambda_0}\right).$$

Finally, the following lemma is used in Theorem 4.

**Lemma 20.** *Given $a, b > 0$, define a positive sequence $\{x_n\}_{n \geq 1}$ recursively by*

$$x_{n+1} = \frac{b}{a - x_n}.$$

*If $a^2 > 4b$ and $x_1 \in [\frac{a-\sqrt{a^2-4b}}{2}, \frac{a+\sqrt{a^2-4b}}{2})$, then, $\{x_n\}$ converges to $\frac{a-\sqrt{a^2-4b}}{2}$.*

*Proof.* **Step 1.** We first show that $x_n \in [\frac{a-\sqrt{a^2-4b}}{2}, \frac{a+\sqrt{a^2-4b}}{2})$.

This is true for $n = 1$, as $x_1 \in [\frac{a-\sqrt{a^2-4b}}{2}, \frac{a+\sqrt{a^2-4b}}{2})$.

Assume that $x_{n-1} \in [\frac{a-\sqrt{a^2-4b}}{2}, \frac{a+\sqrt{a^2-4b}}{2})$. Then, with simple algebra, we can show that

$$x_n = \frac{b}{a - x_{n-1}} \in \left[ \frac{a - \sqrt{a^2 - 4b}}{2}, \frac{a + \sqrt{a^2 - 4b}}{2} \right).$$

**Step 2.** We show that $\{x_n\}$ is a non-increasing sequence.

Indeed, from Step 1, we have

$$|a - 2x_n| \leq \sqrt{a^2 - 4b}$$
$$\Rightarrow a^2 - 4ax_n + 4x_n^2 \leq a^2 - 4b$$
$$\Rightarrow ax_n - x_n^2 \geq b$$
$$\Rightarrow x_n \geq \frac{b}{a - x_n} = x_{n+1}.$$

Therefore, $x_{n+1} \leq x_n$ holds for all $n \geq 1$. Combining Steps 1 and 2, we conclude that there exists a limit of the sequence $\{x_n\}$, denoted by $x^*$.

By the recursive formula, $x^*$ must be a solution to the following equation

$$x^* = \frac{b}{a - x^*}.$$

Since $x^* \leq x_1 < \frac{a+\sqrt{a^2-4b}}{2}$, by solving the above equation, we conclude that

$$\lim_{n \to \infty} x_n = x^* = \frac{a - \sqrt{a^2 - 4b}}{2}.$$

$\square$

