# OpenReview forum: "Safe Exploration Incurs Nearly No Additional Sample Complexity for Reward-Free RL"
_ICLR.cc/2023/Conference — ICLR 2023 poster_

### Official Review · Reviewer_gkGM · 2022-10-23

**Confidence:** 3
**Correctness:** 4
**Technical Novelty And Significance:** 4
**Empirical Novelty And Significance:** 3
**Recommendation:** 8

**Clarity, Quality, Novelty And Reproducibility:**

To the best of my knowledge, the paper is novel and the provided results are of great value. However, some aspects of the framework could have been explained with more details (see comments above). Since I did not carefully checked the analysis, it is hard for me to judge its reproducibility.

**Strength And Weaknesses:**

*Strengths*
- (Relevance and value of the contribution) The main result of this paper, i.e., that reward-free RL under safe constraints is not statistically harder than the unconstrained version is really interesting;
- (Originality of the method) The proposed method is based on several original components, and, to the best of my knowledge, is not an incremental extension of previous approaches;
- (Thorough analysis) The paper provides sample complexity results that match the best-known rates for unconstrained reward-free RL in both tabular MDPs and low-rank MDPs.

*Weaknesses*
- (Computational complexity) The paper does not discuss the computational tractability of the Algorithm 1;
- (Technical novelty) The technical novelty of the analysis is not clearly discussed in the main paper, although the originality of the algorithm suggests that uncommon techniques have been used;
- (Strength of the assumptions) The paper assumes to have access to a safe baseline policy and the presence of a positive safety margin.

*Questions*

(Complexity) Can the authors discuss the computational complexity of Algorithm 1? Especially, the optimization problem in line 6 can be solved efficiently?

(Technical novelty) Can the authors describe the main technical innovations of their analysis?

(Assumptions) Although the assumptions of knowing a safe baseline policy and the positive safety gap are mostly reasonable, it would be great to have a discussion on whether they can be overcome and to what extent. E.g., do the authors believe that the baseline policy assumption can be avoided through a weaker notion of zero constraint violations (such as constraints violations decrease at a fast rate)?

**Summary Of The Paper:**

This paper addresses the problem of reward-free reinforcement learning under (possibly mismatching) cost constraints in both the exploration phase and the planning phase. Crucially, the paper shows how to exploit the knowledge of a safe baseline policy to achieve zero constraints violations while matching the sample complexity of unconstrained methods in both tabular and low-rank MDPs. The proposed method, called SWEET, is based on three novel components: A notion of (eps,t)-greedy policy that allows for some exploration over actions without deviating too much from a reference policy, an approximate error function that accounts for the uncertainty on the estimated model, and an empirical safe policy set built upon the approximation error function.

**Summary Of The Review:**

This looks like a great work to me. Especially, it stands out for the relevance of the main take, i.e., that reward-free RL exploration under safety constraints is not harder than unconstrained reward-free RL (at least when we have access to a baseline policy), for the originality of the methodology and (supposedly) the analysis, and for the completeness of the results, which include both tabular and low-rank MDPs. Thus, I am currently providing a fully positive evaluation. However, I could not check the analysis with such a short reviewing window, and I reserve to update my evaluation when I will have the chance to look into the technical details.

---

> ### Author Response · Authors · 2022-11-16
> **Response to Reviewer gkGM**
>
> We thank the reviewer for the careful reading of our work and thoughtful comments. We provide
> our responses in detail below.
>
> **Q1**: Can the authors discuss the computational efficiency of the algorithm? Especially, can line 6 in the algorithm be solved efficiently?
>
> **A1**: Thanks for the question. Due to the special structure of the problem, in general, we can use policy gradient based approach to solve line 6 efficiently. Below we sketch the main steps. First, we can parameterize policies by $\theta$ and express the objective function and the constraint as functions of $\theta$. Then, noting that the truncated value function is a composition of non-linear operator $\min$ and a linear function of the policy at each step $h$, all we need is to derive the form of policy gradient of those functions. Finally, we can apply standard constrained optimization approaches, such as primal-dual method or switching gradient type of primal method, to solve line 6.
>
>
> **Q2**: Can the authors describe the main technical innovations of their analysis?
>
> **A2**: Our first technical innovation in the analysis is due to the following challenge that our algorithm needs to deal with. Our setting requires safe exploration for a given constraint in the exploration phase, so that only a small fixed set of policies (that satisfy the constraint) can be explored. But the planning phase requires to find a nearly-optimal policy for **arbitrary reward and constraint** without querying any new samples. Such a policy may not be well explored in the exploration phase. Thus, our algorithm introduces a novel approximation error function, and we show that it is concave, which turns out to be critical. We further show that, with such concavity (together with continuity), by controlling the approximation error function within a small policy set, the maximum value of the approximation error function over the entire policy set is also bounded. More details can be found in page 6. This property enables us to upper bound the approximation error function on **any** policy by only controlling the approximation error function within a small policy set, and hence enables the agent to do planning over any arbitrarily given reward and constraints.
>
> We further note that our design is also different from the reward-free RL literature. Here, we employ a clipping strategy on **V-function** rather than **Q-function** (which is conventionally adopted in the reward-free literature) to ensure that the resulting exploration driven function is **concave**. Such a novel clipping technique also plays a critical role in reducing the sample complexity for low-rank MDPs compared with all existing constraint-free algorithms.
>
> Second, for the tabular case, since the exploration policy is random in general due to constraints (as opposed to deterministic policy in constraint-free case), we generalize the law of total variance (See Lemma 6) to stochastic policy.  This is crucial since the tight sample complexity of tabular MDP relies on the analysis of the variance of the value functions.
>
> Finally, for the low-rank case, due to the clipping technique used in truncated value function, the analysis for low-rank MDP are also different from the existing literature. In Lemma 14, we show that this truncated value function is indeed a valid approximation error function of the estimated model. Then, in order to show the termination condition will be satisfied within finite samples, we develop Lemma 13, which proves that the summation of approximation error function is sublinear.
>
>
> **Q3**: Although the assumptions of knowing a safe baseline policy and the positive safety gap are mostly reasonable, it would be great to have a discussion on whether they can be overcome and to what extent. E.g., do the authors believe that the baseline policy assumption can be avoided through a weaker notion of zero constraint violations (such as constraints violations decrease at a fast rate)?
>
> **A3**: Thanks for the comment. We believe that the zero constraint violations can be relaxed to bounded constraint violations so that we can remove the assumption on the prior knowledge of baseline policy. A promising approach is to adopt the CRPO algorithm proposed in Xu et al. (2021). Specifically, in the exploration phase, other than implementing line 6, the agent can first minimize the constraint function to be below the threshold if the initialization does not satisfy the constraint (which is necessary anyway). Then the algorithm performs value function maximization as long as the constraint is satisfied, and immediately performs constraint minimization once it is violated. In this way, the constraint violation will mainly occur in the initial phase (due to the lack of a safe baseline policy information).
>
> Tengyu Xu, Yingbin Liang, Guanghui Lan, “CRPO: A new approach for safe reinforcement learning with convergence guarantee”, Proc. International Conference on Machine Learning (ICML), 2021.

---

> > ### Comment · Reviewer_gkGM · 2022-11-20
> > **After Response**
> >
> > I want to thank the authors for their replies. Having read others' reviews and subsequent response, I am still positive about this work, and I am planning to keep my original score. One clear direction to strengthen this work is to clarify whether the assumptions considered by this paper, e.g., knowledge of $\tau$, are necessary for statistical tractability, or they can perhaps be circumvent with more careful algorithmic choices.

---

> > > ### Author Response · Authors · 2022-11-21
> > > **Many thanks for the further feedback**
> > >
> > > We thank the reviewer very much for the further positive feedback and for pointing out a clear future direction. We also think this is an interesting topic and will definitely look into this with full effort.

---

> > > ### Author Response · Authors · 2022-11-23
> > > **A Follow-up Response**
> > >
> > > Regarding the future direction mentioned by the reviewer to clarify whether the knowledge of safety margine $\kappa$ is necessary, we have some new thoughts to share with the reviewer.
> > >
> > > we can provide the following strategy to estimate a lower confidence bound (LCB) of $\kappa$ such that (1) the estimation of $\kappa$ does not cause order-wise increase of sample complexity; and (2) using such an estimated lower bound to serve the role of $\kappa$ will not cause order-wise change of the sample complexity. In this way, our algorithm won't need the knowledge of the safety margin $\kappa$. We briefly describe such a strategy below.
> > >
> > > At each episode $k$, if the agent executes the baseline policy, then let $C_k$ denote the random total cost of the baseline policy over the entire episode, which takes value witin $[0,1]$. Note that $\mathbb{E}[C_k] = V_{P^*,c}^{\pi^b} = \tau -\kappa$. Then, to obtain an LCB of $\kappa$, the agent will continuously execute the baseline policy to take sample trajectories and collect the corresponding $C_k$. At the end of each episode $K$, the agent checks whether the condition $\tau-\frac{\sum_{k=1}^K C_k}{K}>2\sqrt{\frac{\log 4/\delta}{K}}$ is satisfied. (Since the LHS of the inequality will converge to $\kappa$ by concentration, and the RHS will converge to zero, the condition will be satisfied at a certain $K$). If the condition is satisfied at certain $K_0$, then the agent terminates this procedure and use $\sqrt{\frac{\log 4/\delta}{K
> > > _0}}$ to serve as the LCB of $\kappa$.
> > >
> > > We next provide a brief analysis as follows. Suppose the above estimation strategy terminates at episode $K_0$. By concentration inequality, we can show that with probability at least $1-\delta$,
> > >
> > > $\frac{\sum_{k=1}^{K_0}C_k}{K_0}\geq \tau-\kappa - \sqrt{\frac{\log 4/\delta}{K_0}}$, and $\frac{\sum_{k=1}^{K_0-1}C_k}{K_0-1}\leq  \tau-\kappa + \sqrt{\frac{\log 4/\delta}{K_0-1}}.$
> > >
> > > On the other hand, by the termination condition, we know that
> > >
> > > $\frac{\sum_{k=1}^{K_0}C_k}{K_0}< \tau - 2\sqrt{\frac{\log 4/\delta}{K_0}}$, and $\frac{\sum_{k=1}^{K_0-1}C_k}{K_0-1}\geq \tau - 2\sqrt{\frac{\log 4/\delta}{K_0-1}}$.
> > >
> > > Combining these inequalities, we have $\sqrt{\frac{\log4/\delta}{K_0}}<\kappa\leq 3\sqrt{\frac{\log4/\delta}{K_0-1}}$. Therefore, the sample complexity (i.e., the total episodes we collect) for estimating such a lower bound is at most $O((\log4/\delta)/\kappa^2)$. Further, $\sqrt{\frac{\log4/\delta}{K_0}}$ is indeed an LCB of $\kappa$. Since $K_0 = \Theta((\log4/\delta)/\kappa^2)$, then using such an estimation to serve the role of $\kappa$ in our algorithm SWEET will result in the same order of the sample complexity as our current result (with the knowledge of $\kappa$). Therefore, we can add the above estimation step to SWEET and use the estimated lower bound to replace the role of $\kappa$, and the resulting total sample complexity remains the same orderwisely.

---

### Official Review · Reviewer_5h43 · 2022-10-24

**Confidence:** 3
**Correctness:** 3
**Technical Novelty And Significance:** 3
**Empirical Novelty And Significance:** Not applicable
**Recommendation:** 6

**Clarity, Quality, Novelty And Reproducibility:**

I couldn’t verify the correctness of Theorem 1. If I understand correctly, there are at least some unstated assumptions regarding the safety constraint $(c^*, \tau^*)$:

- In the paragraph after Eq. (16) (page 16), $\gamma$ is defined by $(\Delta_{c^*}-3\mathfrak{U})/\Delta_{c^*}$. But to make $\pi^\gamma$ well-defined, $\gamma$ must belongs to $[0,1]$. It’s unclear to me how to establish such a relationship. In particular, could the authors elaborate why $\Delta_{c^*}>0$ and $\Delta_{c^*}\ge 3\mathfrak{U}$? It seems to me that the choice of $c^\star$ and $\tau^\star$ is arbitrary.
- Fundamentally, I don’t think Theorem 1 can be true without any assumptions on $(c^\star,\tau^\star)$. Imagine that $\tau^\star=\min_\pi V^\pi_{P^\star,c^\star}$. Then finding an $(c^\star,\tau^\star)$-safe policy is equivalent to finding the exact optimal policy under the utility $c^\star$, which is impossible under finite samples.

Some informal statements in this paper are not well-supported:

- In related work section, this paper claims “Chen et al. (2022) studies RF-RL with more general function approximation, but their results cannot recover the upper bound we have for the low-rank MDPs” with no further discussions. Could the author elaborate?
- In the introduction, this paper claims “Remarkably, the sample complexities under both algorithms match or even outperform the state of the art of their constraint-free counterparts up to some constant factors, proving that safety constraint incurs nearly no additional sample complexity for RF-RL.” While the first part of this statement is supported by Theorem 1, I don’t see how the second part is supported --- the safety constraint always introduces an extra multiplicative factor $1/\Delta_{min}$ to the sample complexity, which is exactly the additional cost of having a safety constraint.



**Strength And Weaknesses:**

Strengths:

- This paper proves tight sample complexity bounds for the safe RF-RL setting. This bound matches RF-RL without safety guarantees up to some constant factors.
- The algorithmic design principle (Alg. 1) is applicable to both tabular and linear MDPs.

Weaknesses:

- I have concerns about the paper's soundness (details in below).
- I don’t think the autonomous driving example mentioned in the introduction is a good way to motivate this paper. In the exploration phase of the SWEET algorithm, the safe policy set is essentially a mixture between the safe baseline policy and some rather random policy, so that the average constraint is smaller than the threshold.  In the autonomous driving example, this policy could be crashing the car with some probability and using the policy baseline otherwise, which doesn’t sound the right type of guarantees.


**Summary Of The Paper:**

This paper studies the reward-free reinforcement learning setting with additional safety constraint. Assuming the access to a safe baseline policy, the SWEET algorithm proposed in this paper meets the safety requirement throughout the learning process with high probability. After the reward-free exploration phase, the algorithm can output a near-optimal policy for any  reward function and (new) safety constraint. The SWEET algorithm achieves tight sample complexity guarantees on both tabular and linear MDPs.

**Summary Of The Review:**

My main concerns are the correctness/clarify of this paper. As a result, I recommend a rejection for now, but I will be happy to raise my score if my concerns are addressed.

===== after rebuttal ====

The authors addressed my concerns regarding the correctness of the theorems. Therefore I'll raise my score accordingly.

---

> ### Author Response · Authors · 2022-11-16
> **Response to Reviewer 5h43, Part 1**
>
> We thank the reviewer for the careful reading of our work and thoughtful comments. We provide
> our responses in detail below.
>
> **Q1**: I don’t think the autonomous driving example mentioned in the introduction is a good way to motivate this paper. In the exploration phase of the SWEET algorithm, the safe policy set is essentially a mixture between the safe baseline policy and some rather random policy, so that the average constraint is smaller than the threshold. In the autonomous driving example, this policy could be crashing the car with some probability and using the policy baseline otherwise, which doesn’t sound the right type of guarantees.
>
>
> **A1**: We agree with the reviewer that autonomous driving should satisfy the action constraint rather than the average cost constraint. We use this example to justify the need of safe exploration in reward-free RL, while the definition of safety depends on specific applications. We did not intend to use the autonomous driving example to motivate all aspects of our work. We will clarify this in our revision.
>
>
>
> **Q2**: I couldn’t verify the correctness of Theorem 1. If I understand correctly, there are at least some unstated assumptions regarding the safety constraint: $(c^*,\tau^*)$.
> In the paragraph after Eq. (16) (page 16), $\gamma$ is defined by $(\Delta_{c^*}-3\mathtt{U})/\Delta_{c^*}$. But to make $\pi^{\gamma}$ well-defined, $\gamma$ must belongs to [0,1]. It’s unclear to me how to establish such a relationship. In particular, could the authors elaborate why $\Delta_{c}>0$ and $\Delta_{c^*}\geq 3\mathtt{U}$? It seems to me that the choice of $c^*$ and $\tau^*$ is arbitrary.
> Fundamentally, I don’t think Theorem 1 can be true without any assumptions on $(c^*,\tau^*)$. Imagine that
> $\tau^*=\min_{\pi}V_{P^*,c^*}^{\pi}$. Then finding an $(c^*,\tau^*)$-safe policy is equivalent to finding the exact optimal policy under the utility $c^*$, which is impossible under finite samples.
>
> **A2**: Thanks for the comments. Yes, Theorem 1 holds under Assumption 1, which guarantees that the problem is well-defined and well-conditioned. Otherwise, if Assumption 1 does not hold, either there is no feasible solution, or the safe exploration policy set is too small to allow us to identify a near-optimal safe policy in the planning phase.
>
> Under Assumption 1, we have $\tau^*-\min_{\pi}V_{c^*}^{\pi} = \Delta_{c^*}\geq \Delta_{\min}>0$. In addition, by the definition of $\mathtt{U}$, we have $\mathtt{U}\leq\frac{\epsilon\Delta_{\min}}{5}<\Delta_{c^*}/3$ when $\epsilon<1$. Therefore, the choice of $\gamma$ is well-defined.
>
> **Q3**: In related work section, this paper claims “Chen et al. (2022) studies RF-RL with more general function approximation, but their results cannot recover the upper bound we have for the low-rank MDPs” with no further discussions. Could the author elaborate?
>
> **A3**: Thanks for the comment. Chen et al. (2022) studied model-free reward-free RL with general function approximation, and showed that their results specialize to low-rank MDP with a sample complexity of $O(H^6d^3A/\epsilon^2)$, which has higher order dependence on $H$ and lower order dependence on $d$ and $A$ compared to our results. In addition, Chen et al. (2002) requires stronger computational oracles compared to us. This is because they construct model confidence sets at every iteration and select the model with largest "value function". Both construction of confidence set and selection of the model is hard to implement.

---

> > ### Author Response · Authors · 2022-11-16
> > **Response to Reviewer 5h43, Part 2**
> >
> > **Q4**: In the introduction, this paper claims “Remarkably, the sample complexities under both algorithms match or even outperform the state of the art of their constraint-free counterparts up to some constant factors, proving that safety constraint incurs nearly no additional sample complexity for RF-RL.” While the first part of this statement is supported by Theorem 1, I don’t see how the second part is supported --- the safety constraint always introduces an extra multiplicative factor $1/\Delta_{\min}$ to the sample complexity, which is exactly the additional cost of having a safety constraint.
> >
> > **A4**: Thanks for the comment. Our statement of ``safety incurs nearly no additional sample complexity" focuses on whether the dependence of the sample complexity on the system parameters $H,S,A,d$ increases due to the safety constraint. This is non-trivial, because intuitively, safe exploration in reward-free RL restricts the policy set that can be explored, but planning phase requires the agent to output a near-optimal policy for any reward and safety constraint. Thus, it is highly possible that the near-optimal policy does not lie in the exploration policy set. This mismatch may significantly increase the sample complexity in terms of the model parameters such as $H,d,S,A$. However, in Theorems 2 and 3, we show that the order dependence of the sample complexity on $H,d,S,A$ does not change or may be even lower. That is why in the submission, we state that "the sample complexities under both algorithms match or even outperform the state of the art in their constraint-free counterparts **up to some constant factors**, proving that safety
> > constraint hardly increases the sample complexity for RF-RL."
> >
> > Regarding the dependence of the sample complexity on $\Delta(c,\tau), \Delta_{\min} $ and $\kappa$, we agree with the reviewer that they can cause increase in the sample complexity, which is expected and somewhat unavoidable due to the safety constraints.

---

> > ### Comment · Reviewer_5h43 · 2022-11-16
> > **Follow-up questions**
> >
> > Thanks for the clear response to my questions. I still have the following question regarding Theorem 1:
> >
> > Does theorem 1 require Assumption 1 to hold for $(c^\star,\tau^\star)$ as well? It seems that the statement of Assumption 1 is about the constraint $(c,\tau)$ in the exploration phase, and the paper claims that the algorithm "obtain an $\epsilon$-optimal $(c^\star, \tau^\star)$-safe policy for any given reward $r^\star$ and constraint $(c^\star, \tau^\star)$ in the planning phase."

---

> > > ### Author Response · Authors · 2022-11-17
> > > **Response to Reviewer 5h43, Follow-up**
> > >
> > > Q: I still have the following question regarding Theorem 1: Does theorem 1 require Assumption 1 to hold for $(c^*,\tau^*)$ as well? It seems that the statement of Assumption 1 is about the constraint $(c,\tau)$ in the exploration phase, and the paper claims that the algorithm "obtain an $\epsilon$-optimal $(c^*,\tau^*)$-safe policy for any given reward $r^*$ and constraint $(c^*,\tau^*)$ in the planning phase."
> > >
> > > A: Many thanks for the question. Yes. Theorem 1 requires Assumption 1 to hold for $(c^*,\tau^*)$. In our revised version, we have clarified in Assumption 1 that all $(c,\tau)$ in both the exploration and planning phases should satisfy the safety margin condition. Further, our statement should also require this assumption to hold for $(c^*,\tau^*)$ in the planning phase. We have clarified this in all our statements in the revision.
> > >
> > > We thank the reviewer again for the helpful comments and suggestions for our work. If our response resolves your concerns to a satisfactory level, we kindly ask the reviewer to consider raising the rating of our work. Certainly, we are more than happy to address any further questions that you may have.

---

### Official Review · Reviewer_3Lc4 · 2022-10-24

**Confidence:** 4
**Correctness:** 4
**Technical Novelty And Significance:** 3
**Empirical Novelty And Significance:** 3
**Recommendation:** 6

**Clarity, Quality, Novelty And Reproducibility:**

- This paper is well-written and easy to follow. studying safety in reward-free exploration is interesting and useful
- There are some concerns I mentioned in the weakness where the author needs to further clarify
- In Definition 6, I would suggest the author to includes $\pi'$ in the notation $\pi^\gamma$, since $\pi^\gamma = \gamma \pi + (1 - \gamma) \pi'$, informally speaking.

**Strength And Weaknesses:**

### Strength:

- Defining the convexity in uncertainty quantification is interesting.

### Weakness:

- I'm not sure if Assumption 1 will hold for **any given constraint $(c, \tau)$. For example, if $\tau = \min_{\pi} V_{P^\*, c}^\pi$, then $\Delta(c, \tau) = 0$.
- Following the previous question, does the author want to express that $\tau$ is a selected parameter in the exploration phase, then it's not clear how to select the $\tau$
- In the stopping condition we need $T \ge \Delta(c, \tau)$. Does that mean we need some prior knowledge of the magnitude of the safety margin?


**Summary Of The Paper:**

This paper studies the reward-free RL in general MDP settings with safety constrain. The author then extend their analysis to tabular MDP and low-rank MDP for a detailed result.

**Summary Of The Review:**

This paper provides some new thoughts on reward-free RL, especially on safety concerns. The methodology proposed by the authors is quite interesting. However, It has some issues regarding the Assumptions, parameter selection, and knowledge about the safety margin. Therefore I would suggest a marginal rejection but would like to change my score after the discussion if the authors well clarify my concerns.


***
I've carefully read the author's response and other reviews. The authors' response addressed some of my concerns thus I would raise my score to 6

---

> ### Author Response · Authors · 2022-11-16
> **Response to Reviewer 3Lc4**
>
> We thank the reviewer for the careful reading of our work and thoughtful comments. We provide
> our responses in detail below.
>
> **Q1**: I'm not sure if Assumption 1 will hold for any given constrain $(c,\tau)$. For example, if $\tau = \min_{\pi}V_{P^*,c}^{\pi}$, then $\Delta(c,\tau) = 0.$
>
> **A1**: Thanks for the question. We clarify that Assumption 1 does not hold for all constraints. We need such an assumption to hold for both the exploration and the planning phases. Otherwise, the problem is not well-defined or well-conditioned. This is because if Assumption 1 does not hold, either there is no feasible solution, or the safe exploration policy set is too small to allow us to find a near-optimal safe policy in planning.
>
>
> **Q2**: Following the previous question, does the author want to express that $\tau$ is a selected parameter in the exploration phase, then it's not clear how to select the $\tau$.
>
> **A2**: $\tau$ is the threshold for the cost constraint. As defined in Definition 1, for given  MDP $\mathcal{M}^*=(\mathcal{S},\mathcal{A},P^*,H,s_1)$, a policy $\pi$ is $(c,\tau)$-safe if $V_{P^*,c}^{\pi}\leq \tau$. One objective of safe exploration is to ensure all policies in the exploration phase are $(c,\tau)$-safe for given $(c,\tau)$. Hence, $\tau$ is the target system safety requirement and is provided to the agent by the system designer.
>
>
> **Q3**: In the stopping condition we need $\mathtt{T}\geq \Delta(c,\tau)$. Does that mean we need some prior knowledge of the magnitude of the safety margin?
>
>  **A3**: First, we clarify that we need $\mathtt{T}<\Delta(c,\tau)$ rather than $\mathtt{T}\geq \Delta(c,\tau)$. We note that the results in Theorem 1 hold when the safety margin $\Delta(c,\tau)$ is known beforehand. However, if $\Delta(c,\tau)$ is unknown, we can always replace it by the safety gap of the baseline policy (i.e., $\kappa$) in the algorithm, as $\Delta(c,\tau)\geq \kappa$ according to Assumption 1, and the result in Theorem 1 holds similarly. We note that for a given baseline policy, its safety gap $\kappa$ can always be efficiently assessed by executing the policy and calculating the average cost among the observed trajectories.
>
> **Q4**: In Definition 6, I would suggest the author to includes $\pi'$ in the notation, since $\pi^\gamma=\gamma \pi+(1-\gamma)\pi'$ informally speaking.
>
> **A4**: Thanks for the suggestion. We have modified the notation slightly to include both $\pi$ and $\pi'$ in the revision.

---

> ### Author Response · Authors · 2022-12-01
> **Your feedback is important to us**
>
> Dear Reviewer 3Lc4,
>
> This is a friendly reminder that we have submitted our response to your review comments and uploaded a revision of the paper two weeks ago, and we will appreciate very much if you could give us any feedback. If our response resolves your concerns, we kindly ask you to consider raising the rating of our work. We are also more than happy to answer your further questions. Thank you very much for your time and efforts!

---

### Official Review · Reviewer_kzdP · 2022-10-24

**Confidence:** 3
**Correctness:** 4
**Technical Novelty And Significance:** 2
**Empirical Novelty And Significance:** Not applicable
**Recommendation:** 6

**Clarity, Quality, Novelty And Reproducibility:**

This paper is clearly written and well executed. In my opinion, the idea behind the safe exploration is not novel. The theoretical results in this paper are supported by complete proofs.

**Strength And Weaknesses:**

**Strengths:**

1 The proposed algorithmic framework SWEET is general, and can be applied to the tabular and the low-rank MDP settings to obtain start-of-the-art sample complexity guarantees.

**Weaknesses:**

1 The ideas of safe exploration based on known baseline policy, approximation error function and estimated safe policy set are not novel. There have been several works that study bandit/RL with safe/conservative exploration.

For example,
[1] Kazerouni, Abbas, et al. "Conservative contextual linear bandits." Advances in Neural Information Processing Systems 30 (2017).
[2] Yang, Yunchang, et al. "A unified framework for conservative exploration." arXiv preprint arXiv:2106.11692 (2021).
[3] Amani, Sanae, Christos Thrampoulidis, and Lin Yang. "Safe reinforcement learning with linear function approximation." International Conference on Machine Learning. PMLR, 2021.

It is well known that in the regret minimization setting, the requirement of safe exploration will only incur an additional $O(1)$ regret (independent of the number of timesteps $T$ played in the RL game). Hence, it is also not surprising that safe exploration will not incur additional sample complexity for reward-free RL.

The technical novelties of the proposed techniques, e.g., approximation error function and estimated safe policy set, are unclear to me. In addition, the results of this paper reply on known baseline policy $\pi_0$ and safety margin $\kappa$. While a known baseline policy is widely used in the safe bandit/RL literature, the algorithm design and results in this paper heavily depend on the prior knowledge of safety margin $\kappa$, and are expected.

2 For the low-rank MDP setting, the authors obtain a better sample complexity result under the safe exploration constraint than all existing constraint-free algorithms (without proposing new techniques for low-rank MDP). This result is surprising. Can the authors discuss what techniques used in their algorithm design/analysis enable them to achieve a tighter sample complexity?


**Summary Of The Paper:**

This paper studies reward-free RL with safe exploration. The authors consider the scenario where a
safe baseline policy is given beforehand, propose a unified algorithmic framework called SWEET, and instantiate the SWEET framework to the tabular and low-rank MDP settings. Their algorithms utilize the concavity and continuity of the truncated value functions to achieve zero constraint violation with high probability. The sample complexities of their algorithms match or outperform their constraint-free counterparts, which shows that safety constraint hardly increases the sample complexity for reward-free RL.


**Summary Of The Review:**

This paper is well executed and gives results for both tabular and low-rank MDP settings. My concern mainly falls on the novelty of the used techniques, e.g., approximation error function and estimated safe policy set, and the reliance on the prior knowledge of safety margin $\kappa$. In my opinion, the finding of “safe exploration incurs nearly no additional sample complexity for reward-free RL” is not surprising and similar conclusion has been obtained in the regret minimization bandit/RL. Due to the above reasons, I give borderline rejection.

If the authors can address my concerns on technical novelty (what techniques/findings are novel and unique for reward-free RL compared to existing literature of bandit/RL with safe exploration), I am ready to raise my score.

====After reading the authors' rebuttal====

Thank the authors for their response.

I appreciate that the authors provide a new method which does not need prior knowledge on the safety margin $\kappa$, which relieves my concern on prior knowledge of $\kappa$.

For the novelty, I am not sure if the reward-free-exploration safe RL brings significant unique challenges than existing safe/conservative RL.
In the rebuttal, the authors explained that, the reward-free-exploration safe RL needs to both ensure safety for a given constraint during exploration and find a near-optimal policy *for arbitrary reward and constraint* during planning.
However, since the algorithm can first estimate a sufficiently accurate transition model during exploration and then do planning, I am not sure if finding a near-optimal policy *for arbitrary reward and constraint* will pose significant unique challenges (at least for the tabular setting).
Therefore, I think the finding that "safe exploration incurs nearly no additional sample complexity for reward-free RL" is not surprising, since we already know that safe exploration incurs nearly no additional sample complexity for standard RL.

For the proposed technique, while the authors explained that the truncated value function is critical to their analysis, I did not get the significance and novelty of this technique very much. In my understanding, I think that SWEET does clipping on the value function (instead of the Q-value function) with the universal reward upper bound $1$, while RepUCB does not do clipping on value functions, and thus, SWEET improves the dependency of $H$. This technique does not look very significant and novel to me. The authors may consider to highlight more on how this truncated value function preserves concavity and continuity and brings large advantages in analysis compared to the conventional clipping technique in their revision.

====After reading the authors' additional explanations and discussions=====

The authors' additional explanations relieve my concerns on the unique challenges of reward-free safe RL. In addition, I appreciate the authors' efforts in designing a new algorithmic strategy which does not require the prior knowledge of $\kappa$. So I raised my score from 5 to 6.

I suggest the authors to discuss the unique challenges of reward-free safe RL in a deeper analysis level (instead of just mentioning the differences of problem settings), and elaborate how the concavity of the truncated value function helps to handle these challenges more specifically in their revision. For the current version, it is not easy for readers (at least, to me) to quickly understand the challenges of reward-free safe RL and the novelty of the truncated value function compared to conventional clipping technique.

---

> ### Author Response · Authors · 2022-11-16
> **Response to Reviewer kzdP, Part 1**
>
> We thank the reviewer for the careful reading of our work and thoughtful comments. Below, we
> clarify the major differences between the safe exploration problem studied in this work and existing
> works, elaborate our technical novelties, and explain the major techniques we develop to achieve
> SOTA sample-complexity in low-rank MDPs.
>
> **Q1**: My concern mainly falls on the novelty of the used techniques, e.g., approximation error function and estimated safe policy set, and the reliance on the prior knowledge of safety margin $\kappa$. In my opinion, the finding of "safe exploration incurs nearly no additional sample complexity for reward-free RL is not surprising and similar conclusion has been obtained in the regret minimization bandit/RL.
>
> **A1**: **Key difference of safety requirement and unique challenges:** We clarify that our safe exploration requirement is fundamentally different from the typical safe exploration setting studied in the literature (including those pointed out by the reviewer). Existing literature on safe bandits/RL aims to find a near-optimal safe policy for **a given set of reward and constraints**. Hence, it suffices to control the learning performance (e.g. estimation error) on the state-action pairs or trajectories whose rewards or costs are large. In contrast, our setting not only requires safe exploration for a given constraint in the exploration phase, but also (and more importantly) requires to find a near-optimal policy for **arbitrary reward and constraint** in the planning phase without querying any new samples. Thus, the key difficulty lies in the mismatch between constraints in exploring and planning phases: the given safety constraint in exploration phase significantly limits the size of the safe exploration policy set. Thus, it is very likely the near-optimal policy in planning phase does not lie in this set (because constraint in planning phase can be arbitrary) and hence are not well explored during exploration. Informally speaking, **our setting requires to estimate the performance of an arbitrary unknown policy by exploring only a small and fixed set of policies.** Such a unique challenge never appears in the existing literature, and clearly requires our algorithm to have a significantly different design.
>
>
> **Technical novelty in the proposed algorithm:** In order to deal with the challenge of constraint mismatch between exploration and planning phases and to be able to determine a near-optimal policy that is outside the safe exploration policy set, we construct a novel approximation error function to ensure that it possesses the **concavity** property, which is critical. We show that, with such concavity (together with continuity), by controlling the approximation error function within **a small set of policies**, the maximum value of the approximation error function over the **entire policy set** is also bounded. More details can be found in page 6, section 3.2 and Lemma 2. Such a design is new and motivated by the unique constraint mismatch challenge in the reward-free setting, and is not covered in conventional safe bandits/RL in the regret minimization setting. Moreover, our design is also different from the reward-free RL literature. Here, we employ a clipping strategy on **V-function** rather than **Q-function** (which is conventionally adopted in the reward-free literature) to ensure that the resulting exploration driven function is **concave**. Such a novel clipping technique also plays a critical role in reducing the sample complexity for low-rank MDPs compared with all existing constraint-free algorithms, as we will elaborate in  response A2.
>
> **Technical novelty in analysis:** Our technical novelty in analysis can be summarized as follows: First, we prove that truncated value function is concave on the policy space, which is critical to ensure safety in both exploration and planning. Further, we show that the concave approximation error function is bounded on the whole policy set if it is bounded within a small policy set. In addition, we developed a new lemma of law of total variance (Lemma 6) for stochastic policy. This is crucial since the policies used in exploration phase are highly likely not deterministic, which is very different from the constraint-free reward-free RL. For technique novelties of low-rank MDP, please see our response in A2.

---

> > ### Author Response · Authors · 2022-11-16
> > **Response to Reviewer kzdP, Part 2**
> >
> > **Why the result depends on prior knowledge of safety margin:** As we comment above, our setting requires to estimate the performance of an arbitrary unknown policy by well exploring only a small and fixed set of policies. Towards this end, prior knowledge of safety margin allows us to explore those unknown policies to certain extent without violating the constraint. Specifically, by assigning small probabilities for those under-explored state-action pairs (tabular case) or trajectories (low-rank case) when designing the exploration policies, SWEET can remain safe due to the existence of safety margin, and efficiently explore the environment simultaneously.
> >
> >
> > We note that the requirement of knowing the safety gap $\kappa$ of the baseline policy is reasonable, as $\kappa$ for a given baseline policy can always be accurately estimated by executing the policy and calculating the average cost among the observed trajectories. We note that similar reliance has been assumed in the literature, e.g., [1].
> >
> > **Q2**:  For the low-rank MDP setting, the authors obtain a better sample complexity result under the safe exploration constraint than all existing constraint-free algorithms (without proposing new techniques for low-rank MDP). This result is surprising. Can the authors discuss what techniques used in their algorithm design/analysis enable them to achieve a tighter sample complexity?
> >
> > **A2**: Our algorithm indeed has several new components that set up apart from other reward-free algorithms for low-rank MDPs and enable us to obtain improved sample complexity up to certain constant factors.
> >
> > First, compared with MOFFLE [2], we adopt a model-based approach {while} MOFFLE {uses a model-free approach. It is known that model-based methods are more sample-efficient than model-free methods in general. } Moreover, SWEET only requires two steps of uniform exploration for each exploration policy, while MOFFLE requires {three steps}, leading to higher order dependence on $A$.  Second, compared with FLAMBE [3], we only collect $H$ trajectories for each exploration policy, while FLAMBE collects $O(1/\epsilon^4)$ of them, which depends on the accuracy level $\epsilon$ and can be undesirably large.  Third, we adopt a novel termination condition, which ensures that an accurate model estimate $\hat{P}^{(n)}$ can be identified efficiently, and further guarantees a near-optimal policy under any reward and constraint in planning.
> >
> > Compared with reward-known algorithm Rep-UCB, we develop new exploration strategy due to the reward-free setting. Specifically, we use $\hat{b}$ as the "exploration-driven reward" (See section 5.2) while in Rep-UCB, $\hat{b}$ is used for optimistic action selection. More importantly, we have designed a tighter approximation error of the estimated model through a novel **truncated value function** (defined in Section 3.3), which is crucial for achieving reduced sample complexity. This is because the truncated valued function can provide a tighter bound on the estimation error of any value functions $|V_{\hat{P},r}^{\pi}-V_{P^*,r}^{\pi}|$, leading to more effective exploration for uncertainty reduction.
> >
> > [1] Tao Liu, Ruida Zhou, Dileep Kalathil, Panganamala Kumar, and Chao Tian. Learning policies
> > with zero or bounded constraint violation for constrained mdps. Advances in Neural Information
> > Processing Systems, 34, 2021.
> >
> >
> > [2] Aditya Modi, Jinglin Chen, Akshay Krishnamurthy, Nan Jiang, and Alekh Agarwal. Model-free
> > representation learning and exploration in low-rank mdps. arXiv preprint arXiv:2102.07035,
> > 2021.
> >
> >
> > [3] Alekh Agarwal, Sham Kakade, Akshay Krishnamurthy, and Wen Sun. Flambe: Structural complexity and representation learning of low rank mdps. In H. Larochelle, M. Ranzato, R. Hadsell,
> > M. F. Balcan, and H. Lin (eds.), Advances in Neural Information Processing Systems, volume 33,
> > pp. 20095–20107. Curran Associates, Inc., 2020.

---

> > > ### Comment · Reviewer_kzdP · 2022-11-22
> > > **Reply to the Rebuttal**
> > >
> > > Thank you for your reply!
> > >
> > > Could you give any lower bound or discussions to show the necessity prior knowledge of safety margin $\kappa$? I think the algorithm design and results highly depend on this prior knowledge.
> > >
> > > It seems that the proposed algorithm can also achieve a tighter sample complexity than RepUCB under the same setting as that of RepUCB (i.e., the standard best policy identification setting).
> > > Could you give a more detailed explanation on why the truncated value function can improve the result of RepUCB?

---

> > > > ### Author Response · Authors · 2022-11-23
> > > > **Response to Follow-up Questions**
> > > >
> > > > **Q1:** Could you give any lower bound or discussions to show the necessity prior knowledge of safety margin $\kappa$? I think the algorithm design and results highly depend on this prior knowledge.
> > > >
> > > > **A1:** Great question! Let us explain below.
> > > >
> > > > **A design to remove the dependence on the knowledge of $\kappa$:** In fact, we can provide the following strategy to estimate a lower confidence bound (LCB) of $\kappa$ such that (1) the estimation of $\kappa$ does not cause order-wise increase of sample complexity; and (2) using such an estimated lower bound to serve the role of $\kappa$ will not cause order-wise change of the sample complexity. In this way, our algorithm won't need the knowledge of the safety margin $\kappa$. We briefly describe such a strategy below.
> > > >
> > > > At each episode $k$, if the agent executes the baseline policy, then let $C_k$ denote the random total cost of the baseline policy over the entire episode, which takes value witin $[0,1]$. Note that $\mathbb{E}[C_k] = V_{P^*,c}^{\pi^b} = \tau -\kappa$. Then, to obtain an LCB of $\kappa$, the agent will continuously execute the baseline policy to take sample trajectories and collect the corresponding $C_k$. At the end of each episode $K$, the agent checks whether the condition $\tau-\frac{\sum_{k=1}^K C_k}{K}>2\sqrt{\frac{\log 4/\delta}{K}}$ is satisfied. (Since the LHS of the inequality will converge to $\kappa$ by concentration, and the RHS will converge to zero, the condition will be satisfied at a certain $K$). If the condition is satisfied at certain $K_0$, then the agent terminates this procedure and use $\sqrt{\frac{\log 4/\delta}{K_0}}$ to serve as the LCB of $\kappa$.
> > > >
> > > > We next provide a brief analysis as follows. Suppose the above estimation strategy terminates at episode $K_0$. By concentration inequality, we can show that with probability at least $1-\delta$,
> > > >
> > > > $\frac{\sum_{k=1}^{K_0}C_k}{K_0}\geq \tau-\kappa - \sqrt{\frac{\log 4/\delta}{K_0}}$, and $\frac{\sum_{k=1}^{K_0-1}C_k}{K_0-1}\leq  \tau-\kappa + \sqrt{\frac{\log 4/\delta}{K_0-1}}.$
> > > >
> > > > On the other hand, by the termination condition, we know that
> > > >
> > > > $\frac{\sum_{k=1}^{K_0}C_k}{K_0}< \tau - 2\sqrt{\frac{\log 4/\delta}{K_0}}$, and $\frac{\sum_{k=1}^{K_0-1}C_k}{K_0-1}\geq \tau - 2\sqrt{\frac{\log 4/\delta}{K_0-1}}$.
> > > >
> > > > Combining these inequalities, we have $\sqrt{\frac{\log4/\delta}{K_0}}<\kappa\leq 3\sqrt{\frac{\log4/\delta}{K_0-1}}$. Therefore, the sample complexity (i.e., the total episodes we collect) for estimating such a lower bound is at most $O((\log4/\delta)/\kappa^2)$. Further, $\sqrt{\frac{\log4/\delta}{K_0}}$ is indeed an LCB of $\kappa$. Since $K_0 = \Theta((\log4/\delta)/\kappa^2)$, then using such an estimation to serve the role of $\kappa$ in our algorithm SWEET will result in the same order of the sample complexity as our current result (with the knowledge of $\kappa$). Therefore, we can add the above estimation step to SWEET and use the estimated lower bound to replace the role of $\kappa$, and the resulting total sample complexity remains the same orderwisely.
> > > >
> > > > **Q2:** It seems that the proposed algorithm can also achieve a tighter sample complexity than RepUCB under the same setting as that of RepUCB (i.e., the standard best policy identification setting). Could you give a more detailed explanation on why the truncated value function can improve the result of RepUCB?
> > > >
> > > > **A2:** By the normalization assumption of the reward, the value functions are within $[0,1]$, and hence the value function is always bounded by 1. Therefore, the truncated value function with such an upper bound may yield a tighter bound compared with the standard value function proposed in RepUCB. Such a benefit can be seen from the proof of Lemma 13, where we prove that due to the truncation, the difference between $V_{\hat{P}^n,\hat{b}^n}^{\pi^n} $
> > > > and $\bar{V}_{P^*,\hat{b}^n}^{\pi^n}$ is upper bounded by the total variation distance of the estimated model at each step.
> > > >
> > > > Mathematically, $ V_{P^*,f^n}^{\pi^n} \geq |V_{\hat{P}^n,\hat{b}^n}^{\pi^n} - \bar{V}_{P^*,\hat{b}^n}^{\pi^n}|$
> > > >
> > > > In contrast, without this truncation, RepUCB have an additional $H$ factor in their bound as $ HV_{P^*,f^n}^{\pi^n} \geq |V_{\hat{P}^n,\hat{b}^n}^{\pi^n} - \bar{V}_{P^*,\hat{b}^n}^{\pi^n}|$. Therefore, due to the truncation, the summation of the truncated value functions (the last step) is upper bounded by $Hd^2A\sqrt{N}$, whereas RepUCB has a resulting upper bound of $H^2d^2A\sqrt{N}$ on the corresponding quantity.
> > > > Therefore, we have a tighter sample complexity.
> > > >
> > > >
> > > > We thank the reviewer again for continuing to devote the time and effort on our paper. If our response resolves your concerns to a satisfactory level, we kindly ask the reviewer to consider raising the rating of our work. Certainly, we are more than happy to address any further questions
> > > > that you may have.

---

> > > > > ### Comment · Reviewer_kzdP · 2022-11-26
> > > > > **Further Questions on the Improvement over RepUCB**
> > > > >
> > > > > Thank you for providing this new algorithmic strategy! My concern on the prior knowledge of $\kappa$ is largely relieved.
> > > > >
> > > > > Could you elaborate the difference between the proposed truncated value function technique and the conventional clipping technique in the RL theory literature. After reading the current version of the paper, I did not understand the difference very clearly.
> > > > >
> > > > > Does the improvement of factor $H$ over RepUCB come from some technique similar to "law of total variance"? It seems that you do not use the Bernsern-type inequality, and how do you achieve this? Could you explain more on how to obtain a tighter dependency on $H$ by just doing the truncation on value functions?

---

> > > > > > ### Author Response · Authors · 2022-11-28
> > > > > > **Response to Reviewer kzdP**
> > > > > >
> > > > > > **Q1**: Could you elaborate the difference between the proposed truncated value function technique and the conventional clipping technique in the RL theory literature. After reading the current version of the paper, I did not understand the difference very clearly.
> > > > > >
> > > > > >
> > > > > > **A1**: Thanks for the question. First, the mathematical expressions of these two clipping techniques are different. Conventional clipping technique is applied on the $Q$ function. i.e. $\hat{Q}^{\pi}(s,a) = \min(Q^{\pi}(s,a),1)$. Hence, the corresponding value function is $\hat{V}^{\pi}(s) = \mathbb{E}{\pi}[\hat{Q}^{\pi}(s,a)]$. However, in this work, the truncated value function is defined by taking expectation first and then clipping on the value function, i.e. $\bar{V}^{\pi}(s) = \min(\mathbb{E}_{\pi}[Q^{\pi}(s,a)],1)$. Intuitively, for the same $Q^{\pi}(s,a)$ and $\pi$, by truncating in those two different ways, we have $\bar{V}^{\pi}(s)\geq \hat{V}^{\pi}(s)$. As verified in Lemma 3, the resulting $\bar{V}^{\pi}(s)$ is concave on the policy space, while such property can not be established for $\hat{V}^{\pi}(s)$. Therefore, by designing the truncated value function in this way, we ensure the desired concavity on $\bar{V}^{\pi}(s)$, which is critical to achieve the safe planning requirement in our algorithm design.
> > > > > >
> > > > > >
> > > > > >
> > > > > > **Q2**: Does the improvement of factor $H$ over RepUCB come from some technique similar to "law of total variance"? It seems that you do not use the Bernsern-type inequality, and how do you achieve this? Could you explain more on how to obtain a tighter dependency on $H$ by just doing the truncation on value functions?
> > > > > >
> > > > > >
> > > > > >
> > > > > > **A2**: We note that Bernstein-type inequality and "law of total variance" can lead to improvement of factor $H$ in certain RL problems, such as [1][2]. However, we did not adopt such techniques in this work, as they cannot be easily applied to our analysis, mainly due to the MLE oracle in our algorithm design. The technique that improves the order of $H$ compared with RepUCB is the truncated value function. In fact, the truncated value function provides a tighter bound on the estimation error (see Lemma 13 in our work) compared with the standard value function proposed in RepUCB (see Lemma 9 in [3] by replacing $\frac{1}{1-\gamma}$ with $H$). Specifically, by clipping the value function at one in each step, it ensures that the estimation error in the corresponding value function is within $[0,1]$. Since RepUCB does not have such clipping, the range of the corresponding estimation error is $[0,H]$. Thus, our estimation error decreases faster than that in RepUCB, which helps us to achieve tighter dependency on $H$.
> > > > > >
> > > > > > We thank the reviewer again for your careful reading of our work. We hope the responses address your questions satisfactorily, and will be more than happy to address any further questions you may have.
> > > > > >
> > > > > >
> > > > > > [1] Emilie Kaufmann, Pierre Ménard, Omar Darwiche Domingues, Anders Jonsson, Edouard Leurent,
> > > > > > and Michal Valko. Adaptive reward-free exploration. arXiv preprint arXiv:2006.06294, 2020.
> > > > > >
> > > > > > [2] Pierre Menard, Omar Darwiche Domingues, Anders Jonsson, Emilie Kaufmann, Edouard Leurent, ´
> > > > > > and Michal Valko. Fast active learning for pure exploration in reinforcement learning. In International Conference on Machine Learning, pp. 7599–7608. PMLR, 2021
> > > > > >
> > > > > > [3] Masatoshi Uehara, Xuezhou Zhang, and Wen Sun. Representation learning for online and offline RL
> > > > > > in low-rank MDPs. arXiv preprint arXiv:2110.04652, 2021

---

> > > > > > > ### Comment · Reviewer_kzdP · 2022-11-28
> > > > > > > **Thank the Authors for Their Response**
> > > > > > >
> > > > > > > Thank you for your response!
> > > > > > >
> > > > > > > Please see my updated reviews for my current opinion for this work.
> > > > > > >
> > > > > > > After rebuttal, my current rating is between 5 and 6. I would like to listen to the opinions of AC and other reviewers.

---

> ### Author Response · Authors · 2022-12-03
> **Author's Further Clarification**
>
> **Q1**: For the novelty, I am not sure if the reward-free-exploration safe RL brings significant unique challenges than existing safe/conservative RL. In the rebuttal, the authors explained that, the reward-free-exploration safe RL needs to both ensure safety for a given constraint during exploration and find a near-optimal policy for arbitrary reward and constraint during planning. However, **since the algorithm can first estimate a sufficiently accurate transition model during exploration** and then do planning, I am not sure if finding a near-optimal policy for arbitrary reward and constraint will pose significant unique challenges (at least for the tabular setting). Therefore, I think the finding that "safe exploration incurs nearly no additional sample complexity for reward-free RL" is not surprising, since we already know that safe exploration incurs nearly no additional sample complexity for standard RL.
>
>
> **A1**: As a matter of fact, the main difficulty of our problem exactly lies in how to estimate the transition model accurately. First note that RL algorithms estimate the transition model via exploration policies. Under the safety constraint, our exploration policies here cannot take samples from every part of state-action space to estimate their corresponding transition distributions, so that not every part of transition model can be estimated directly. But in the planning, since reward and constraint can be arbitrary, then if the near-optimal policy happens to be over the state-action space where the transition model isn't accurately estimated, without accurate knowledge of this part of transition model, there is typically no way for us to identify such a near-optimal policy. To resolve this difficulty, our design makes sure that although only a small set of safe exploration policies are used to estimate the model over limited range of state-action space, the estimation error of the value functions for all policies is well controlled.
>
> In contrast, in conventional safe RL, since planning is required only for a given reward and constraint, then the model estimation can just be estimated over the state-action space where reward is high/non-zero and constraint is satisfied, which is much easier.
>
> Given that our problem and conventional safe RL problem are very different (as we explain above), there is no logic connection that no additional sample complexity in conventional safe RL will imply the same for our setting.
>
>
>
> **Q2**: "For the proposed technique, while the authors explained that the truncated value function is critical to their analysis, I did not get the significance and novelty of this technique very much. In my understanding, I think that SWEET does clipping on the value function (instead of the Q-value function) with the universal reward upper bound 1, while RepUCB does not do clipping on value functions, and thus, SWEET improves the dependency of $H$. This technique does not look very significant and novel to me. The authors may consider to highlight more on how this truncated value function preserves concavity and continuity and brings large advantages in analysis compared to the conventional clipping technique in their revision."
>
>
> **A2**: We clarify that our paper does not sell the clipping technique as our contribution, and we didn't sell the performance improvement over RepUCB as our main contribution either. So although the reviewer doesn't find it novel, we kindly request the reviewer not to count this as the weakness of the paper.
>
> However, as the reviewer commented, finding that the clipping function (only with V-function) has concavity property and making use of such a nice property to handle the unique challenge arising in our problem is novel, which is not straightforward and has not been used in the literature.

---

> > ### Comment · Reviewer_kzdP · 2022-12-03
> > **Thank you for your additional response**
> >
> > Thank you for your clarification on the improvement over RepUCB. I asked questions about this just to try to figure out if any interesting technique is used.
> >
> > While the authors explained the differences between reward-free safe RL and standard safe RL in the level of problem settings, I did not very much get the unique (fundamental) challenges brought by reward-free safe RL in a deeper level, e.g., algorithm design and analysis.
> >
> > My current evaluation for this paper on borderline. I tend to keep my score and will engage in the discussion with other reviewers and AC.

---

> > > ### Author Response · Authors · 2022-12-05
> > > **Further Explanations on Challenges of Algorithm Design and Analysis**
> > >
> > > We truly thank the reviewer for devoting your time and efforts in discussing with us. We really appreciate it. Below we elaborate the technical challenges in algorithm design and analysis as well as how we deal with these challenges. Note that the notations we define below are mainly for explaining the idea to the reviewer, which are not used in the paper.
> > >
> > >
> > > Since our problem setting is very different from conventional safe RL, our design goal here is to make sure that although only **a small set of safe exploration policies** are used to estimate the model over limited range of state-action space, the model estimation error for **all policies** is well controlled.
> > >
> > >
> > >
> > > - **Algorithm Design:** In exploration phase, we need to design exploration policy (denoted as $\pi_E$ for convenience) that maximizes the model estimation error $Error$ and satisfies the safety constraint. Note that such $\pi_E$ can guarantee only the model estimation accuracy (i.e., provide a bound on $Error$) over a subset of state-action space (say $\mathcal{Z}_E$) because of the safety constraint. $\pi_E$ can still sample beyond $\mathcal{Z}_E$, but must be within $\epsilon$-violation of safety constraint. As a result, those state-actions outside $\mathcal{Z}_E$ won't be sufficient to make $Error$ small outside $\mathcal{Z}_E$. Now, in the planning phase, we need to bound the model estimation error $Error$ over the state-action space $\mathcal{Z}_P$ determined by any arbitrary reward and constraint, and hence $\mathcal{Z}_P \neq \mathcal{Z}_E$ (can be very different). Clearly, it is not possible to bound the error $Error$ over $\mathcal{Z}_P$ in planning by exploration error $Error$ (which has guarantee only over a smaller set $\mathcal{Z}_E$).
> > >
> > >
> > > In order to deal with the above challenge, the key idea is to design a new model estimation error metric (as exploration driven reward) that has some favorable structure to facilitate the guarantee transfer from exploration to planning. In our paper, we introduce a truncated error metric denoted as $TrunError$, which satisfies concavity as a function of policy. Now our goal becomes to bound $TrunError$ over $\mathcal{Z}_P$ by $TrunError$ over $\mathcal{Z}_E$ (where $\mathcal{Z}_P \neq \mathcal{Z}_E$). We can show that since $TrunError$ is concave, even a small exploration outside $\mathcal{Z}_E$ (within $\epsilon$ violation of safety constraint) can leverage the bound on $TrunError$ within $\mathcal{Z}_E$ to provide a good estimate for $TrunError$ outside $\mathcal{Z}_E$.
> > >
> > > Besides the above challenge, another design challenge as pointed by the reviewer is how to develop an algorithm that does not rely on the prior knowledge of safety margin. We appreciate that the reviewer's comment motivated us to think hard on this issue and were able to come up with a strategy that can remove such a requirement (as we explain in our previous response).
> > >
> > >
> > >
> > > - **Analysis:** Technically, the exploration phase solves $\max_{\pi} TrunError$ s.t. $\pi$ satisfies exploration safety constraint. The planning phase solves $\max_{\pi} TrunError$ s.t. $\pi$ satisfies planning constraint. Our goal here is to bound the optimal planning objective function by the optimal exploration objective function, given that the constraint sets for the two problems are very different. In general, in constrained optimization, as we change the constraint set, it is typically very difficult to analytically characterize how the optimal objective function value will change. In this paper, we exploit the concavity structure of the objective function $TrunError$ and was able to bound the optimal value of planning problem given the optimal value of exploration problem.
> > >
> > > The technical challenges when we instantiate to tabular and low-rank MDPs are summarized as follows: **(a)** For tabular MDP, since the policies used in exploration phase are highly likely not deterministic, a new lemma of law of total variance needs to be developed for stochastic policy for reward-free RL under tabular MDP. Existing literature has only such a property for deterministic policy. Our lemma establishes such a technical property. **(b)** For the low-rank case, due to the clipping technique introduced in $TrunError$, we need to develop that such a truncated value function is indeed a valid approximation error function of the estimated model, which we establish in Lemma 14. Also, in order to show the termination condition will be satisfied within finite samples, we develop Lemma 13, which proves that the summation of approximation error function is sublinear.

---

> > > > ### Comment · Reviewer_kzdP · 2022-12-05
> > > > **Thank you for your additional explanation. My concerns are relieved**
> > > >
> > > > Thank you for your additional explanation! My concerns on the unique challenges of reward-free safe RL are relieved. I raised my score from 5 to 6.
> > > >
> > > > I suggest the authors to discuss the unique challenges of reward-free safe RL in a deeper analysis level (instead of just mentioning the differences of problem settings), and elaborate how the concavity of the truncated value function helps to handle these challenges more specifically in their revision. For the current version, it is not easy for readers (at least, for me) to quickly understand the challenges of reward-free safe RL and the novelty of the truncated value function compared to the conventional clipping technique.

---

> > > > > ### Author Response · Authors · 2022-12-05
> > > > > **Thank you**
> > > > >
> > > > > Many thanks for your prompt feedback and raising the score! We will elaborate the challenges of our problem in a deeper analysis level in the revision.

---

### Decision · Program_Chairs · 2023-01-20

**Decision:**

Accept: poster

**Justification For Why Not Higher Score:**

The paper has some issues regarding the assumptions, parameter selection, and knowledge about the safety margin and did not discusses issues like computational complexity. The technical novelty is also limited.

**Justification For Why Not Lower Score:**

The paper stands out for the relevance of the main take:  when we have access to a safe baseline policy, reward-free RL exploration under safety constraints is not harder than unconstrained reward-free RL. The methodology is new and  the result is complete.

**Metareview: Summary, Strengths And Weaknesses:**

Summary:
This paper proposes a framework called SWEET for reward-free RL with safety constraint. In the setting, a safe baseline policy is given beforehand, and the MDP is low-rank. Their algorithms utilize the concavity and continuity of the truncated value functions to achieve zero constraint violation with high probability. The sample complexities of their algorithms match or outperform their constraint-free counterparts, which shows that the safety requirement does not increase the sample complexity of reward-free RL.

Strength:
- The proposed algorithmic framework SWEET is general, and can be applied to the tabular and the low-rank MDP settings to obtain start-of-the-art sample complexity guarantees.
- Technical assumes appears interesting.
- The algorithm do not need the knowledge of the safety margin.
- Paper clearly written.

Weakness:
-(Computational complexity) The paper does not discuss the computational tractability of the Algorithm 1;
- (Strength of the assumptions) The paper assumes to have access to a safe baseline policy and the presence of a positive safety margin.


**Note From Pc:**

if the above contains the word "oral" or "spotlight" please see: "oral" presentation means -> notable-top-5% and "spotlight" means -> notable-top-25%. As stated in our emails, we are disassociating presentation type from AC recommendations